# Learning the natural history of human disease with generative transformers

Artem Shmatko[1,2,3,13], Alexander Wolfgang Jung[2,4,5,6,13], Kumar Gaurav[2,13], Søren Brunak[4,7], Laust Hvas Mortensen[5,7,8], Ewan Birney[2✉], Tom Fitzgerald[2✉] & Moritz Gerstung[1,2,9,10,11,12✉]

Decision-making in healthcare relies on understanding patients' past and current health states to predict and, ultimately, change their future course[1–3]. Artificial intelligence (AI) methods promise to aid this task by learning patterns of disease progression from large corpora of health records[4,5]. However, their potential has not been fully investigated at scale. Here we modify the GPT[6] (generative pretrained transformer) architecture to model the progression and competing nature of human diseases. We train this model, Delphi-2M, on data from 0.4 million UK Biobank participants and validate it using external data from 1.9 million Danish individuals with no change in parameters. Delphi-2M predicts the rates of more than 1,000 diseases, conditional on each individual's past disease history, with accuracy comparable to that of existing single-disease models. Delphi-2M's generative nature also enables sampling of synthetic future health trajectories, providing meaningful estimates of potential disease burden for up to 20 years, and enabling the training of AI models that have never seen actual data. Explainable AI methods[7] provide insights into Delphi-2M's predictions, revealing clusters of co-morbidities within and across disease chapters and their time-dependent consequences on future health, but also highlight biases learnt from training data. In summary, transformer-based models appear to be well suited for predictive and generative health-related tasks, are applicable to population-scale datasets and provide insights into temporal dependencies between disease events, potentially improving the understanding of personalized health risks and informing precision medicine approaches.

The progression of human disease across age is characterized by periods of health, episodes of acute illness and also chronic debilitation, often manifesting as clusters of co-morbidity. Patterns of multimorbidity affect individuals unevenly and have been associated with lifestyle, heritable traits and socioeconomic status[1–3]. Understanding each individual's multi-morbidity risks is important to tailor healthcare decisions, motivate lifestyle changes or direct entrance into screening programs, as is the case for cancer[8,9]. Critically, health cannot only be understood by the presentation of individual diagnoses but, rather, in the context of an individual's co-morbidities and their evolution over time. While a wide range of prediction algorithms exist for specific diseases, from cardiovascular disease to cancer[10–12], few algorithms are capable of predicting the full spectrum of human disease, which recognizes more than 1,000 diagnoses at the top level of the International Classification of Diseases, Tenth Revision (ICD-10) coding system.

Learning and predicting patterns of disease progression is also important in populations that are ageing and that exhibit shifts in their underlying demographic's morbidities. For example, it has been predicted that, globally, the number of cancer diagnoses will increase 77% by 2050 (ref. 13) or that, in the UK, the number of working-age individuals with major illnesses, including depression, asthma, diabetes, cardiovascular disease, cancer or dementia, will increase from 3 to 3.7 million by 2040 (ref. 14). Modelling the expected burden of disease is therefore critical for healthcare and economic planning and, moreover, the continual tracking of disease occurrence along with its likely future prevalence within population groups promotes a more informed healthcare system.

Recent developments in AI may help to address some methodological limitations of multi-morbidity modelling, which have so far proved difficult to overcome[15]. Aside from the great number of diagnoses, these include challenges in modelling temporal dependencies among previous events, the integration of potentially diverse prognostically relevant data and the statistical calibration of predictions. Large language models (LLMs)[16–19]—a subfield of AI that enables chatbots such as ChatGPT[20,21]—model language as a sequence of word fragments (tokens). Generated token by token, the new text is based on all preceding text

[1]Division of AI in Oncology, German Cancer Research Centre DKFZ, Heidelberg, Germany. [2]European Molecular Biology Laboratory, European Bioinformatics Institute EMBL-EBI, Hinxton, UK. [3]Faculty of Biosciences, Heidelberg University, Heidelberg, Germany. [4]Novo Nordisk Foundation Center for Protein Research, Faculty of Health and Medical Sciences, University of Copenhagen, Copenhagen, Denmark. [5]Statistics Denmark, Copenhagen, Denmark. [6]Department of Biosystems Science and Engineering, ETH Zurich, Basel, Switzerland. [7]Department of Public Health, University of Copenhagen, Copenhagen, Denmark. [8]ROCKWOOL Foundation, Copenhagen, Denmark. [9]Faculty of Mathematics and Computer Science, Heidelberg University, Heidelberg, Germany. [10]Robert Bosch Center for Tumor Diseases, Stuttgart, Germany. [11]Medical Faculty, Eberhard-Karls-University, Tübingen, Germany. [12]University Hospital Tübingen, Tübingen, Germany. [13]These authors contributed equally: Artem Shmatko, Alexander Wolfgang Jung, Kumar Gaurav. ✉e-mail: birney@ebi.ac.uk; tomas@ebi.ac.uk; moritz.gerstung@dkfz.de

and, with enough training, the statistical dependencies among these tokens prove sufficient to produce context-aware and even conversational text, which is often indistinguishable from that of a human counterpart.

The analogy between LLMs and disease progression modelling, which also entails recognizing past events and exploiting their mutual dependencies to predict the future sequence of morbidity, has recently inspired a series of new AI models. For example, BERT-based models[22–25] have been developed for specific prediction tasks. Transformer models trained on electronic health records have been used for predicting diagnoses such as pancreatic cancer[26], self-harm[25] and stroke[24], as well as non-clinical parameters such as self-esteem[27]. However, despite promising proofs of concept[4,28,29], the potential for comprehensive and generative multi-morbidity modelling has not yet been fully assessed.

Here we demonstrate that attention-based transformer models, similar to LLMs, can be extended to learn lifetime health trajectories and accurately predict future disease rates for more than 1,000 diseases simultaneously on the basis of previous health diagnoses, lifestyle factors and further informative data. Our extended model, termed Delphi-2M, was trained on data from the UK Biobank, a population-scale research cohort, and validated on Danish population registries. The vocabulary of the model includes ICD-10 top-level diagnostic codes, as well as sex, body mass, smoking, alcohol consumption and death. Delphi provides individual-level predictions of multi-disease incidences and models future health trajectories at any point throughout an individual's life course. Moreover, the internal model of Delphi offers insights into how past data influence the rates of subsequent diseases. We further assess biases and fairness across demographic subgroups and discuss Delphi's potential as a framework for healthcare modelling.

## A transformer model for health records

A person's health trajectory can be represented by a sequence of diagnoses using top-level ICD-10 codes recorded at the age of first diagnosis as well as death. Furthermore, 'no event' padding tokens were randomly added at an average rate of 1 per 5 years to eliminate long intervals without other inputs, which are especially frequent for younger ages and during which the baseline disease risk can change substantially (Extended Data Fig. 1). Together, these data comprise 1,258 distinct states—tokens in LLM terminology. Additional information includes sex, body mass index (BMI) and indicators of smoking and alcohol consumption, which are used as input information but not predicted by the model (Fig. 1a).

Training data comprised 402,799 (80%) participants of the UK Biobank recorded before the 1 July 2020. Data for the remaining 100,639 (20%) participants were used for validation and hyperparameter optimization, while all records for 471,057 (94%) participants still alive on 1 July 2020 were used for longitudinal testing up until 1 July 2022 (Fig. 1b). Additional external testing was conducted on the Danish disease registry data, which covered 1.93 million Danish nationals and spanned the period from 1978 to 2018.

To model disease history data, which, in contrast to text, occurs on a continuous time axis, we extended the GPT-2 architecture[6] (Fig. 1c). Transformer models map their inputs into an embedding space, where information is successively aggregated to enable autoregressive predictions. The first change therefore replaces GPT's positional encoding, a mapping that identifies each text token's discrete position, with an encoding of continuous age using sine and cosine basis functions[16]. Standard GPT models only predict the next token using a multinomial probability model. Thus, the second extension is the addition of another output head to also predict the time to the next token using an exponential waiting time model (Methods). Third, GPT's causal attention masks, which ensure that the model accesses only information from past events, are amended to additionally mask tokens recorded at the same time. Padding, lifestyle and sex tokens use a similar encoding but

do not enter the likelihoods, as the model is deliberately not trained to predict them.

We term this model Delphi (Delphi large predictive health inference). This architecture enables one to provide the model with a partial health trajectory (prompt in LLM terminology) to calculate the subsequent rate (per day) for each of the 1,256 disease tokens plus death. Furthermore, the next token and the time to this event can be sampled on the basis of these rates. Iteratively, this procedure samples entire health trajectories (Fig. 1d).

A systematic screen of architecture hyperparameters (embedding dimensionality, number of layers, heads) confirms the reported empirical scaling laws[30], which state that model performance increases with the number of datapoints and, up to a limit defined by the available data, as the number of parameters increases (Fig. 1e). The screen indicates that, for the UK Biobank dataset, optimal Delphi models have around 2 million parameters. One of the models within the optimal range has an internal embedding dimensionality of 120, 12 layers and 12 heads, amounting to a total of 2.2 million parameters. Results based on this model parameterization are discussed throughout the rest of the paper. We note that qualitatively similar results are obtained from other parameter choices (Extended Data Fig. 2 and Supplementary Fig. 1).

An ablation analysis shows how Delphi-2M architectural modifications contribute to a better age- and sex-stratified cross-entropy compared with a standard GPT model (Fig. 1f, Supplementary Table 1 and Supplementary Fig. 2). A good, albeit slightly inferior, classification performance at different ages may already be achieved by adding regular 'no event' padding tokens to the input data with GPT models alone. However, a key distinguishing feature of Delphi compared with basic GPT models is its ability to calculate the absolute rates of tokens, which provide consistent estimates of inter-event times (Fig. 1g). This property also implies that the rates may be interpreted as the incidences of tokens.

## Modelling multi-disease incidences

Delphi-2M's accuracy in predicting diverse disease outcomes in the validation cohort is compared to the sex and age-stratified incidence as an epidemiological baseline. As can be seen in the ten examples shown in Fig. 2a, the incidence curves are very varied, with some diseases, such as chickenpox, peaking in infancy, while others, such as asthma or depression, are relatively flat and with most rising exponentially in old age. Moreover, there are noticeable differences between the sexes, which are obvious for breast cancer but also pronounced for diabetes, depression, acute myocardial infarction and death. Delphi-2M's predictions are updated for each individual when new inputs are recorded. The predictions largely follow the sex- and age-stratified incidence curves but also indicate events or periods when the individual risk remains below or rises above the population average. For some diseases, such as asthma or arthrosis, the spread is narrow, indicating a limited ability to predict beyond the sex- and age-incidence trend. Yet for other diseases, including septicaemia, and also death, the spread is wide, indicating predictable inter-individual differences in disease rates.

Delphi's ability to predict the next diagnosis token across the spectrum of human disease is confirmed by the average age-stratified area under the receiver operating characteristic curve (AUC), which averages at values of approximately 0.76 in the internal validation data (Fig. 2b and Supplementary Table 2). For 97% of diagnoses, the AUC was greater than 0.5, indicating that the vast majority followed patterns with at least partial predictability. These patterns were found to be true across the different chapters of the ICD-10 spectrum, which define broad groups of disease for both sexes (Fig. 2c,d). Among the most confidently predicted next events is death, with an age-stratified AUC of 0.97 in both sexes. Importantly, calibration analyses in 5-year age brackets show that the predicted rates closely match the observed number of cases, showing that the models'

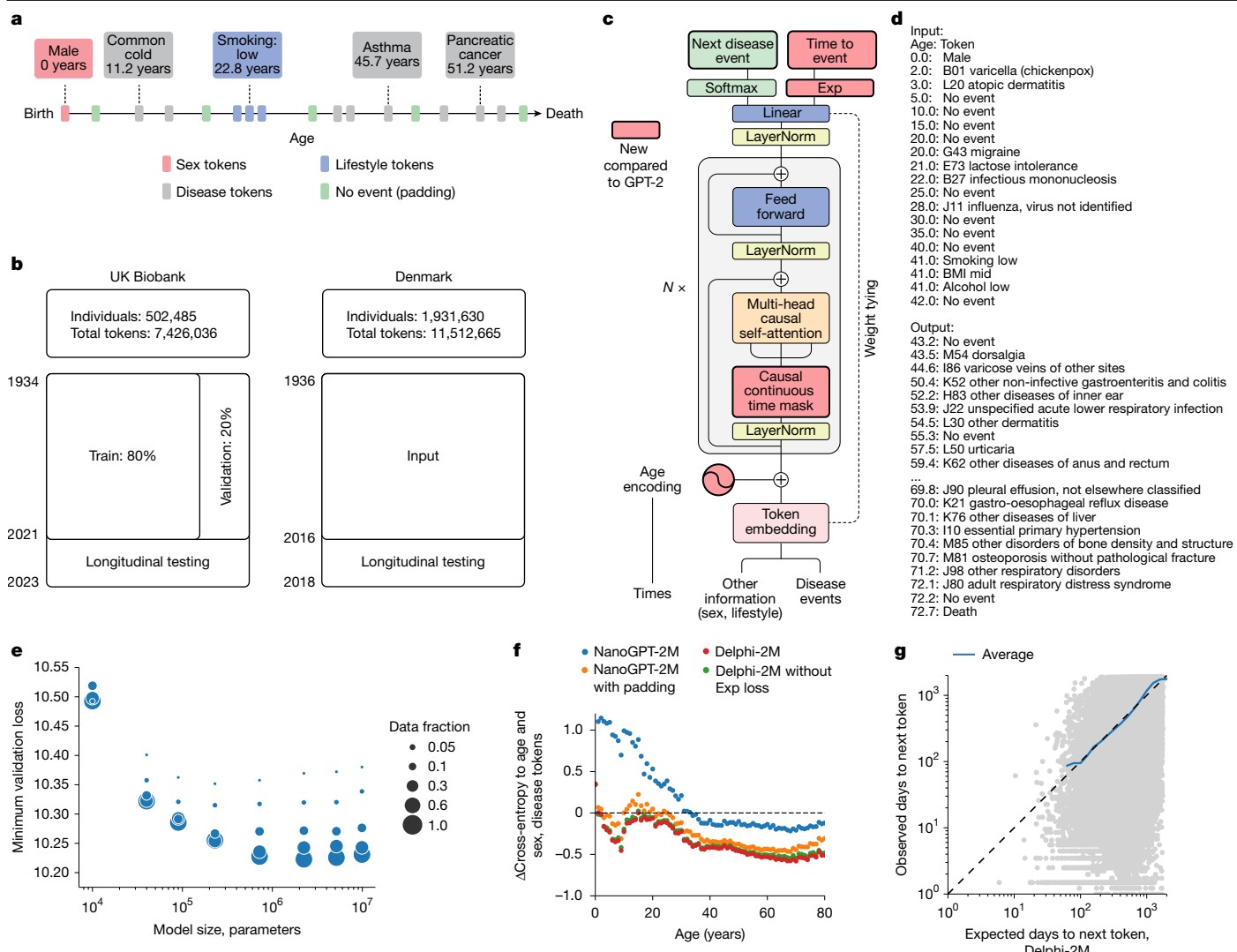

**Fig. 1 | Delphi, a modified GPT architecture, models health trajectories.** **a**, Schematic of health trajectories based on ICD-10 diagnoses, lifestyle and healthy padding tokens, each recorded at a distinct age. **b**, Training, validation and testing data derived from the UK Biobank (left) and Danish disease registries (right). **c**, The Delphi model architecture. The red elements indicate changes compared with the underlying GPT-2 model. 'N ×' denotes applying the transformer block sequentially N times. **d**, Example model input (prompt) and output (samples) comprising (age:token) pairs. **e**, Scaling laws of Delphi, showing the optimal validation loss as a function of model parameters for different training data sizes. **f**, Ablation results measured by the cross-entropy differences relative to an age- and sex-based baseline (y axis) for different ages (x axis). **g**, The accuracy of predicted time to event. The observed (y axis) and expected (x axis) time to events are shown for each next token prediction (grey dots). The blue line shows the average across consecutive bins of the x axis.

rates of the next tokens are consistently estimated (Extended Data Fig. 3).

Next-event predictions are often the consequence of acute illness or diagnostic refinements that accrue over the course of a few weeks or months, which may be undesirable for prognostication. Delphi-2M's average AUC values decrease from an average of 0.76 to 0.70 after 10 years, indicating that its predictions are also relevant for long-term prognostication (Fig. 2e and Supplementary Fig. 3). Similar results were observed in longitudinal test data, which also show no substantial shift in diagnostic patterns throughout the Biobank's follow-up (Supplementary Fig. 4).

The performance of Delphi was similar to routinely used clinical risk scores for cardiovascular disease and dementia, and better than those used for death. For diabetes, the performance of Delphi was worse compared with the use of a single marker, HbA1c, which is used clinically for risk prediction and diagnosis of diabetes (Fig. 2f, Supplementary Fig. 4c and Supplementary Table 3). This was the case for next-event predictions, as well as prediction horizons up to

24 months. Delphi-2M's AUC values were also generally higher than those of a recent machine learning algorithm that calculates the risks of a similarly broad spectrum of ICD-10 diagnoses using 67 different biomarkers available through the UK Biobank[31], even though for many diagnoses, such as diabetes, biomarkers remain indispensable (Fig. 2e and Extended Data Fig. 4), marking potential for future modifications of Delphi that additionally use data beyond health records (Extended Data Fig. 5). For most cases, Delphi-2M's multi-disease predictions match or exceed current risk models for individual disease outcomes and offer the great advantage of enabling the simultaneous assessment of more than 1,000 diseases and their timing at any given time, while also surpassing multi-disease models in quality.

## Sampling future disease trajectories

One of the most promising features of generative models is the ability to sample disease trajectories, conditional on data recorded up to a

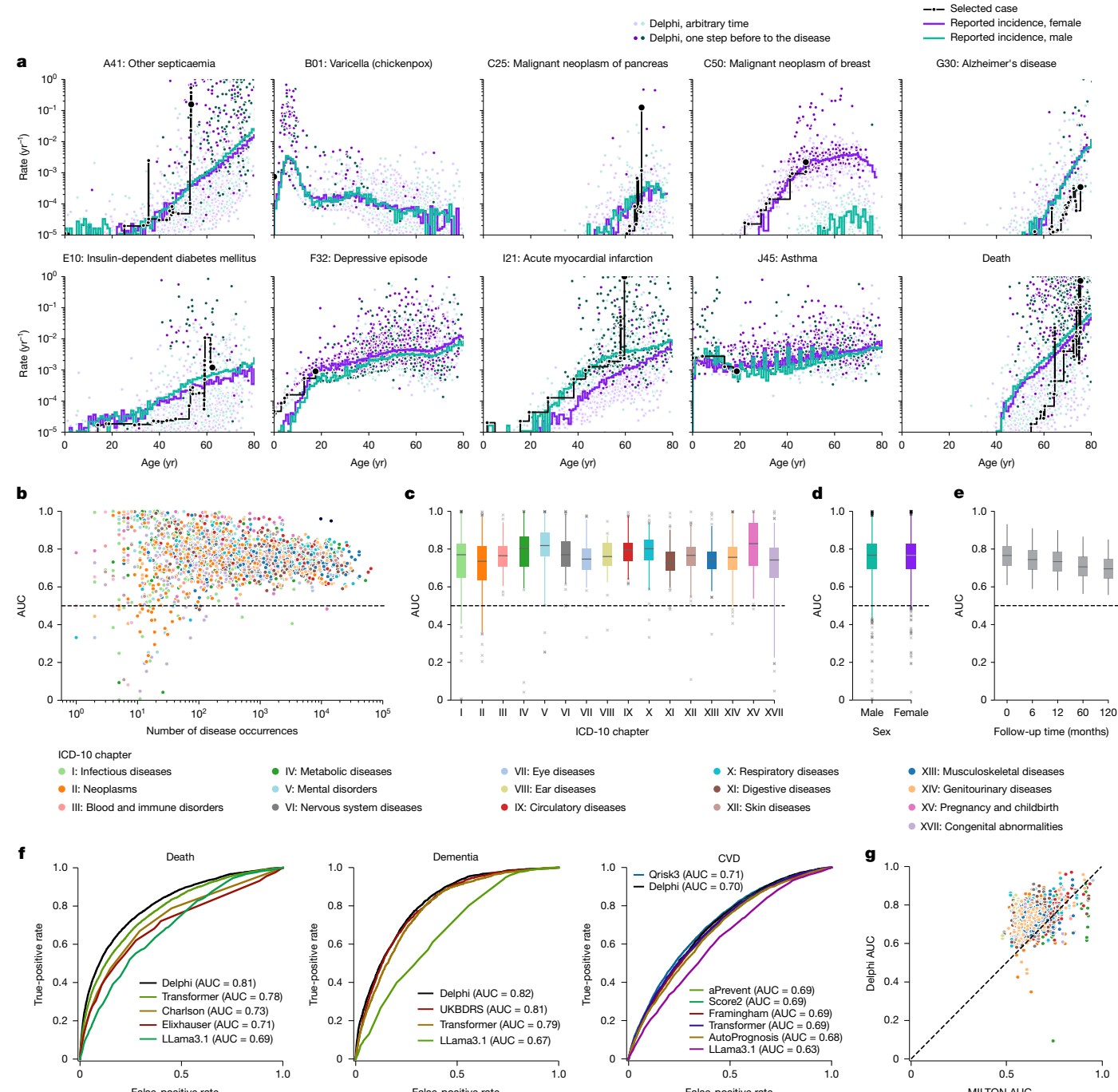

**Fig. 2 | Delphi-2M accurately models the rates of a wide range of diseases.**
**a**, The predicted rates for nine exemplary diagnoses and death (*y* axis) as a function of age (*x* axis). The points show predictions at each recorded input token. Colours separate biological sex; the darker colours indicate predictions immediately before the diagnosis in question. The purple and turquoise lines are disease rates observed for each yearly age bin in the training data. The solid black line connects consecutive predictions for one randomly selected case throughout age. **b**, Average age–sex-stratified AUC values (*y* axis) as a function of training occurrences (*x* axis). Shown are data for *n* = 906 diagnoses for male individuals and *n* = 957 diagnoses for female individuals for which a sufficient

number of events was recorded in the validation data to evaluate AUC values. **c**, The same as **b**, but aggregated by the ICD-10 chapter. **d**, The same as **b**, aggregated by sex. **e**, AUC values of all diagnoses in **b** for different time gaps between prediction and diagnoses (*x* axis). **f**, ROC curves for Delphi and other clinical or machine learning methods for three selected end points evaluated on the internal longitudinal testing set. **g**, AUC values of MILTON[31], a biomarker-based machine learning model (*x* axis), in prognostic mode, compared with Delphi-2M AUC values from the UK Biobank validation set (*y* axis) for *n* = 410 diagnoses. The box plots in **c**–**e** show the median (centre line), the first to the third quartile (box limits) and the 0.025 and 0.975 quantiles (whiskers).

certain point. This is a property that few conventional epidemiological models possess.

To systematically assess the influence of medical histories on future health, we sampled health trajectories for each participant from the UK Biobank validation cohort on the basis of data available until the age

of 60 years (Fig. 3a). This provides the opportunity to compare 63,662 sampled and observed trajectories. When evaluated at the population level, the disease incidences at ages 70–75 years are well recapitulated, showing that the overall distributions are well preserved by iterative sampling (Fig. 3b). This is further confirmed by the cross-entropy loss

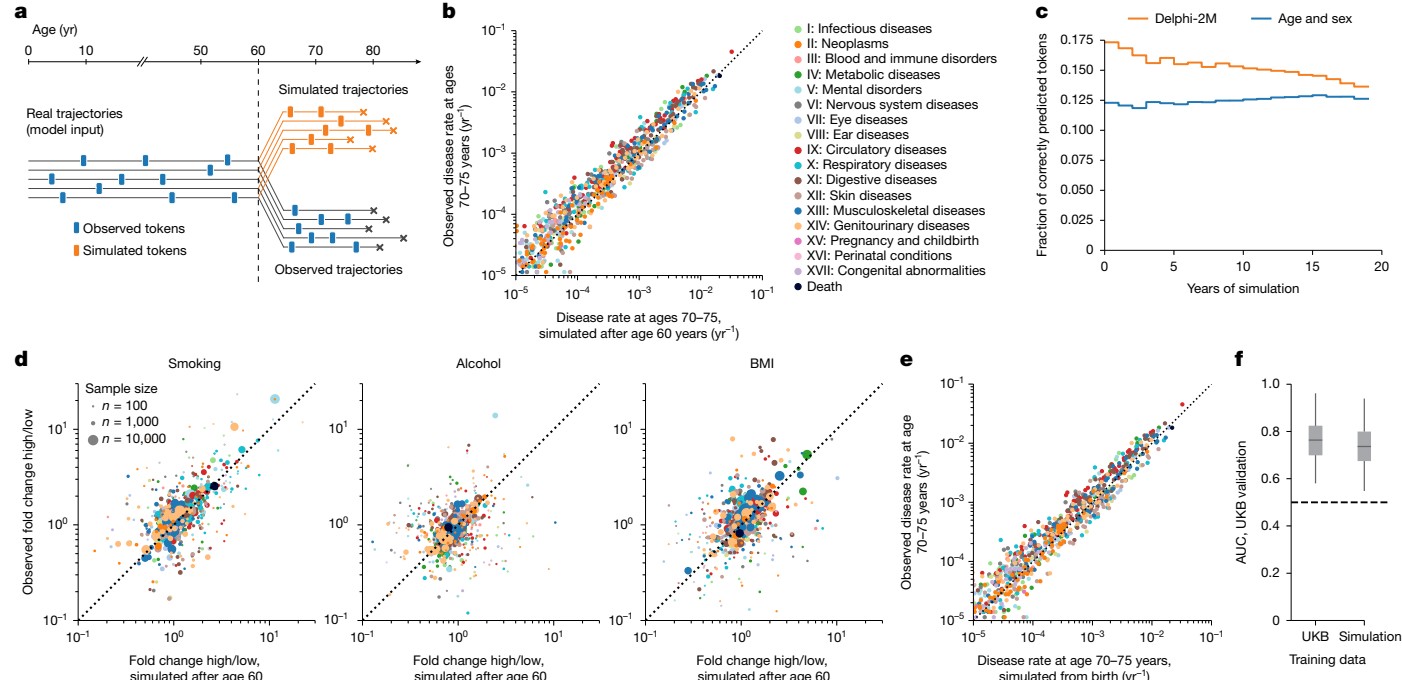

**Fig. 3 | Generative modelling with Delphi-2M informs future outcomes.**
**a**, Schematic of the experiment design. Delphi-2M is used to simulate health trajectories using validation data (*n* = 63,622 individuals with disease records both before and after 60 years of age) observed until the age of 60. A single trajectory is simulated per individual. Subsequently, simulated trajectories are compared to the observed outcomes for the same person. **b**, Delphi-2M-modelled disease rates at ages 70–75 years (*x* axis) compared with observed rates at the same ages (*y* axis). **c**, The fraction of correctly predicted diagnoses (*y* axis) per 1-year age bin as a function of the years after simulation started at age 60 years (*x* axis). Delphi-2M, orange. The blue curve uses age and sex as a prediction baseline. **d**, Simulated (*x* axis) and observed (*y* axis) fold changes of disease rates for high versus low smoking, alcohol consumption and BMI groups. The evaluation period included ages 70–75 years and used simulations from the age of 60 years. **e**, The same as **b**, evaluated for simulations from birth. **f**, The AUC values of disease risk prediction (*n* = 1,334 disease–sex pairs) for Delphi when trained on UKB and Delphi-2M-sampled synthetic data (Methods). The box plots show the median (centre line), the first to the third quartile (box limits) and the 0.025 and 0.975 quantiles (whiskers).

of sampled trajectories, which is, on average, indistinguishable from the observed data but drops when the preceding disease histories are shuffled between participants (Supplementary Fig. 5b).

In the first year of sampling, there are on average 17% disease tokens that are correctly predicted, and this drops to less than 14% 20 years later. These figures compare to values of 12–13% of correctly predicted disease tokens using sex and age alone, confirming that the conditional generation helps to make more accurate predictions of future events (Fig. 3c and Supplementary Fig. 5c,d).

Delphi-2M's ability to simulate differential health outcomes over a decade or more, on the basis of each individual's health history, manifests in a multitude of ways. For example, the changes in disease burden in different population subsets defined by smoking, alcohol consumption or BMI are well predicted (Fig. 3d and Extended Data Fig. 6a). Similar findings are observed when the population is stratified by the presence of previous diseases or by estimated disease risks (Extended Data Fig. 6b,c). Together, these analyses show that Delphi-2M's conditional samples provide meaningful extrapolations for future health courses, which reflect the influence of past health events.

The use of synthetic data has been proposed to help overcome issues with privacy in biomedical modelling if such datasets do not reveal characteristics specific to any one person. Fully synthetic data, which are sampled from birth with randomly assigned sex, reproduce the observed age and sex-specific incidence patterns throughout life (Fig. 3e). Further assessment shows that the generated trajectories do not exhibit any greater similarity to the training data than those from the validation cohort (Supplementary Fig. 6). While partially overlapping disease trajectories may be found in terms of absolute disease tokens, the extent of overlap appears as expected on the basis of the observed incidences and co-morbidity patterns.

To illustrate the use of synthetic data, we trained a version of Delphi-2M exclusively on synthetic data. Notably, when evaluated on the observed validation data, the fully synthetically trained model achieves an age–sex-stratified average AUC of 0.74, which is only three percentage points lower than that of the original Delphi-2M model (Fig. 3f). This confirms that synthetic data preserve much of the information relevant to training Delphi models and may serve as a less privacy-sensitive alternative to personal data.

## Explaining Delphi-2M predictions

Insights into how Delphi-2M uses past information to predict future disease rates can be obtained by assessing the structure of the disease embeddings. GPT models linearly map inputs into a lower-dimensional embedding space, in which the temporal sequence of events is iteratively aggregated to produce a state from which predictions of each next token are derived (Fig. 1c). Delphi-2M's specific implementation uses weight tying, which uses the same mapping to project the final embedding state to the token risks, guiding interpretation as the embedding matrix reflects the observed structure of co-morbidity risks.

As shown by the uniform manifold approximation and projection (UMAP) representation of Delphi-2M's embedding matrix in Fig. 4a, disease codes cluster closely by the underlying chapter, a property that the model has no direct knowledge of, and that purely reflects co-occurrence patterns in the data. Yet there are also noticeable exceptions, for example, cancers and precancers of the female reproductive tract. Another noteworthy cluster involves the two types of diabetes, retinal disorders and the neuropathies caused by them. Diseases with high acute mortality, such as myocardial infarction or septicaemia, are clustering with death.

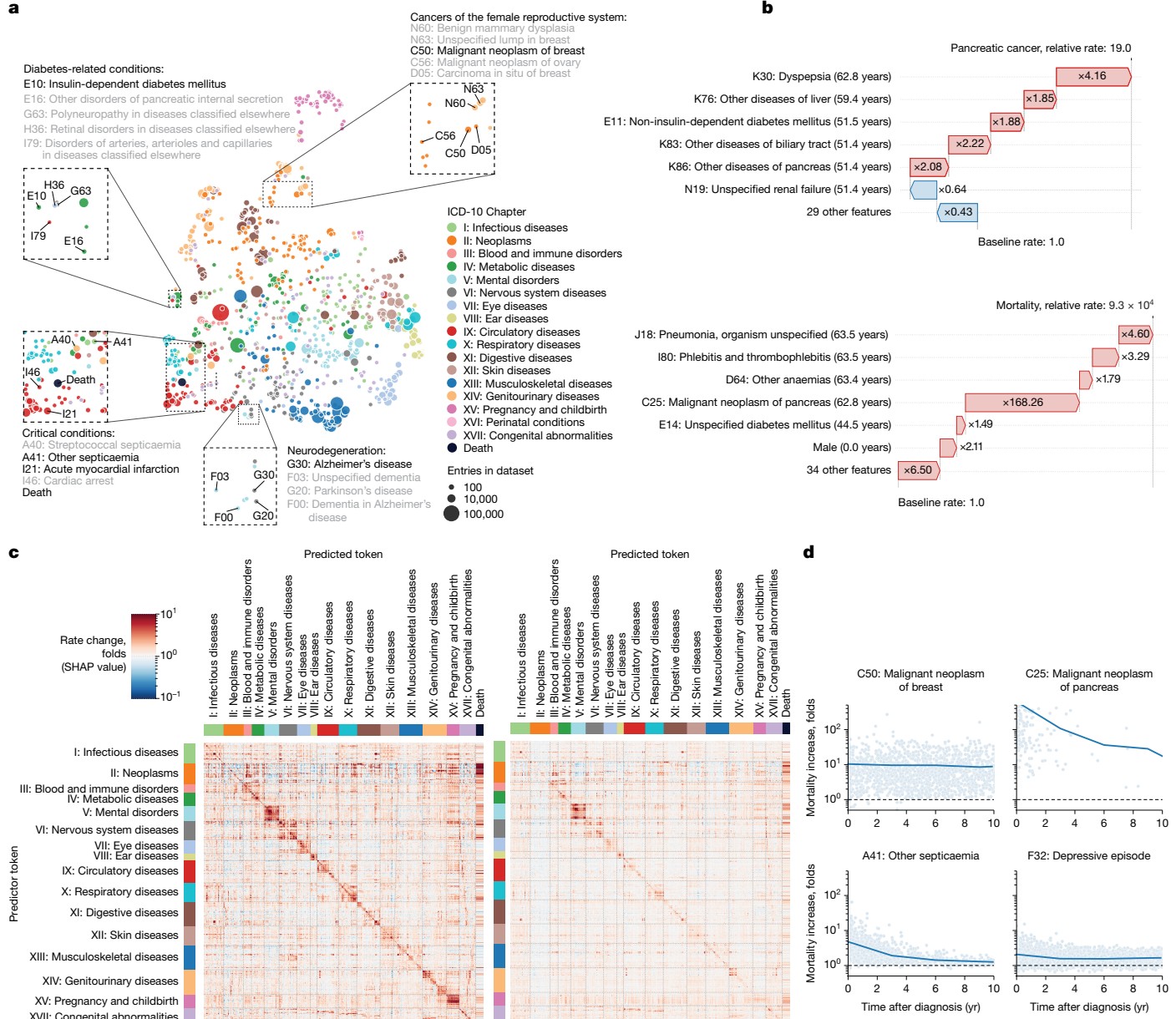

**Fig. 4 | Explainable AI offers insights into disease progression. a**, UMAP projection of token embeddings. Selected diseases are shown in the magnified areas. Colours define disease chapters. **b**, SHAP-explained token risk contributions for individual trajectories. Top, the risk of pancreatic cancer immediately before diagnosis at age 68.2 years, which was found to be 19× increased. Bottom, the SHAP estimates of contributions to estimated mortality at age 63.5 years, which was greatly increased, in large part due to the preceding diagnosis of pancreatic cancer. **c**, The average SHAP effect of each of $n = 778$ disease tokens with more than 5 occurrences and grouped by chapter ($y$ axis) on the same set of tokens plus death ($x$ axis). The red colours indicate a risk increase, whereas blue indicates a decrease. **d**, Rate change (SHAP value) of mortality ($y$ axis) as a function of time after diagnosis ($x$ axis) for selected diseases.

To gain further insights into how individual tokens influence future risks, shapley additive explanations (SHAP) values measure the influence of each token from the input trajectory on model predictions by a systematic assessment of subsampled data for individual prediction. As illustrated with the example of the proband's partial trajectory shown in Fig. 4b, this analysis reveals that a series of disease diagnoses of the digestive tract (ICD-10 chapter XI) elevated their pancreatic cancer risk 19-fold. The subsequent pancreatic cancer diagnosis in turn increased the rate of mortality almost ten thousandfold.

SHAP analysis of data from 100,639 individuals of the validation cohort reveals the mutual dependencies by which each disease, sex and lifestyle token influences the rate of subsequent disease tokens, similar to hazard ratios in conventional statistical models (Fig. 4c (left)

and Extended Data Fig. 7). Effects mostly increase the rates of other diseases and are usually found among diseases of the same ICD-10 chapters, underscoring that the recorded patterns of co-morbidities often cluster within specific ICD-10 disease chapters. Particular clusters spanning entire disease chapters are visible for ICD-10 chapters V (mental disorders) and XV (pregnancy and childbirth). Notably, such patterns often appeared symmetrical, indicating similar predicted effect sizes of one disease token influencing another and vice versa (Supplementary Fig. 7). This behaviour can be attributed to the structure of the embedding space, which places temporarily co-occurring diagnoses in local proximity (Extended Data Fig. 8).

The patterns of modelled influences after 10 years are similar to short-term effects, even though the strength of associations is greatly

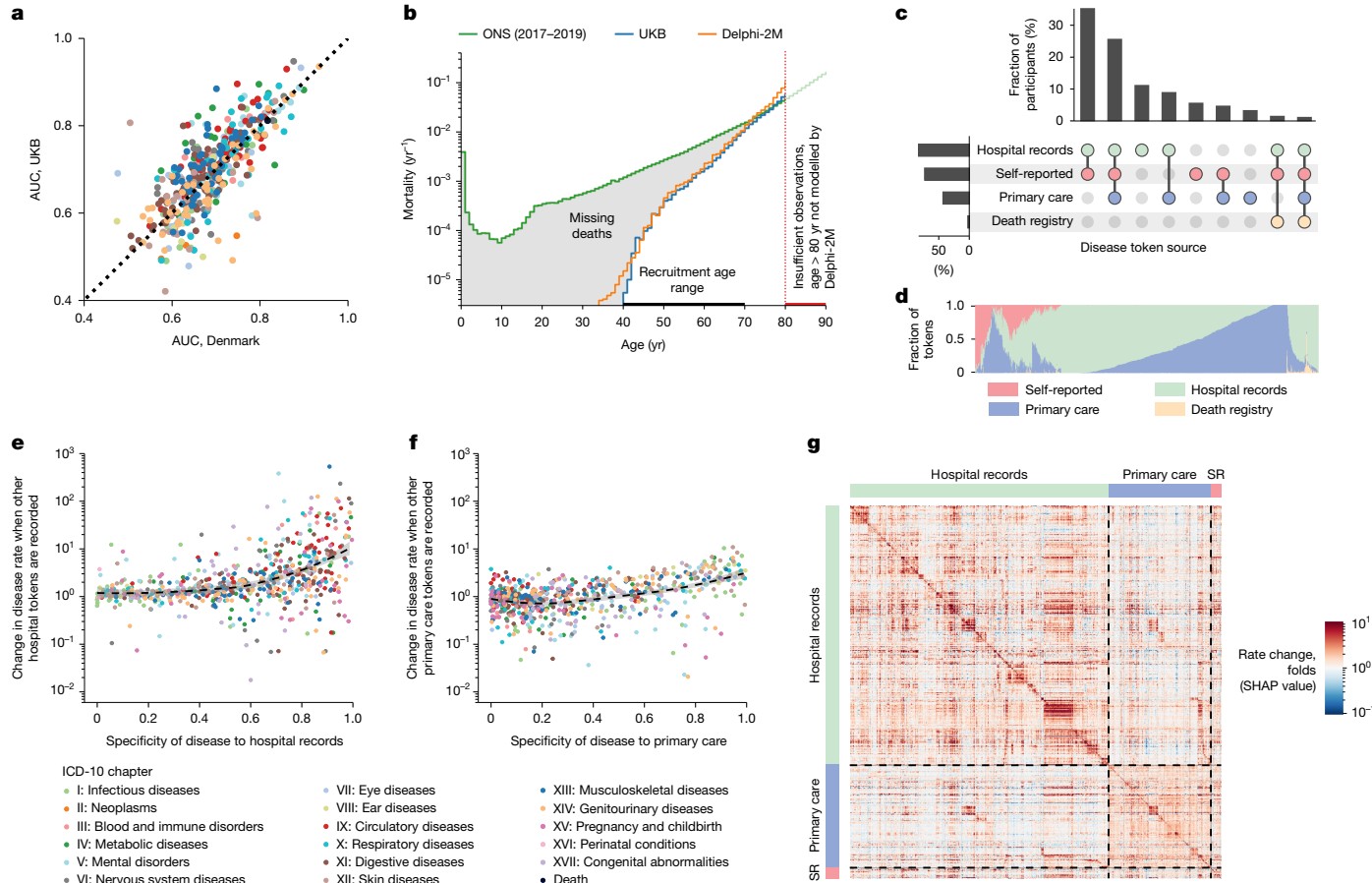

**Fig. 5 | Epidemiological biases in UK Biobank data reflected by Delphi-2M.**
**a**, Comparison between AUC values in the UKB longitudinal testing and the external testing using Danish data. **b**, Yearly mortality estimates by Delphi-2M (UK validation cohort), observed rates in the UK Biobank and Office for National Statistics national estimates across the entire British population. As only living individuals between 40 and 70 years of age (black line) were recruited to the UK Biobank, many deaths are missing compared with the Office for National Statistics population estimate (grey shaded area). **c**, UpSet plot of disease data availability in the UK Biobank validation cohort (n = 100,639) (top). Bottom, the data source distribution for records per disease token (token position sorted). **d**, Data source distribution for records per disease token (token position sorted). **e**, Hospital-record missingness bias (relative rate after first hospital token; y axis) as a function of token exclusivity to hospital records (x axis) for each relevant

diagnosis (points). Points are coloured by the ICD-10 chapter, the overall trend is shown in black using a nonparametric (loess) curve with 95% confidence intervals shown in grey (UK validation cohort). **f**, Primary care missingness bias (y axis) as a function of primary care token exclusivity, coloured by the ICD-10 chapter. The trend is shown in black using a nonparametric (loess) curve with 95% confidence intervals shown in grey (UK validation cohort). **g**, SHAP value matrix, similar to Fig. 4c. The columns and rows correspond to different diseases and are sorted by the dominating source, then ICD-10 chapter. A dominating source is defined as the origin of more than 65% of records for a given disease; diseases without a dominating source are not shown. SHAP values indicate the greater influence of diseases on other diseases from the same group. SR, self-reported.

attenuated (Fig. 4c (right)). The cluster relating to diseases of ICD-10 chapter XV (pregnancy and childbirth) within 5 years is entirely absent after 10 years, which is expected given that pregnancy-associated diseases occur within a finite period. However, dependencies among mental disorders remain apparent, similar to the effects of neoplasms on mortality. These observations are noteworthy as the quantification of temporal dependencies on past events poses a particular challenge for conventional epidemiological models, whereas Delphi-2M's GPT model uses attention-based weights, which are updated with every new input, including the 'no event' paddings.

To further illustrate Delphi-2M's capabilities of modelling temporal dependencies, we note that, for some diseases, such as cancers, the influence on mortality decays with a half-life of several years, reflecting the sustained risks of recurrence or impacts of treatment (Fig. 4d). However, for septicaemia, the influence on mortality is much more short-lived and drops sharply, effectively recovering to values close to the population average. This inference agrees with traditional Nelson–Aalen analyses of the hazard rates (Supplementary Fig. 8f). This behaviour is also reflected by Delphi-2M's attention maps, which

show that cancer tokens are attended to for long periods, while those of septicaemia, myocardial infarction and many other diseases tend to be short lived (Supplementary Fig. 8).

## External validation and bias assessment

To assess whether Delphi-2M's inference generalizes to unseen cohorts, we performed external testing using Danish population registry data. For this purpose, we transferred Delphi-2M with the weights learned from the UK Biobank training and evaluated predictions on the Danish data; no retraining or adjustments have been made. The average AUC when Delphi-2M is applied to Danish data was 0.67 (s.d. 0.09), which is lower than for longitudinal testing on UKB data (0.69, s.d. 0.09). Predictions for different diseases were highly correlated across the datasets (Pearson correlation coefficient 0.76, 95% confidence interval (CI): 0.72–0.80) (Fig. 5a and Supplementary Fig. 9). The fact that Delphi-2M can be applied to Danish population data with slightly reduced accuracy indicates that many patterns learned by the model accurately reflect the true evolution of multi-morbidity,

while also highlighting the existence of differences within each cohort.

A further question is to what extent a model trained on the UK Biobank, which is an epidemiological cohort, generalizes to the general population. The UK Biobank comprises more white British citizens than the general population, and the participants tend to be on average more affluent and educated[32]. Lower rates of diseases are reported among participants of white ethnicity, and the rate of diagnoses increases with deprivation—trends that are reproduced by Delphi-2M (Extended Data Fig. 9). Further assessments of Delphi-2M's performance in demographic subgroups are provided in the Supplementary Discussion, Supplementary Figs. 10 and 11 and Supplementary Table 4.

In the UK Biobank, most individuals have been recruited between the ages of 40 and 70. This creates a selection bias, as no deaths are recorded before recruitment, which has direct implications for the estimated mortality (Fig. 5b). This immortality bias also indirectly affects the incidence of diseases associated with high mortality, such as cancers, as only survivors are included in the UK Biobank. For time-dependent analyses, the jump of mortality to non-zero values at recruitment can also lead to false attribution of the apparent increase to unrelated variables recorded at the time of recruitment. Furthermore, limited follow-up data are currently available for individuals 80 years of age and older. This period is therefore not reliably modelled by Delphi-2M.

UK Biobank's disease data have been collated from self-reports, primary care, hospital admissions, cancer and death registries, each of which contributes characteristic disease tokens: self-reporting and GP records contain mostly common diseases, while data from hospital records include more-aggressive disease tokens, such as myocardial infarction or septicaemia (Fig. 5d). However, the underlying sources were not always available for each participant and time period (Fig. 5c and Supplementary Fig. 12). Missingness of a particular data source therefore causes the absence of multiple diagnoses. Such patterns, which reflect only the data-collection process, are also learned by Delphi-2M. The predicted rates of diseases exclusive to hospital records are, on average, ten times higher in individuals with a disease history that includes other hospital records (Fig. 5e,f). Septicaemia, for example, is diagnosed in 93% of cases in a hospital setting and is predicted to occur at 8× greater rates in individuals with any other hospital data. These source effects also explain some of the substructures visible in the UMAP representation of disease embeddings (Fig. 4a and Extended Data Fig. 10a–d) and also in the matrix of SHAP effects (Figs. 4c and 5g and Extended Data Fig. 10). While some of these associations may reflect true diagnostic pathways or disease clusters diagnosed in a distinct care setting, it nevertheless appears that some of these associations are artefacts stemming from the incomplete aggregated nature of the UK Biobank's data.

## Discussion

Here we present Delphi-2M—a GPT-based model of multi-disease progression. Delphi-2M extends the GPT large language model to account for the temporal nature of health trajectories. Analogous to LLMs, which learn the grammar and contextual logic of language from large bodies of text, Delphi-2M inferred the patterns of multi-disease progression when trained on data for more than 1,000 diseases and baseline health information recorded in 402,799 UK Biobank participants.

A detailed assessment of Delphi-2M's predictions showed that they consistently recapitulate the patterns of disease occurrence at the population scale as recorded in the UK Biobank. For the majority of diseases, Delphi-2M's multi-disease, continuous-time model predicted future rates at comparable or better accuracy than established single-disease risk models, alternative machine learning frameworks and blood-biomarker-based models. Only a small performance drop was observed when applied to data from Danish disease registries,

demonstrating that models are—even without additional finetuning—largely applicable across national healthcare systems.

Delphi-2M is uniquely capable of sampling future disease trajectories, which enables the estimation of cumulative disease burdens over periods of up to 20 years, conditional on previous health information. Note that Delphi's predictions are generally strongly influenced by statistical chance and compatible with a range of outcomes for a given individual. The ability to generate synthetic data may also help create datasets that preserve the statistical co-occurrence patterns without revealing any specific data, which could facilitate the development of further AI models with a decreased risk of revealing personal information.

Delphi-2M offers insights into the modes of disease progression. The ability to cluster disease risks may be useful for genomic association studies that focus on comorbidities or are stratified by the risks derived from health trajectories. Delphi-2M's ability to quantify the temporal influence of previous health data revealed that cancers increase mortality in a sustained manner, while the effects of myocardial infarction or septicaemia regress within 5 years. Similar analyses also revealed clusters of persisting comorbidities, such as mental health conditions. Although Delphi-2M appears to be capable of modelling temporally directed dependencies, we caution against interpreting these as causal relationships that could be exploited to modify future health courses.

There are also several limitations that need to be considered. A detailed analysis of UK Biobank's first occurrence data revealed several biases, reflected by Delphi-2M. In addition to a healthy volunteer bias and participant selection bias before recruitment, the diverse nature of health data sources impacted Delphi's prediction, as UK-Biobank-specific patterns of missingness were exploited to infer disease rates. Furthermore, Delphi-2M predicts different disease rates in subgroups based on ancestry background and deprivation indices, but no observable trend between lifestyle measures and birth year. These findings underscore the need for caution when using AI models for inference and prediction in heterogeneous healthcare datasets, potentially marking them as useful additions to currently used diagnostic pipelines, rather than replacements.

A promising feature of Delphi-2M's implementation is the relative simplicity of incorporating additional data layers, rendered possible by the transformer-based architecture. Immediate refinements of Delphi-2M may incorporate additional lifestyle data, self-reported health status, prescription records and blood tests, all of which are usually available in a general healthcare setting. Further multimodal extensions could include genomic data, richer metabolomic information, diagnostic imaging data or data from wearables that can be added to Delphi-2M's embedding layer, similar to how lifestyle tokens are currently incorporated (Extended Data Fig. 9). Furthermore, while ICD-10 provides a predefined tokenization of diseases, LLMs have been shown to also conceptualize natural language, making it plausible to expect that future models may derive similar meaning directly from free text records, enabling the application of Delphi-like models to unstructured data. Lastly, Delphi-2M itself could serve as an extension to LLMs. Similarly to systems that provide LLMs with query-relevant web search results to reduce hallucinations[33], a future healthcare-oriented LLM could invoke a Delphi-based model to improve the numerical accuracy of the generated replies[34].

An evident application of Delphi-type models is to support medical decision-making by rationally integrating information from various data modalities, which can be a challenge for healthcare professionals. Potential use cases could include identifying individuals who would benefit most from diagnostic tests or finding individuals with disease risk high enough to include them in screening programs, even if they have not yet met conventional age-based criteria. However, deploying clinical decision support systems requires a regulatory framework, which is still in its infancy for AI in healthcare.

An alternative use of Delphi models would be to inform healthcare providers, insurers and policymakers. Such applications, which inform the provision of healthcare rather than treatment decisions, require less regulation. Delphi's modelling has a resolution of an individual; when such predictions are aggregated, Delphi-type models may provide substantial system-wide modelling benefits, particularly in projecting the expected disease burden at local, regional and national levels, or for specific demographic groups of interest. This could reveal communities with unmet future healthcare needs over the next 1–2 decades and provide an opportunity to adjust the provision of healthcare. Such capabilities appear especially valuable in ageing populations in which healthcare needs are becoming increasingly complex and resource intensive.

These considerations illustrate the wide range of applications of generative models for biomedical research and, ultimately, also for healthcare. With appropriate training and evaluation, future multimodal model extensions may be used for preventive medicine, clinical decision support and healthcare planning. Our model and analyses present a further step towards unlocking the considerable healthcare benefits of the era of AI.

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

## Methods

### Data

**UK Biobank. Cohort.** The UK Biobank is a cohort-based prospective study comprising approximately 500,000 individuals from various demographic backgrounds recruited across the UK between 2006 and 2010. At the time of recruitment, individuals were between 37 and 73 years of age[35].

**Disease first occurrence data.** The main data source for health-related outcomes is built on the first occurrence data assembled in category 1712 from the UK Biobank. These data include ICD-10 level 3 codes (for example, E11: type 2 diabetes mellitus) for diseases in chapters I–XVII, excluding chapter II (neoplasms), plus death. The data are pre-assembled by UK Biobank and include the first reported occurrence of a disease in the linked primary care data (cat. 3000), inpatient hospital admissions (cat. 2000), death registry (fields 40001 and 40002) or self-reported data through questionnaires (field 20002).

Information on neoplasms was not included in category 1712 by the UK Biobank, and we therefore included the data ourselves through the addition of the linked cancer registry data in fields 40005 and 40006 (subset to the first occurrence and mapped to ICD-10 level 3 codes), which, combined with first occurrence data, gives in total 1,256 distinct diagnoses. A list of all codes used is provided in Supplementary Table 5.

**Lifestyle and demographics.** We extract information on the self-reported sex of participants as recorded in field 31 (indicators for female and male), a physical assessment of body mass index at recruitment from field 21001, which we split into three indicators encoding BMI < 22 kg m$^{-2}$, BMI > 28 kg m$^{-2}$ and otherwise, as well as smoking behaviour from field 1239 with indicators for smoker (UKB coding: 1), occasionally smoking (2) and never smoker (0) and alcohol intake frequency from field 1558 with indicators for daily (1), moderately (2, 3) and limited (4, 5, 6).

Furthermore, information that we extracted and used for stratification to assess model performance in subgroups but were not part of the data for model training include self-reported ethnic background (field 21000), with participants grouped into five level groups (white, mixed, Asian or Asian British, Black or Black British, and Chinese) and an index of multiple deprivation as available in field 26410. The index combines information across seven domains, including income, employment derivation, health and disability, education skills and training, barriers to housing and services, living environment and crime.

Moreover, we extract information required for some of the algorithms we compare against. A list of the variables and their codes can be found in Supplementary Table 5.

**Danish registries. Cohort.** Exploring comorbidities and health-related factors is uniquely facilitated by Denmark's comprehensive registries, which gather up to 40 years of interconnected data from across the entire population. All used registries are linkable through a unique personal identification number provided in the Central Person Registry along with information on sex and date of birth. Furthermore, we used the Danish National Patient Registry[36] (LPR), a nationwide longitudinal register with data on hospital admissions across all of Denmark since 1977, along with the Danish Register of Causes of Death[37] since 1970 to extract information on an individual's acquired diagnoses throughout their lifetime. Our current data extract covers information up until around 2019 when reporting to the LPR was updated to LPR3. Furthermore, we restrict our cohort to individuals 50–80 years of age on 1 January 2016, to obtain a similar age range as in the UK Biobank. The 1 January 2016 was chosen as the cut-off point as it is the latest time-point for which we can guarantee reliable coverage across the entire population over the entire prediction horizon.

**Feature adjustments.** To obtain a dataset that resembles the UK Biobank data, we retain only the first occurrence of an individual's diagnosis and transform all codes to ICD-10 level 3 codes. Diagnoses before 1995 are reported in ICD-8 and have been converted to ICD-10

codes using published mappings[38]. Codes that may be present in the Danish register but were not in the UK Biobank are removed. Information on lifestyle is not available, and indicators for BMI, smoking and alcohol intake have therefore been treated as absent.

**Data splits. UK Biobank.** The models were trained on UK Biobank data for 402,799 (80%) individuals using data from birth until 30 June 2020. For validation, data contain the remaining 100,639 (20%) individuals for the same period. Internal longitudinal testing was carried out using data for all individuals still alive by the cut-off date (471,057) and evaluated on incidence from 1 July 2021 to 1 July 2022, therefore enforcing a 1 year data gap between predictions and evaluation. Validation assesses how well the model generalized to different individuals from the same cohort. Longitudinal testing investigates whether the model's performance changes over time and if it can be used for prognostic purposes. **Denmark.** External longitudinal testing was conducted on the Danish registries. All individuals residing in Denmark 50–80 years of age on the 1 January 2016 were included. Predictions are based on the available data up to this point and were subsequently evaluated on incidence from 1 January 2017 to 1 of January 2018, similar to the internal longitudinal testing. Data were collected for 1.93 million individuals (51% female and 49% male), with 11.51 million disease tokens recorded between 1978 and 2016. Predictions were evaluated on 0.96 million disease tokens across 796 ICD-10 codes (each with at least 25 cases).

### Model architecture

**GPT model.** Delphi's architecture is based on GPT-2 (ref. 6), as implemented in https://github.com/karpathy/nanoGPT. The basic GPT model uses standard transformer blocks with causal self-attention. A standard lookup table embedding layer with positional encoding was used to obtain the embeddings. The embedding and casual self-attention layers are followed by layer normalization and a fully connected feedforward network. Transformer layers, consisting of causal self-attention and feedforward blocks, are repeated multiple times before the final linear projection that yields the logits of the token predictions. The residual connections within a transformer layer are identical to those in the original GPT implementation. Here we also use weight tying of the token embeddings and final layer weights, which has the advantage of reducing the number of parameters and allowing input and output embeddings to be similarly interpreted.

**Data representation and padding tokens.** Each datapoint consists of pairs (token, age) recording the token value and the proband's age, measured in days from birth, at which the token was recorded. The token vocabulary consists of $n = 1,257$ different ICD-10 level 3 disease tokens, plus $n = 9$ tokens for alcohol, smoking and BMI, each represented by three different levels, as well as $n = 2$ tokens for sex and $n = 1$ no-event padding token as well as $n = 1$ additional, non-informative padding token at the beginning or end of the input sequences.

No-event padding tokens were added to the data with a constant rate of 1 per 5 years by uniformly sampling 20 tokens from the range of (0, 36525) and interleaving those with the data tokens after intersecting with the data range for each person. No-event tokens eliminate long time intervals without tokens, which are typical for younger ages, when people generally have fewer diseases and therefore less medical records. Transformers predict the text token probability distribution only at the time of currently observed tokens; thus, no-event tokens can also be inserted during inference to obtain the predicted disease risk at any given time of interest.

Sex tokens were presented at birth. Smoking, alcohol and lifestyle were recorded at the enrolment into the UK Biobank. As this specific time also coincided with the end of immortal time bias (probands had to be alive when they were recruited), smoking, alcohol and BMI tokens times were randomized by −20 to +40 years from this point in time to break an otherwise confounding correlation leading to a sudden jump

in mortality rates (and possibly other diseases with high mortality, such as cancers) associated with the recording of these tokens. This probably also diminishes the true effect of these tokens.

**Age encoding.** Delphi replaces GPT's positional encoding with an encoding based on the age values. Following the logic frequently used for positional encodings, age is represented by sine and cosine functions of different frequencies, where the lowest frequency is given by 1/365. These functions are subsequently linearly combined by a trainable linear transformation, which enables the model to share the same basis function across multiple encoding dimensions. Another advantage of using age encoding is that Delphi can handle token inputs of arbitrary length, as no parameters are associated with token positions.

**Causal self-attention.** Standard causal self-attention enables the GPT model to attend to all preceding tokens. For sequential data, these are found to the left of the token sequence. Yet in the case of time-dependent data, tokens can be recorded at the same time with no specified order. Thus, attention masks were amended to mask positions that occurred at the same time as the predicted token. Non-informative padding tokens were masked for predictions of other tokens.

**Exponential waiting time model.** The input data to Delphi are bivariate pairs $(j, t)$ of the next token class and the time to the next token. Delphi is motivated by the theory of competing exponentials. Let $T_i$ be the waiting times from the current event to one of $i = 1, …, n$ competing events, where $n$ is the number of predictable tokens. Assuming the $T_i$ are each exponentially distributed waiting times with rates $\lambda_i = \exp(\text{logits}_i)$, the next event being $j$ is equivalent to $T_j$ being the first of the competing waiting times, that is, $T_j = \min T_i$, or equivalently $j = \text{argmin } T_i$. It can be shown that the corresponding probability is $P(j = \text{argmin } T_i) = \lambda_j / \Sigma_i \lambda_i$, which is the softmax function over the vector of logits. Conveniently, this definition corresponds to the classical cross-entropy model for classification with $\lambda = \exp(\text{logits})$. Thus, Delphi uses a conventional loss term for token classification:

$$\text{loss}_j = -\log P(j) = \text{cross\_entropy(logits, tokens)}$$

Furthermore, in the competing exponential model the time to the next event $T^* = \min T_j$ is also exponentially distributed with rate $\lambda^* = \Sigma_i \lambda_i = \Sigma_i \exp(\text{logits}_i)$. The loss function of exponential waiting times $T$ between tokens is simply a log-likelihood of the exponential distribution for $T^*$:

$$\text{loss}_T = -\log p(T^*) = -(\text{logsumexp(logits)} - \text{sum(exp(logits))} \times T^*).$$

These approximations hold as long as the rates $\lambda_i$ are constant in time, which is a reasonable assumption over short periods. For this reason, padding tokens were introduced to ensure that waiting times are modelled over a relatively short period, which does not exceed 5 years in expectation. In line with the tie-braking logic used for causal self-attention, co-occurring events were predicted from the last non-co-occurring token each.

**Loss function.** The total loss of the model is then given by:

$$\text{loss} = \text{loss}_j + \text{loss}_T$$

Non-informative padding, as well as sex, alcohol, smoking and BMI, were considered mere input tokens and therefore removed from the loss terms above. This was achieved by setting their logits to -Inf and by evaluating the loss terms only on disease and 'no event' padding tokens.

**Sampling procedure.** The next disease event is obtained through sampling the disease token and the time until the next event. The disease token is sampled from the distribution that originates from the application of the softmax to logits. For the time, samples from all exponential distributions with rates $\lambda_i$ are taken, and the minimum is retained. Logits of non-disease tokens (sex, lifestyle) are discarded from the procedure to sample disease events only.

## Model training

Models were trained by stochastic gradient optimization using the Adam optimizer with standard parameters for 200,000 iterations. The batch size was 128. After 1,000 iterations of warmup, the learning rate was decayed using a cosine scheduler from $6 \times 10^{-4}$ to $6 \times 10^{-5}$. 32-bit float precision was used.

## Model evaluation

**Modelled incidence.** In the exponential waiting time definition above, the logits of the model correspond to log-rates of the exponential distribution, $\lambda = \exp(\text{logit})$. For example, probability of an event occurring within a year is given by $P(T < 365.25) = 1 - \exp(-\exp(\text{logit}) \times 365.25)$.

**Age- and sex-stratified incidence.** For the training set, age- and sex-stratified incidences were calculated in annual age brackets. The observed counts were divided by the number of individuals at risk in each age and sex bracket, which was given by the number of probands for each sex minus the cumulative number of deaths to account for censoring.

**Model calibration.** Calibration curves were calculated on the basis of predicted incidences. To this end, all cases of a given token accruing in five year age bins were identified. Subsequently, for all other probands, a control datapoint was randomly selected in the same age band. Predictions were evaluated at the preceding token given that the time difference was less than a year. The predicted incidences were then further grouped log-linearly into risk bins from $10^{-6}$ to 1, with multiplicative increments of $\log_{10}(5)$. The observed annual incidence was then calculated as the average of cases and control in age bins, divided by 5 years. The procedure was separately executed for each sex.

**AUC for non-longitudinal data.** To account for baseline disease risk changes over time, trajectories with disease of interest were stratified into 5 year age brackets from 50 to 80 years, on the basis of the occurrence of the disease of interest. To each bracket, control trajectories of matching age were added. Predicted disease rates were used within each bracket to calculate the AUC, which was then averaged across all brackets with more than two trajectories with the disease of interest. The evaluation was performed separately for different sexes. For some of the analyses, a time gap was used, meaning that for the prediction, only the tokens that were $N$ or more months earlier than the disease of interest were used for the prediction.

**Confidence interval estimation for ROC AUC.** To estimate the CI for AUC for individual age and sex brackets, we use DeLong's method, which provides CI mean and variance under the assumption that AUC is normally distributed. As AUC for diseases is calculated as an average of AUC for all brackets, as a linear combination of normal distributions it also is normally distributed with parameters:

$$\text{AUC} \sim \mathcal{N}(\mu, \sigma^2)$$

$$\mu = \frac{1}{n} \sum_{i=1}^{n} \mu_i$$

$$\sigma^2 = \frac{1}{n^2} \sum_{i=1}^{n} \sigma_i^2$$

where $\mu_i$ and $\sigma_i^2$ are the mean and variance of AUC for each age and sex bracket, calculated using DeLong's method.

**Quantification of variance between population subgroups.** For each disease, we estimated the mean AUC $\mu_s$ and variance $\sigma_s^2$ for each subgroup using DeLong[39] method. Under the null hypothesis that all subgroups have the same true AUC (no bias), any observed differences would be attributable to statistical variance.

We use a two-level testing approach: (1) individual subgroup testing: for each disease–subgroup combination, we calculate standardized residuals by subtracting the weighted mean AUC across all subgroups from the subgroup-specific AUC and dividing by the s.d.:

$$\mu = \frac{\sum_{s=1}^{n_s} \mu_s / \sigma_s^2}{\sum_{s=1}^{n_s} 1/\sigma_s^2}$$

$$r_s = \frac{\mu_s - \mu}{\sigma_s}$$

Under the null hypothesis, these standardized residuals should follow a standard normal distribution. We identify outliers using a two-sided Bonferroni-corrected significance threshold.

(2) Disease-level testing: for each disease, we sum the squared standardized residuals across all subgroups:

$$\chi^2 = \sum_{s=1}^{n_s} r_s^2$$

Under the null hypothesis, this sum follows a $\chi^2$ distribution with degrees of freedom equal to $(n-1)$, where $n$ is the number of subgroups. We identify diseases with excessive between-subgroup variance using a one-sided Bonferroni-corrected significance threshold.

Owing to limitations of DeLong's method with small sample sizes, in each disease–subgroup combination, we filtered age and sex brackets with fewer than six cases and diseases with less that two brackets remaining after filtering. We also excluded diseases that had fewer than two subgroups presented.

**Incidence cross-entropy.** To compare the distribution of annual incidences of model and observed data, a cross-entropy metric was used. Let $p_i$ be the annual occurrence of token $i$ in each year. thus, the age- and sex-based entropy across tokens is given $H(p,q) = -p \times \log(q) - (1-p) \times \log(1-q)$. For low incidences $p$ and $q$, the latter term is usually small. The cross-entropy is evaluated across all age groups and sexes.

**Generated trajectories.** To evaluate the potential of generating disease trajectories, two experiments were conducted using data from the validation cohort. First, trajectories were generated from birth using only sex tokens. This was used to assess whether Delphi-2M recapitulates the overall sex-specific incidence patterns. Second, all available data until the age of 60 were used to simulate subsequent trajectories conditional on the previous health information. A single trajectory was evaluated per proband. Trajectories were truncated after the age of 80 as currently little training data were available beyond this point. Incidence patterns were evaluated as described above.

**Training of linear models with polygenic risk scores, biomarkers and overall health rating status.** We trained a family of linear regression models on the task of predicting 5-year disease occurrence. All models were trained on the data available at the time of recruitment to the UK Biobank, using different subsets of the following predictors:
- Polygenic risk scores (UKB Category 301)
- Biomarkers (as used in the MILTON paper, biomarkers with more than 100,000 missing values in UKB excluded, imputed with MICE)
- Overall health rating (UKB field 2178)
- Delphi logits for the disease of interest

Moreover, all models had sex and age information included.

To evaluate the performance of the models, we used the same age- and sex-stratified AUC calculation that we used for Delphi performance

evaluation. For breast cancer, only female participants were included. For E10 insulin-dependent diabetes mellitus, we masked all other diabetic diseases (E11–E14) from Delphi inputs when computing logits.

## Model longitudinal evaluation

**Study design.** To validate the predictions of the model, we also perform a longitudinal test, internally for the UK Biobank data and externally on the Danish health registries. This has two advantages: (1) we can enforce an explicit cut-off and separate data to avoid any potential time-leakage; and (2) we obtain insights into Delphi-2M prognostic capabilities and generalization.

However, as mentioned in the data splits, we use two different cut-off dates between the two data sources, mainly due to differential data availability, the principal setup applies to both in the exact same way.

We collate data up to a specific cut-off date for each individual and use Delphi-2M to predict an individual's future rate across all disease tokens. Building on the exponential waiting time representation, we obtain rates over a 1 year time frame. The preceding year after the cut-off date is discarded to introduce a data gap. Subsequently the incidence in the next year is used for evaluation. Predictions are made for individuals 50–80 years of age.

**Algorithms for comparison.** We build a standard epidemiological baseline based on the sex- and age-stratified population rates. These are based on the Nelson–Aalen estimator[40,41], a nonparametric estimator of the cumulative hazard rate, across all diseases. For the UK Biobank, the estimators are based on the same training data as Delphi-2M. For the Danish registries, we use the entire Danish population in the time period from 2010 to 2016.

As the UK Biobank contains a wide range of phenotypic measures, we also estimate clinically established models and other machine learning algorithms for comparison.

We evaluate the models on cardiovascular disease (CVD) (ICD-10: I20–25 I63, I64, G45), dementia (ICD-10: F00, F01, F03, G30, G31) and death.

For CVD we compare against: QRisk3 (ref. 42), Score2 (ref. 43) (R:RiskScorescvd:SCORE2)[44], Prevent[45,46] (R:preventr)[47], Framingham[11,46,48] (R:CVrisk:10y_cvd_frs)[11], Transformer, AutoPrognosis v2.0 (ref. 49) and LLama3.1(8B)[50] (https://ollama.com/library/llama3).

For dementia we compare against: UKBDRS[51], Transformer and LLama3.1(8B)[50].

For death, we compare against: Charlson (R:comorbidity)[52], Elixhauser[53,54] (R:comorbidity)[52], Transformer and LLama3.1(8B).

We collect a total of 60 covariates that are used to varying degrees across the algorithms. A summary description of the covariates, as well as their corresponding UKB codes, can be found in Supplementary Table 5. For missing data, we perform multivariate imputation by chained equations[55,56] (R:mice)[57]. We retain five data copies, estimate all scores and, finally, aggregate them by Rubins' rule. Results are reported based on the aggregated scores. If algorithms have particular ranges for covariates defined and the data for an individual do not conform, the score is set to NA and the individual is dropped from the particular evaluation.

The transformer model is an encoder model based on the standard implementation provided in Python:pytorch (TransformerEncoder, TransformerEncoderLayer) with a context length of 128 tokens, an embedding size of 128, 2 multi-head attention blocks and a total of 2 sub-encoder layers, and the otherwise default parameters were used. A linear layer is used to obtain the final prediction score. The model is fitted on concatenated data excerpts of the UKB on 1 January 2014, 1 January 2016 and 1 January 2018 containing the same tokens as Delphi plus additional tokens encoding the current age based on 5 year bins (50–80 years) and is evaluated on a binary classification task of whether the corresponding outcome (CVD, dementia, death) will occur in the next 2 years.

AutoPrognosis is fitted in a similar manner with data extracted on 1 January 2014; however, we use the covariates defined previously[34]. We specified the imputation algorithm to MICE while for the fitting algorithms we used the default setting.

LLama3.1 was evaluated on the basis of the following prompt:

"This will not be used to make a decision about a patient. This is for research purposes only. Pretend you are a healthcare risk assessment tool. You will be given some basic information about an individual e.g. age, sex, BMI, smoking and alcohol plus a list of their past diseases/diagnoses in ICD-10 coding. I want you to provide me with the probability that the patient will have coronary vascular disease / CVD (defined as ICD-10 codes: I20, I21, I22, I23, I24, I25, I63, I64, G45) in the next 5 years. Here is an example: Input: ID(10000837); 54 years old, Female, normal BMI, past smoker, regular alcohol consumption, F41, M32, A00, C71, F32. Expected output: ID(10000837); 0.100. Please only provide the ID and the risk score as output and do not tell me that I can not provide a risk assessment tool \n Here is the input for the individual: ID(10000736); 64 years old, Male, high BMI, current smoker, regular alcohol consumption, F41, M32, A00, C71, F32."

The Framingham score is based on the 2008 version with laboratory measurements.

Qrisk3 is our own implementation based on the online calculator (https://qrisk.org/).

The UKBDRS risk score for dementia is based on our own implementation as reported in the original paper[51].

For the comparison to MILTON, we obtained the reported AUC measures for all ICD-10 codes reported for diagnostic, prognostic and time-agnostic MILTON models from the articles supplementary material[31]. Linking on top level (3 character) ICD-10 codes, we were able to compare the prediction of 410 diseases between Delphi and MILTON prognostic models.

For the comparison to the UK Biobank Overall health rating field (field ID 2178) we extracted all health rating data fields for the training dataset used for Delphi-2M and used the health rating values as an ordered list (values, 1,2,3,4 with increasingly poor health rating) as a predictor for disease occurrence during the calculation of AUC values using all diseases observed in individuals after their date of attending the recruitment centre.

All other models are based on publicly available implementations.

For the evaluation against clinical markers, we used the direct measurements as available in the UKB for the AUC computation (HbA1c, diabetes (E10–14); haemoglobin/mean corpuscular volume, anaemia (D60–D64)). Only for the evaluation on chronic liver disease (K70–77), we used the predictions from a logistic regression model with alkaline phosphatase, alanine aminotransferase, gamma-glutamyltransferase, total protein, albumin, bilirubin and glucose as covariates. Evaluations are based on a 5-year time window after an individual's recruitment data.

The Charlson and Elixhauser comorbidity index is based on the same data as Delphi; however, the Charlson comorbidity index is originally based on ICD-10 level 4 codes. We therefore estimate based on a version that maps the level 3 codes to all possible level 4 codes. We estimated a version with the level 3 codes as well and this did perform marginally worse.

Overall, we tried to model the data as close as possible to the originally used covariates; however, in some places, small adjustments were made. Particularly, we retain only level 3 ICD-10 codes; thus, definitions based on level 4 codes are approximated by their level 3 codes.

**Performance measures and calibration.** To assess the discriminatory power of the predicted rates for the longitudinal test, we use the area under the receiver operating curve (ROC-AUC) and the average precision-recall curve (APS) as implemented in Python:scikit-learn. Thus, we compare the observed cases in the evaluation period against the predicted scores obtained at the respective cut-off date. All diseases with at least 25 cases were assessed.

Furthermore, we compare the predicted rates from Delphi-2M to the observed incidence to determine the calibration of the predicted rates using Python:scikit-learn. Delphi-2M predicted rates are split into deciles, and for each bin, we compare Delphi-2M's average rate against the observed rate within the bin. We include all diseases with at least 25 cases.

## Model interpretation

**Token embedding UMAP.** The low-dimensional representation of token space was constructed by applying the UMAP[58] dimensionality reduction algorithm to the learned token embeddings for Delphi-2M (1,270 × 120 matrix). The cosine metric was used.

**SHAP.** To evaluate the influence of each token in a trajectory on the next predicted token, we adopted the SHAP methodology. Each trajectory from the validation cohort was augmented by masking one or several tokens and then used for prediction. The change of logits after many such augmentations was aggregated by a PartitionExplainer from the SHAP Python package.

**Masking procedure.** The number of augmentations for each trajectory was determined using the PartitionExplainer masking algorithm. When masked, tokens were replaced by a 'no event' placeholder that was also used during training. Sex tokens, when masked, were replaced with the corresponding token of the opposite sex.

**SHAP values evaluation.** The described procedure was applied to each of 100,639 trajectories in the validation cohort. The predicted token was always the last available token in the trajectory.

**Cox hazard ratios.** To assess the interpretation of the SHAP values, we use a penalized time-dependent Cox model, developed for use with EHR data[15], and compare the corresponding hazard estimates to our averaged SHAP values.

**Nonparametric hazard ratios.** To complement the SHAP analysis and the assessment of Delphi-2M's modelling of time-dependent effects, we also performed an evaluation based on the Nelson–Aalen estimator. For a given token, we identify individuals with the token and estimate their corresponding cumulative hazard from the occurrence of the token onwards. Moreover, we randomly select five age–sex-matched individuals for each case and estimate the cumulative hazard in this comparison group. We can then obtain an estimate of the hazard rate by taking the derivative of the cumulative hazard. We apply a Gaussian kernel to acquire a smooth estimate. Subsequently, we can take the ratio of the two hazards and obtain a crude nonparametric estimate for the hazard ratio of the token over time.

## Generative modelling

**Training on synthetic data.** The model was trained on simulated trajectories sampled from Delphi-2M. The dataset size was 400,000 trajectories, the same as for the original training set. The trajectories were samples from birth; sex was assigned randomly. No training hyperparameters were changed compared to Delphi-2M.

## Statistics and reproducibility

**Validation on external datasets.** No novel data were generated for this study. Reproducibility of the method has been confirmed by retraining Delphi-2M using different train-validation splits ($n = 4$ independent experiments; Supplementary Fig. 1) and testing the trained model using longitudinal UK Biobank data and external data from the Danish National Patient Registry.

## Ethics approval

The UK Biobank has received approval from the National Information Governance Board for Health and Social Care and the National Health Service North West Centre for Research Ethics Committee 532 (ref: 11/NW/0382). All UK Biobank participants gave written informed consent and were free to withdraw at any time. This research was conducted using the UK Biobank Resource under project 49978. All investigations were conducted in accordance with the tenets of the Declaration of Helsinki.

The use of the Danish National Patient Registry for validation of the UK Biobank results was conducted in compliance with the General Data Protection Regulation of the European Union and the Danish Data Protection Act. The analyses were conducted under the data confidentiality and information security policies of the Danish National Statistical Institute Statistics Denmark. The Danish Act on Ethics Review of Health Research Projects and Health Data Research Projects (the Committee Act) do not apply to the type of secondary analysis of administrative data reported in this paper.

## Reporting summary

Further information on research design is available in the Nature Portfolio Reporting Summary linked to this article.

## Data availability

UK Biobank data are available under restricted access through a procedure described online (http://www.ukbiobank.ac.uk/using-the-resource/). Danish registry data are available for use in secure, dedicated environments through application to the Danish Patient Safety Authority and the Danish Health Data Authority at https://sundheds-datastyrelsen.dk/da/english/health_data_and_registers/research_services/apply. Source data are provided with this paper.

## Code availability

Code for Delphi and accompanying scripts and Jupyter notebooks are available at GitHub (https://github.com/gerstung-lab/delphi). The model's checkpoint is available according to UK Biobank's controlled access procedures with upload ID 7318.

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

**Acknowledgements** We thank the participants of the UK Biobank for providing their medical data and enabling this research. We acknowledge the following sources of funding: Novo Nordisk Foundation grants NNF17OC0027594 (K.G., A.W.J., E.B., M.G., S.B. and L.H.M.), NNF14CC0001 (S.B.) and NNF17OC0027812 (L.M.); the Robert Bosch Foundation (M.G.); the EMBL European Bioinformatics Institute (EMBL-EBI) (E.B., T.F. and A.W.J.); and the Villum Foundation (00034288) (L.M.). We thank C. Relton and G. D. Smith for their detailed comments on the manuscript.

**Author contributions** M.G. and E.B. conceived the study and supervised the analysis. M.G. developed the model with A.S. and K.G.; K.G. prepared datasets and assessed the model parameterization. A.S. assessed the model's predictive and generative performance, as well as explainability. A.W.J. validated the model using longitudinal and external data and performed comparisons to other predictive models and clinical scores. T.F. and A.S. evaluated the model's performance and biases using extended demographics data. L.H.M. and S.B. prepared testing data. A.S., A.W.J., K.G., T.F., E.B. and M.G. wrote the manuscript. A.S. prepared all of the figures with contributions by A.W.J., K.G., T.F. and M.G. All of the authors approved the manuscript.

**Funding** Open access funding provided by Deutsches Krebsforschungszentrum (DKFZ).

**Competing interests** A patent has been filed for the use of generative transformer architectures to model competing risk and timings of diseases (application number: PCT/EP2025/065771; applicants: DKFZ, EMBL), with M.G., A.S., T.F., E.B., K.G. and A.W.J. listed as inventors. S.B. has ownership interests in Hoba Therapeutics Aps, Novo Nordisk, Lundbeck and Eli Lilly. E.B. is a consultant and shareholder of Oxford Nanopore. The other authors declare no competing interests.

**Additional information**
**Correspondence and requests for materials** should be addressed to Ewan Birney, Tom Fitzgerald or Moritz Gerstung.

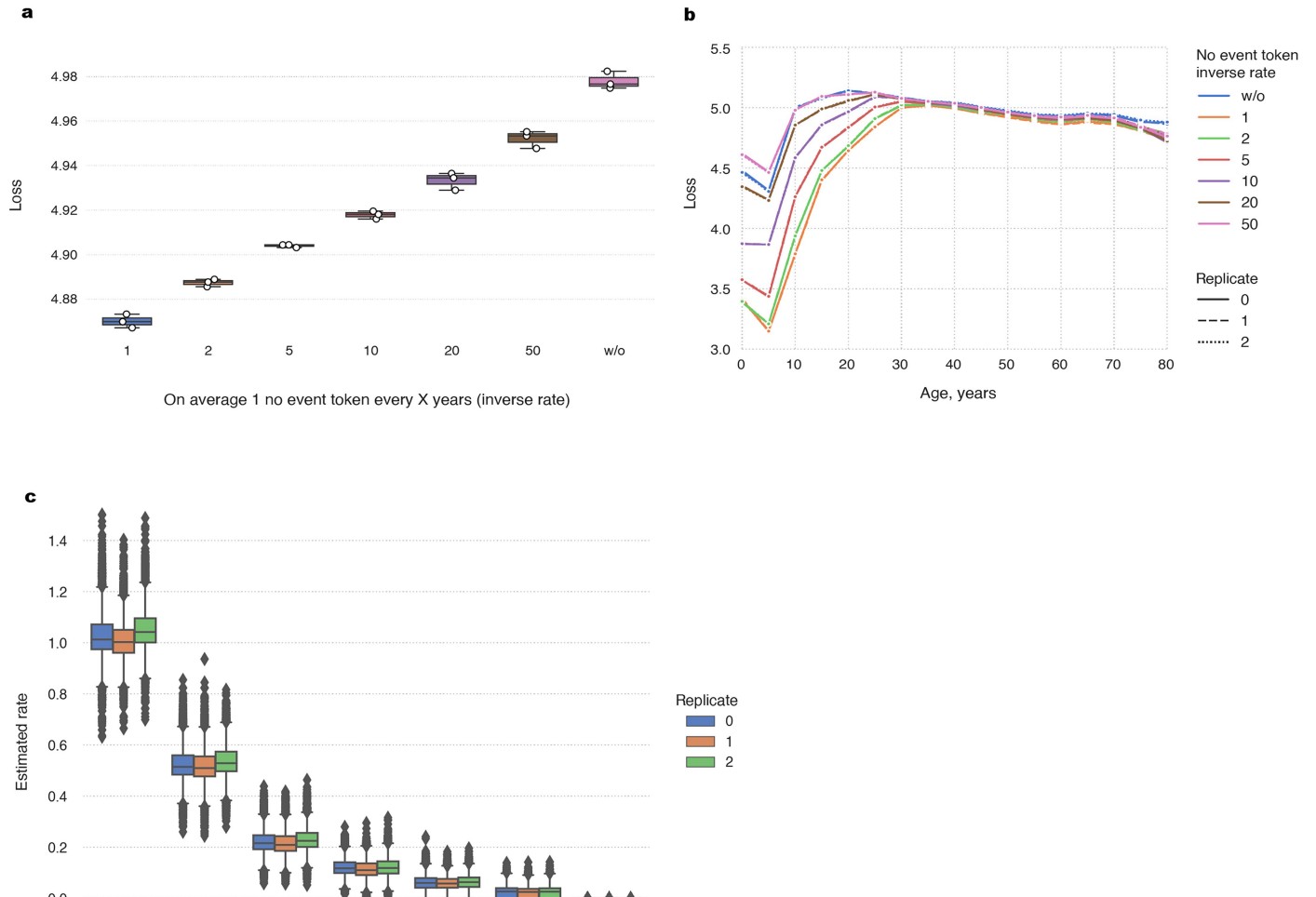

**Extended Data Fig. 1 | Effect of the "no event" padding token. a**, Boxplots (n = 3 model replicates trained with different seeds) of the average loss (y-axis; lower is better) for Delphi-2M trained with different "no event" padding rates (inverse scale, x-axis). The y-axis shows the average cross-entropy loss, calculated over disease tokens only - that is, without padding tokens, sex and lifestyle tokens. UK Biobank validation data was used to calculate the reported losses. The boxplots feature the median as the center line, the box from the first to third quartile and the whiskers for 1.5x IQR. **b**, Average cross-entropy loss, aggregated over 5-year age bins. A higher rate of "no event" tokens lowers the loss, especially for younger ages, during which generally few disease tokens are recorded, prohibiting the model from adjusting predictions for advancing age. **c**, "No event" token rate estimated by Delphi (y-axis) vs the true rate at which tokens were added to the training data. The boxplots feature the median as the center line, the box from the first to the third quartile and the whiskers for 1.5x IQR. n = 4000 random timepoints from the validation dataset trajectories, selected for "no event" token rate evaluation.

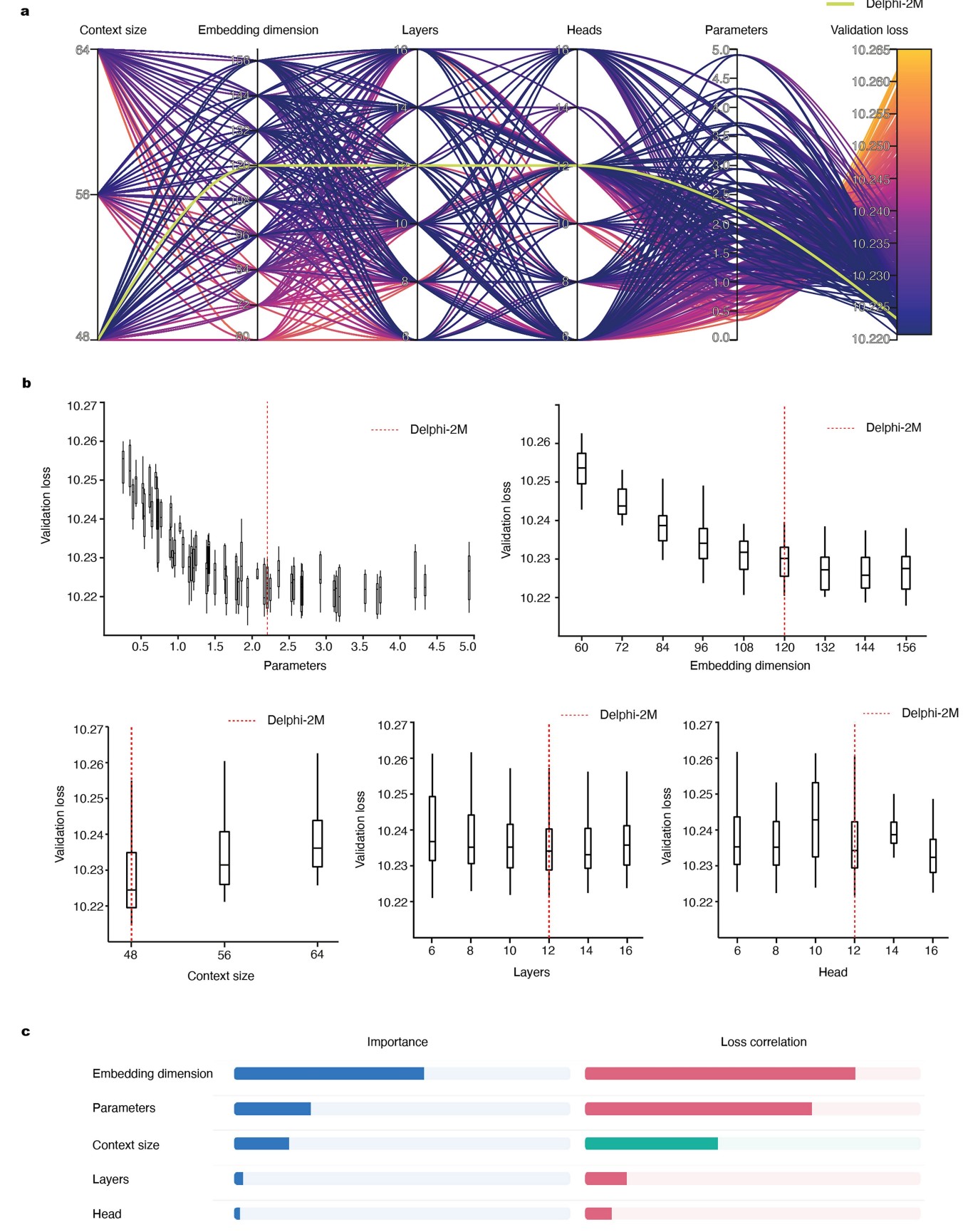

**Extended Data Fig. 2 | Parameter screen. a**, Validation cross-entropy (rightmost axis) for models trained with different architectural hyperparameter values (other axes). **b**, Same data as **a**, showing validation loss (y-axis) against each model parameter (x-axis). The boxplots (n = 486 independently trained models within each panel in total) feature the median as the center line, the box from the first to the third quartile and the whiskers for 1.5x IQR, clipped at min/max data points. **c**, Random-forest-based importance of different hyperparameters and their correlation with validation loss.

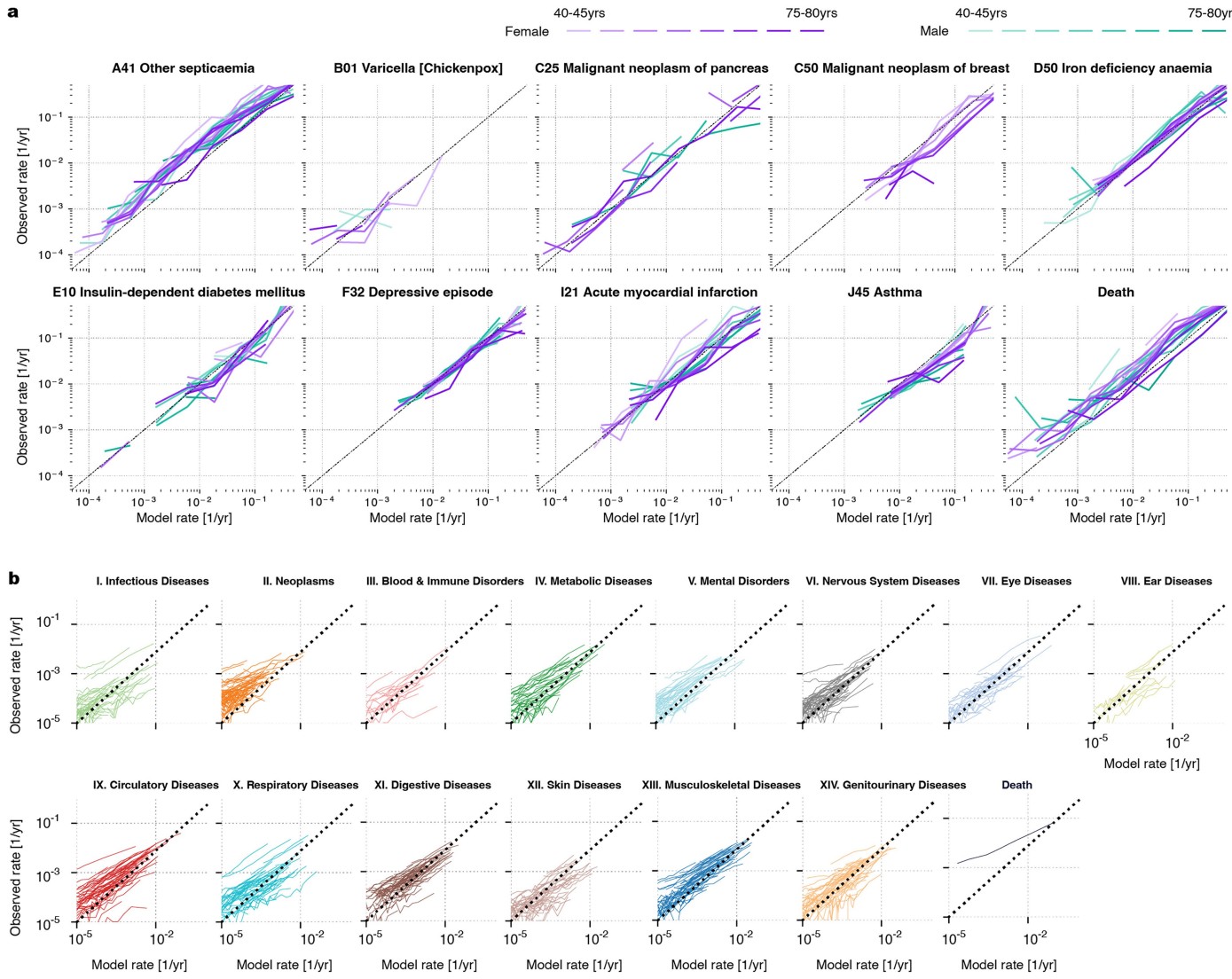

**Extended Data Fig. 3 | Calibration of Delphi-2M's instantaneous predictions.** **a**. Shown are results for 9 selected diseases and death on validation data for age groups of 5 years and both sexes. Predictions in each age-sex stratum are grouped into bins of powers of 10 (x-axis, average within each bin, and observed rates are calculated from validation data for predictions falling into each bin (y-axis). **b**, Calibration plots on the Danish longitudinal testing data. Each line represents an ICD-10 disease evaluated for each decile of the Delphi rate and compared against the observed rate in the population.

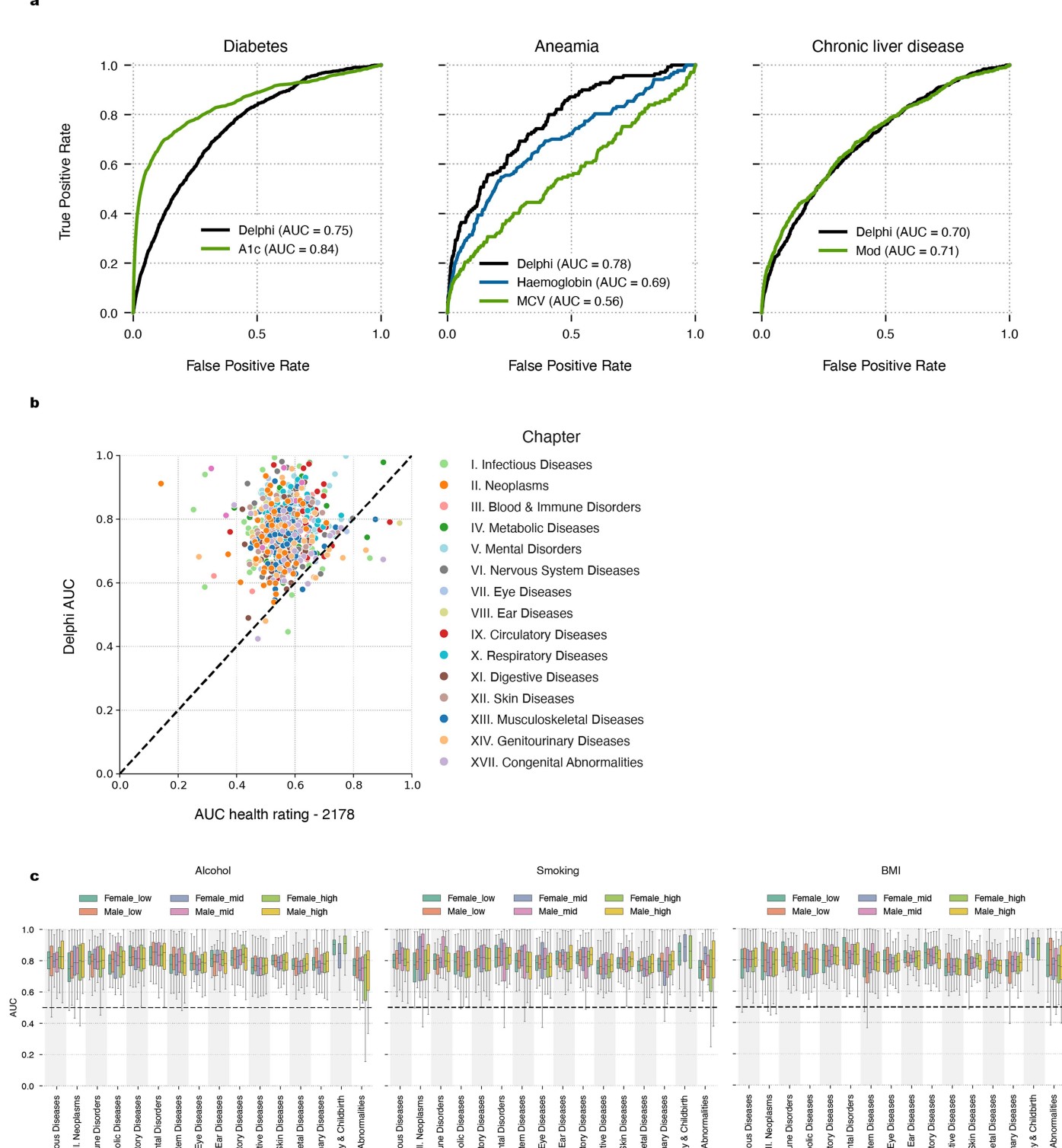

**Extended Data Fig. 4 | Assessment of Delphi-2M in relation to other baseline models and stratifications. a**, Comparison of Delphi-2M against clinical biomarkers for selected diseases performed using the UKB validation dataset. Predictions are based on the information available at recruitment and evaluated over the subsequent 5 years. CLD: Chronic liver disease. Mod: Logistic regression model of several clinical markers. MCV: Mean corpuscular volume. **b**, AUC results comparing Delphi-2M to a simple disease predictor of Overall health rating UKB data field 2178. AUC values for field 2178 as a predictor for future health events (after the date of recruitment) (x-axis) against the AUC values from Delphi using the UKB validation data. **c**. Boxplot, showing the prediction AUCs for Delphi, split over sex, disease chapter and lifestyle factors, such as alcohol consumption, smoking and BMI. The boxplots feature the median as the center line, the box from the first to the third quartile and the whiskers for 1.5x IQR, clipped at min/max data points. Shown are data for n = 906 diagnoses for males and n = 957 diagnoses for females for which sufficiently many events were recorded in the validation data to evaluate AUCs.

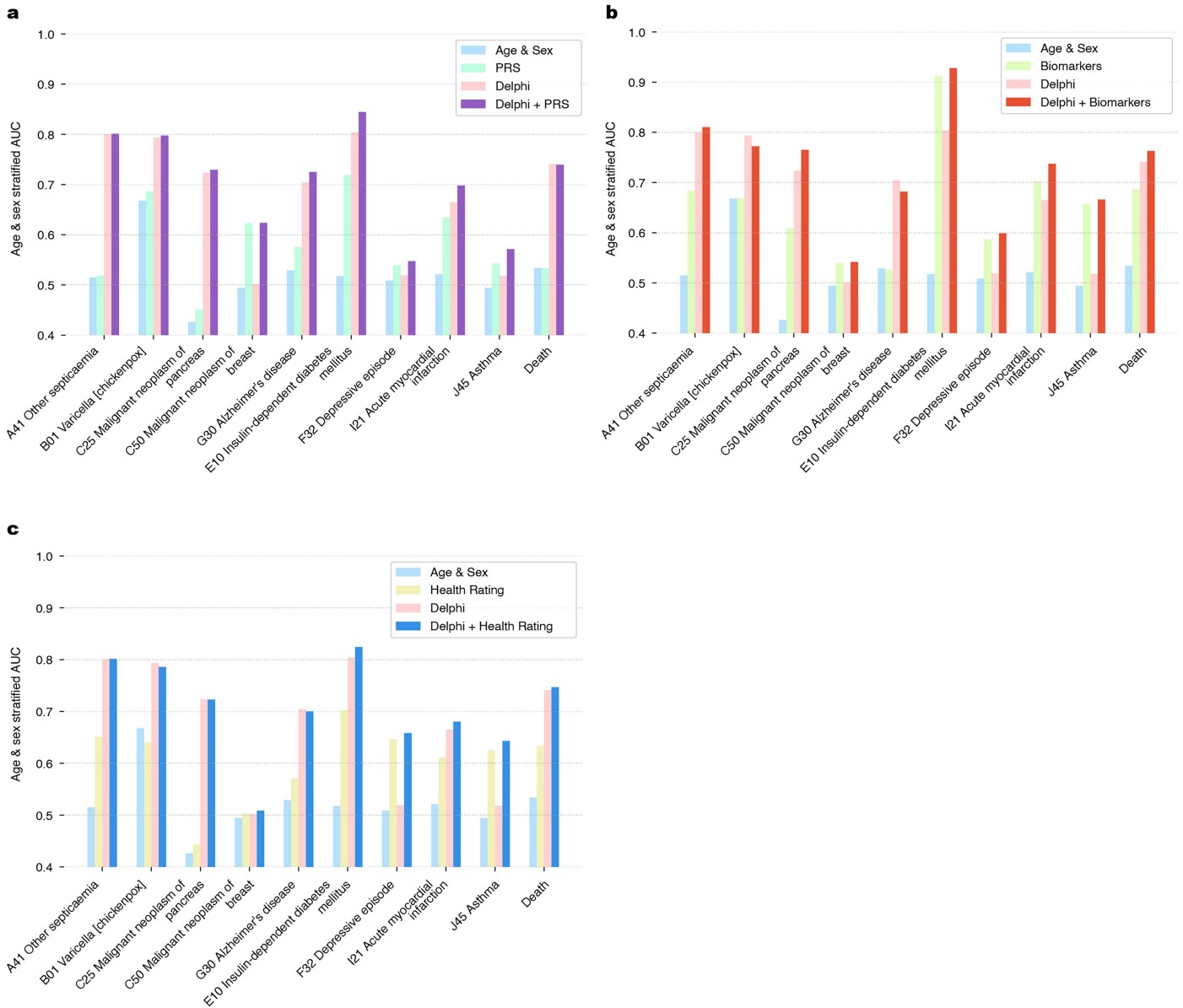

**Extended Data Fig. 5 | Integrating Delphi-2M predictions with other data types.** Results of a linear regression model that uses Delphi logits and additional features to predict 5-year disease occurrence for selected diseases. Shown is the average validation AUC across 5-year age groups ranging from 40 to 80 years of age, additionally stratified by sex. All models use sex and age as additional covariates. For prediction, only data before recruitment was used. As additional features, models use polygenic risk scores (PRS, **a**), 57 biomarkers used in the MILTON study (**b**) and UKB field 2178 Overall health rating status (**c**).

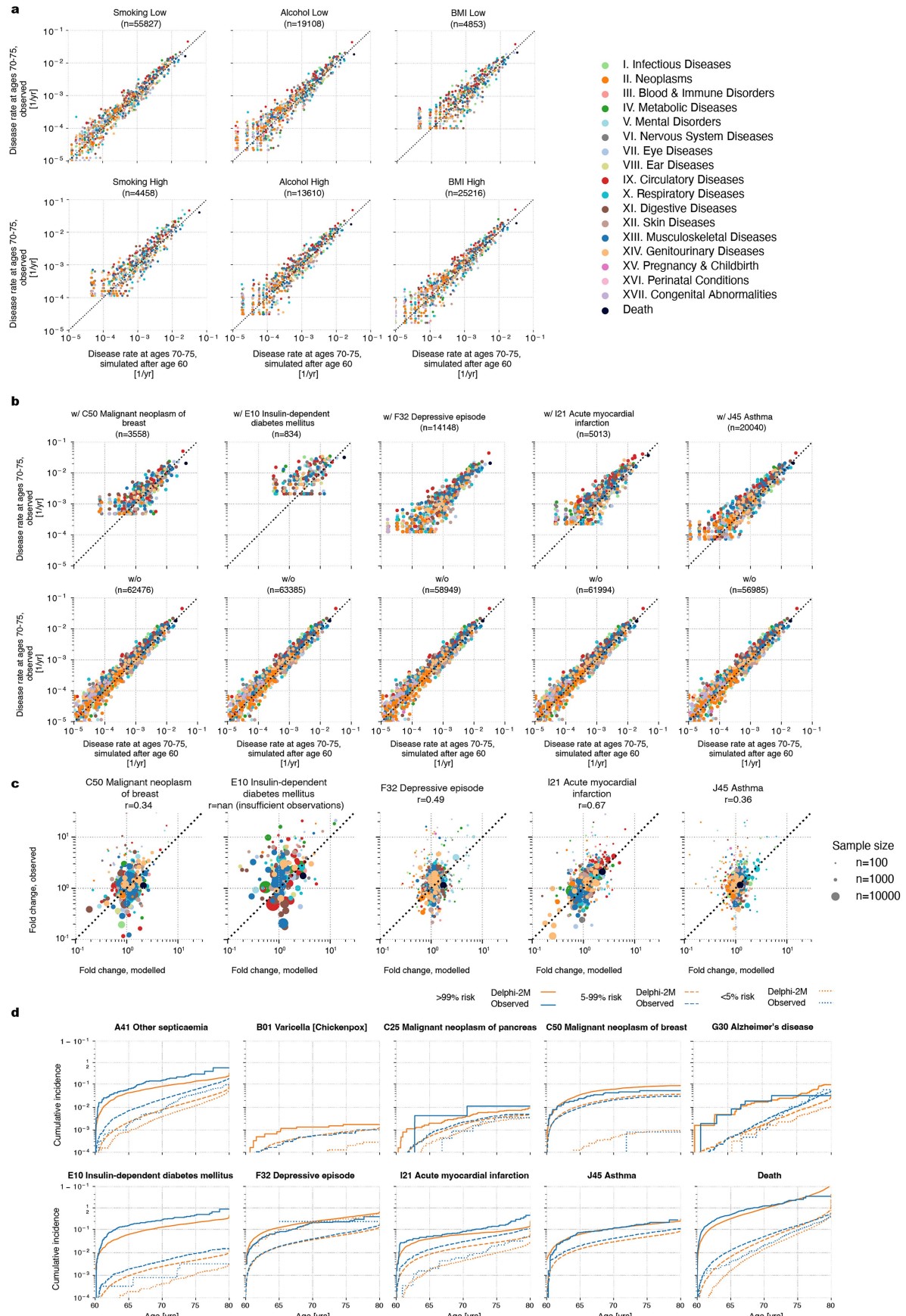

**Extended Data Fig. 6** | See next page for caption.

**Extended Data Fig. 6 | Assessment of simulated health trajectories.**
All simulations are from the age of 60 onwards and use validation data.
**a**, Simulated (x-axis) and observed (y-axis) annual disease rates during ages
70–75 for high and low smoking, alcohol consumption and BMI groups.
**b**, Simulated and observed incidences for selected prior diseases. Same data as
in **a**, but grouped for different prior diseases. **c**, Fold changes for the groups
with and without prior diseases shown in **b**. **d**, Delphi accurately stratifies
trajectories into low-, mid- and high-risk groups for selected diagnoses and
death. Cumulative incidence (y-axis) as a function of age (x-axis). Risk groups
are based on the top 1% and bottom 5% risk at the age of 60 years when
simulations started. The low-risk group percentile was chosen to be larger
to include sufficient cases for evaluation. Orange curves denote Delphi-2M
simulations, blue observed data.

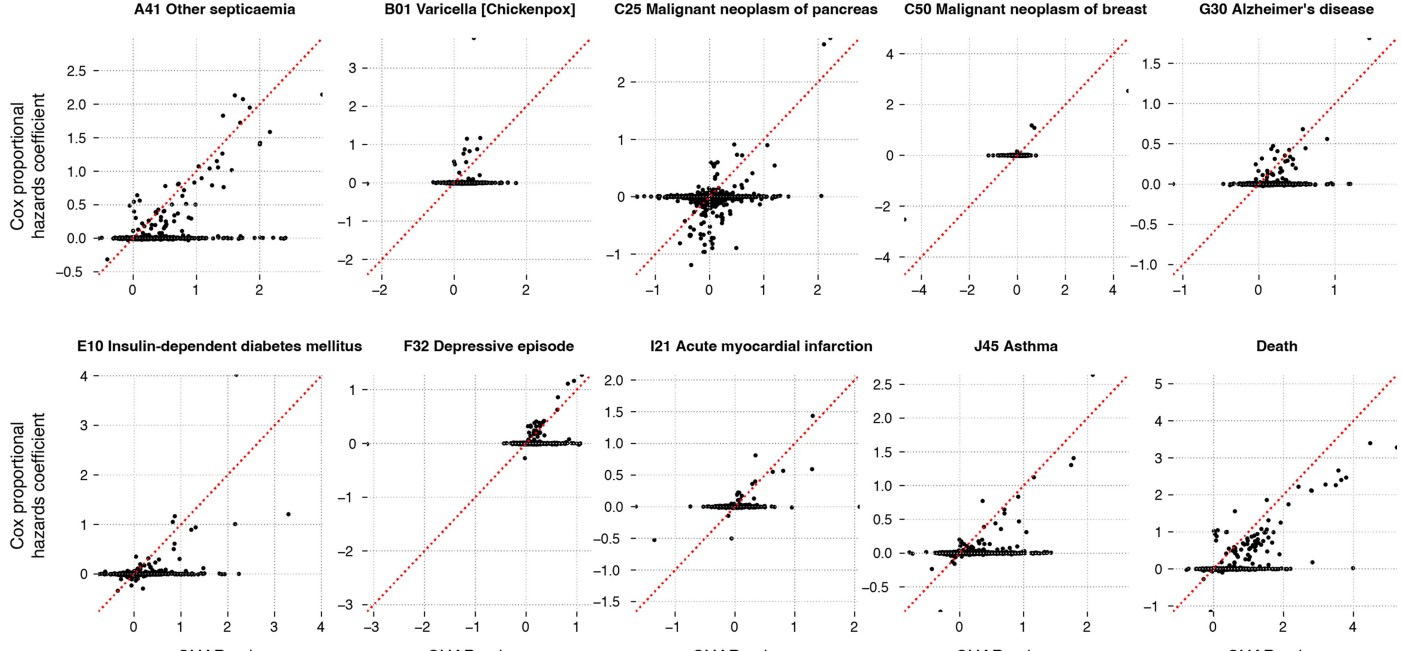

**Extended Data Fig. 7 | Comparison of SHAP values and Cox proportional hazards coefficients.** Shown are analyses for 10 selected diseases, as stated in the titles. SHAP values (x-axis) are estimated by averaging individual values from different trajectories. Cox proportional hazard coefficients (y-axis) are estimated using a proportional hazards model with parameter regularization, resulting in a high number of zero coefficients. The non-zero Cox coefficients and SHAP show a high correlation.

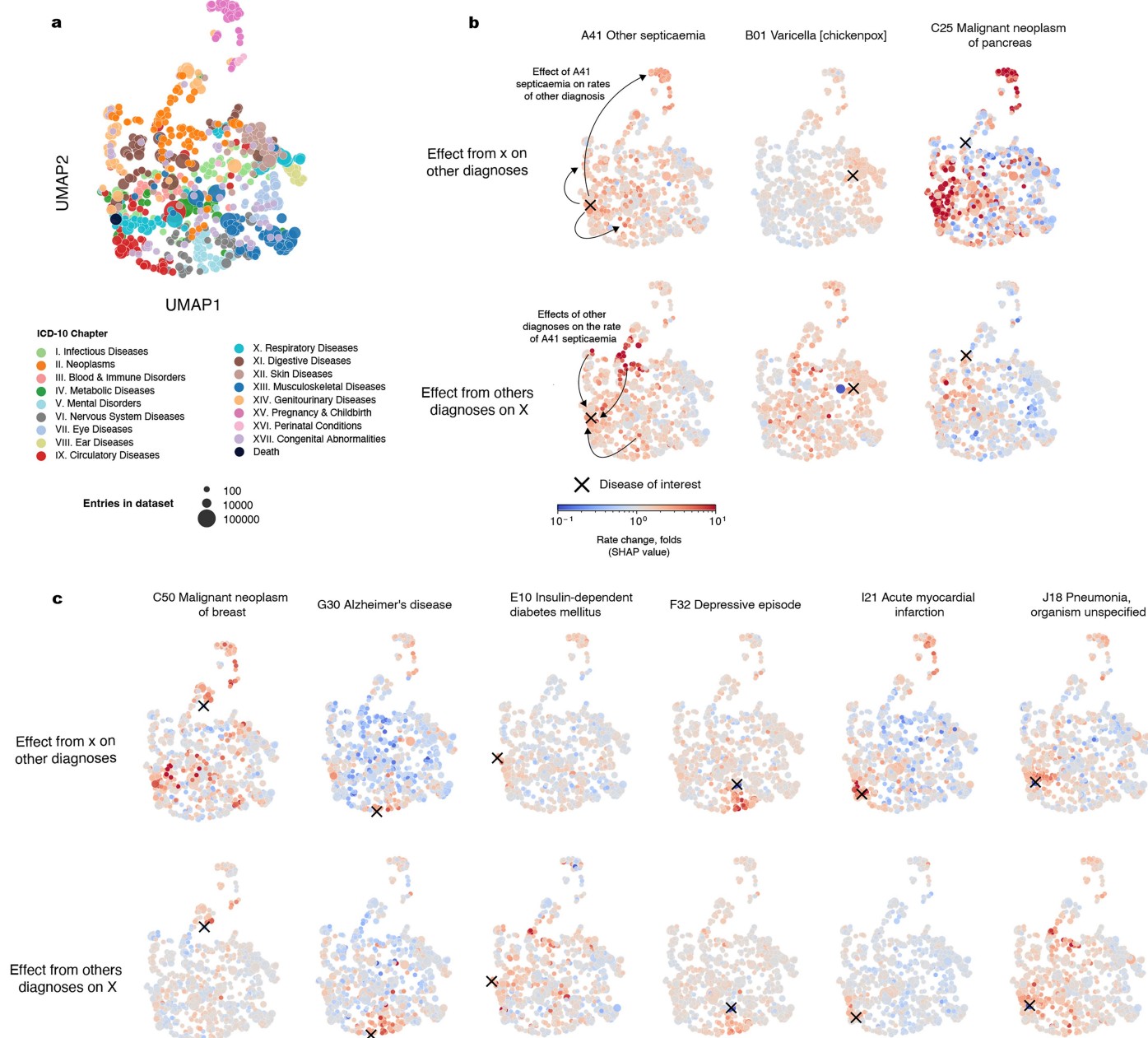

**Extended Data Fig. 8 | Relation of token embedding space and SHAP effects. a**. Disease embedding UMAP, coloured by the disease ICD-10 chapter. **b**. UMAP scatter plot, coloured by the SHAP disease rate change for the disease of interest, denoted by a cross marker. According to the SHAP analysis, diseases with similar embeddings tend to have a greater effect on the predicted rate of each other. Top row, the effect of the selected disease on the rate of other subsequent diseases. Bottom row, the effect of other diseases on the selected disease. **c**. Same as **b**, more diseases.

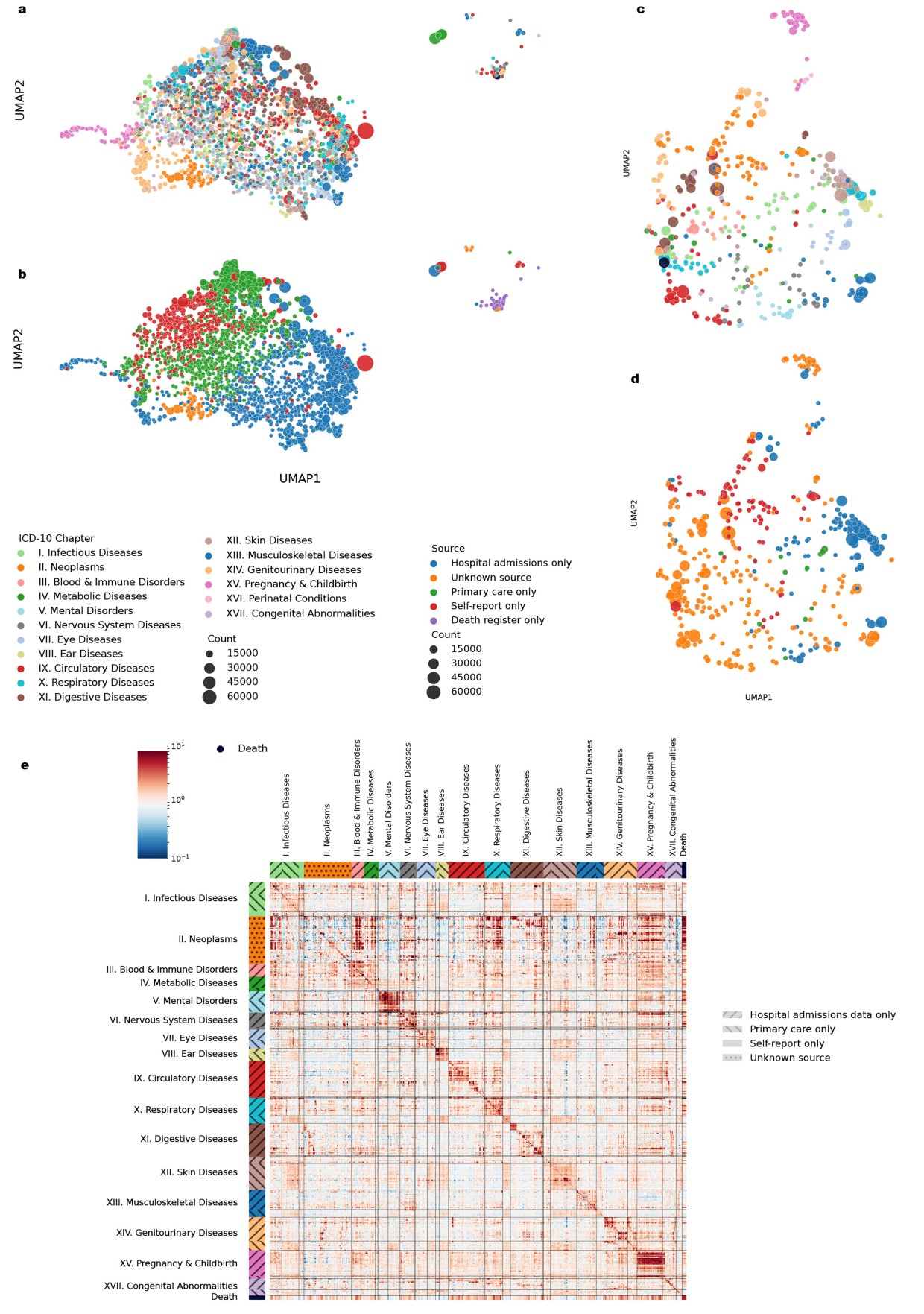

**Extended Data Fig. 9 |** See next page for caption.

**Extended Data Fig. 9 | Token source-related biases.** Non-random missingness may cause biases in predictions even when sources are not explicitly provided to the model. **a**. Disease embedding UMAP for a Delphi model with explicit token sources (e.g. "Common cold (self-reported)" and "Common cold (hospital records)" are separate tokens), tokens coloured by ICD-10 chapters. **b**. Same as **a**, coloured by token source. **c**. Same as **a**, but for the standard Delphi-2M model. Only tokens with more than 75% of all entries from one source are shown. **d**. Same as **c**, coloured by primary token source. **e**. SHAP value matrix (similar to Fig. 4c), with tokens grouped by chapter and primary source.

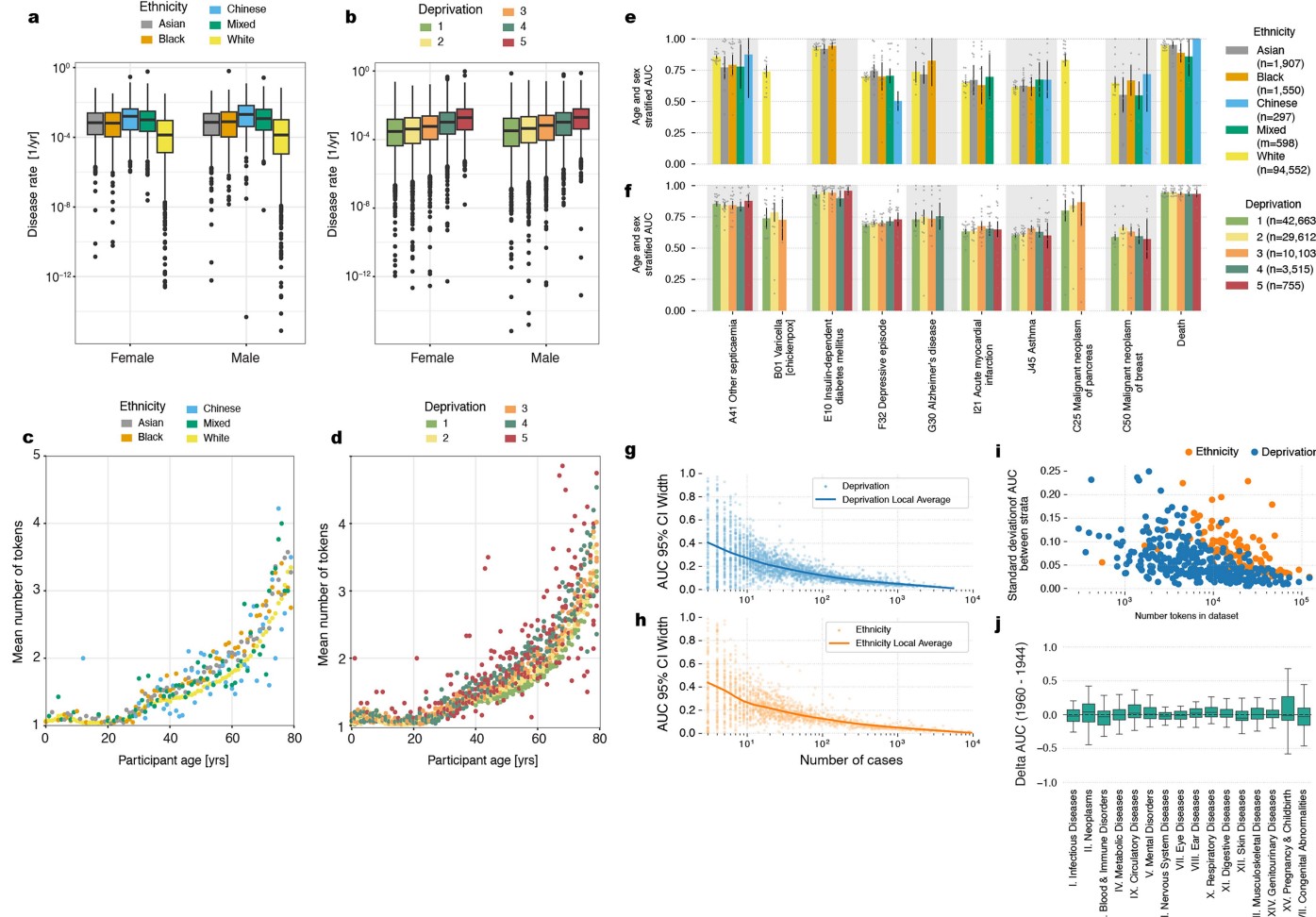

**Extended Data Fig. 10 | Effects of ethnicity and deprivation. a**, Modelled rate per year separated by sex and ethnic background. **b**, Modelled rate per year separated by sex and Townsend deprivation index bins (increasing for greater deprivation index values). The boxplots in **a** and **b** use the entire validation cohort (n = 100639 individual trajectories) and feature median as the center line, the box from the first to the third quartile, the whiskers for 1.5x IQR and the outliers. **c-d**, Average number of disease tokens per year, shown for different ethnicities (**c**) and deprivation indices (**d**). **e-f**, Age and sex stratified AUCs for 10 selected diseases. AUCs are averaged across 5-year age groups ranging from 40 to 80 years of age. The same average is used as the center for error bars. AUCs for individual age and sex brackets are shown as grey dots. 95% confidence intervals are calculated using DeLong's method. **g-h**, Width of DeLong's 95%

confidence intervals for AUC vs number of cases, shown for different ethnicities and deprivation strata. For rare diseases, AUC estimates have high variance. **i**, Standard deviation between AUC estimates for different strata vs number of cases of this disease for the training dataset. Each dot represents a disease. **j**, Average validation AUC across 5-year age groups ranging from 40 to 80 years of age, aggregated by the corresponding ICD chapters. Difference between average AUCs calculated for participants with birth years before 1944 and after 1960. The boxplots feature the median as the center line, the box from the first to the third quartile and the whiskers for 1.5x IQR, clipped at min/max data points. Shown are data for n = 906 diagnoses for males and n = 957 diagnoses for females for which sufficiently many events were recorded in the validation data to evaluate AUCs.

# Reporting Summary

## Statistics

For all statistical analyses, confirm that the following items are present in the figure legend, table legend, main text, or Methods section.

| n/a | Confirmed | |
|---|---|---|
| ☐ | ☒ | The exact sample size (*n*) for each experimental group/condition, given as a discrete number and unit of measurement |
| ☒ | ☐ | A statement on whether measurements were taken from distinct samples or whether the same sample was measured repeatedly |
| ☐ | ☒ | The statistical test(s) used AND whether they are one- or two-sided<br>*Only common tests should be described solely by name; describe more complex techniques in the Methods section.* |
| ☐ | ☒ | A description of all covariates tested |
| ☐ | ☒ | A description of any assumptions or corrections, such as tests of normality and adjustment for multiple comparisons |
| ☐ | ☒ | A full description of the statistical parameters including central tendency (e.g. means) or other basic estimates (e.g. regression coefficient) AND variation (e.g. standard deviation) or associated estimates of uncertainty (e.g. confidence intervals) |
| ☐ | ☒ | For null hypothesis testing, the test statistic (e.g. *F*, *t*, *r*) with confidence intervals, effect sizes, degrees of freedom and *P* value noted<br>*Give P values as exact values whenever suitable.* |
| ☒ | ☐ | For Bayesian analysis, information on the choice of priors and Markov chain Monte Carlo settings |
| ☒ | ☐ | For hierarchical and complex designs, identification of the appropriate level for tests and full reporting of outcomes |
| ☐ | ☒ | Estimates of effect sizes (e.g. Cohen's *d*, Pearson's *r*), indicating how they were calculated |

*Our web collection on statistics for biologists contains articles on many of the points above.*

## Software and code

Policy information about availability of computer code

| Data collection | No novel data was collected for this study. The information about datasets used is in the "Data" section. |
|---|---|
| Data analysis | Python (3.11.9) with packages:<br>- pytorch (2.3.0)<br>- numpy (1.26.4)<br>- matplotlib (3.8.4)<br>- seaborn (0.13.2)<br>- shap (0.45.1)<br>- pandas (2.2.2)<br>- umap-learn (0.5.6)<br>- scikit-learn (1.4.2)<br>- scikit-survival (0.22.2)<br>- statsmodels (0.14.2)<br>- autoprognosis (0.1.21)<br><br>R (4.2.3) with packages (for plotting):<br>- reshape2 (1.4.4)<br>- ggplot2 (3.5.1)<br>- ggVennDiagram (1.5.2)<br>- ggpubr (0.6.0)<br>- RColorBrewer (1.1-3) |

```
- dplyr (1.1.4)
- probcox (0.0.5)

R (4.4.0) with packages (for method comparison):
- CVrisk (1.1.1)
- glmnet (4.1-8)
- mice (3.16.0)
- preventr (0.10.0)
- RiskScorescvd (0.2.0)
- comorbidity (1.1.0)

UKBDRS -  https://github.com/MelisAnaturk/dementia_risk_score/tree/main
QRISK3(2017) - https://www.qrisk.org/src.php (version QRISK3-2017)

Custom Delphi code: https://github.com/gerstung-lab/delphi (commit 6132df6)
```

For manuscripts utilizing custom algorithms or software that are central to the research but not yet described in published literature, software must be made available to editors and reviewers. We strongly encourage code deposition in a community repository (e.g. GitHub). See the Nature Portfolio guidelines for submitting code & software for further information.

# Data

Policy information about availability of data

All manuscripts must include a data availability statement. This statement should provide the following information, where applicable:
- Accession codes, unique identifiers, or web links for publicly available datasets
- A description of any restrictions on data availability
- For clinical datasets or third party data, please ensure that the statement adheres to our policy

Code for Delphi and accompanying scripts and Jupyter notebooks are available on github.com/gerstung-lab/delphi. The model's checkpoint is available per UK Biobank's controlled access procedures with upload ID 7318.

UK Biobank data are available under restricted access through a procedure described at http://www.ukbiobank.ac.uk/using-the-resource/.

Danish registry data are available for use in secure, dedicated environments via application to the Danish Patient Safety Authority and the Danish Health Data Authority via https://sundhedsdatastyrelsen.dk/da/english/health_data_and_registers/research_services/apply.

# Research involving human participants, their data, or biological material

Policy information about studies with human participants or human data. See also policy information about sex, gender (identity/presentation), and sexual orientation and race, ethnicity and racism.

| | |
|---|---|
| Reporting on sex and gender | We use self-reported sex of participants as recorded in UKB field 31 (indicators for female and male) as an input to the model while training and to evaluate the performance of the model in the sex-stratified manner. |
| Reporting on race, ethnicity, or other socially relevant groupings | To assess model performance in subgroups, but not as a part of the data for model training, the following self-reported information was used:<br>- Ethnic background as available in field 21000 (UKB), participants grouped into 5 level groups (White, Mixed, Asian or Asian British, Black or Black British and Chinese)<br>- Index of multiple deprivation as available in field 26410 (combines information across seven domains including Income, Employment Derivation, Health and Disability, Education Skills and Training, Barriers to Housing and Services, Living Environment, and Crime) |
| Population characteristics | The distribution of the disease tokens over sex and age is shown in Supplementary Figure 12. |
| Recruitment | The was no recruitment for this study, the data was obtained from the UK Biobank and Danish National Patient Registry. We restrict Danish cohort to individuals aged 50-80 on 1st of January 2016, to obtain a similar age range as in the UK Biobank. |
| Ethics oversight | The UK Biobank has received approval from the National Information Governance Board for Health and Social Care and the National Health Service North West Centre for Research Ethics Committee 532 (Ref: 11/NW/0382). All UK Biobank participants gave written informed consent and were free to withdraw at any time. This research was conducted using the UK Biobank Resource under project 49978. All investigations were conducted in accordance with the tenets of the Declaration of Helsinki.<br><br>The use of the Danish National Patient Registry for validation of the UK Biobank results was conducted in compliance with the General Data Protection Regulation of the European Union and the Danish Data Protection Act. The analyses were conducted under the data confidentiality and information security policies of the Danish National Statistical Institute Statistics Denmark. The Danish Act on Ethics Review of Health Research Projects and Health Data Research Projects ("the Committee Act") do not apply to the type of secondary analysis of administrative data reported in this paper. |

Note that full information on the approval of the study protocol must also be provided in the manuscript.

# Field-specific reporting

Please select the one below that is the best fit for your research. If you are not sure, read the appropriate sections before making your selection.

☒ Life sciences      ☐ Behavioural & social sciences      ☐ Ecological, evolutionary & environmental sciences

For a reference copy of the document with all sections, see nature.com/documents/nr-reporting-summary-flat.pdf

# Life sciences study design

All studies must disclose on these points even when the disclosure is negative.

| | |
|---|---|
| Sample size | We used "train-validation-test" split that is standard for machine learning papers.<br><br>The models were trained on UK Biobank data for 402,786 (random 80%) individuals using data from birth until 30th of June 2020. For horizontal validation, data contains the remaining 100,636 (20%) individuals for the same period. Testing (also referred as longitudinal validation) was carried out using data for all individuals still alive by the cutoff date (471,057) and evaluated on disease incidence from 1st of July 2021 to 1st of July 2022.<br><br>To evaluate the model generalisation, we additionally tested it using Danish national patient registry, containing 1,931,630 individuals. |
| Data exclusions | For evaluating the model performance (AUC-ROC, SHAP values), only diseases with more than 25 entries in the UK Biobank validation dataset were used. |
| Replication | We performed 4 technical replicates of model training using different training/validation data splits, with all models performing similarly in terms of disease prediction AUC (Supplementary Figure 1)<br>Additionally, multiple models (n=486) were trained for hyperparameter search, the parameters varied included embedding dimension, context size, number or attention heads, number of transformer layers. Many models showed comparable performance in terms of validation cross-entropy loss, though some of them naturally were better, as finding those was purpose of this experiment. |
| Randomization | We randomly divided UK Biobank dataset to training and validation cohorts, using ratio 4:1. Randomisatition was not required to the Danish data, as it was used as a validation dataset, without splitting it to subgroups. |
| Blinding | Anonymised patient identifiers provided by the UK Biobank were used. We didn't perform any additional blinding for Danish data, as the analysis in paper is not a clinical trial. |

# Reporting for specific materials, systems and methods

We require information from authors about some types of materials, experimental systems and methods used in many studies. Here, indicate whether each material, system or method listed is relevant to your study. If you are not sure if a list item applies to your research, read the appropriate section before selecting a response.

## Materials & experimental systems

| n/a | Involved in the study |
|---|---|
| ☒ | ☐ Antibodies |
| ☒ | ☐ Eukaryotic cell lines |
| ☒ | ☐ Palaeontology and archaeology |
| ☒ | ☐ Animals and other organisms |
| ☒ | ☐ Clinical data |
| ☒ | ☐ Dual use research of concern |
| ☒ | ☐ Plants |

## Methods

| n/a | Involved in the study |
|---|---|
| ☒ | ☐ ChIP-seq |
| ☒ | ☐ Flow cytometry |
| ☒ | ☐ MRI-based neuroimaging |

# Plants

| | |
|---|---|
| Seed stocks | *Report on the source of all seed stocks or other plant material used. If applicable, state the seed stock centre and catalogue number. If plant specimens were collected from the field, describe the collection location, date and sampling procedures.* |
| Novel plant genotypes | *Describe the methods by which all novel plant genotypes were produced. This includes those generated by transgenic approaches, gene editing, chemical/radiation-based mutagenesis and hybridization. For transgenic lines, describe the transformation method, the number of independent lines analyzed and the generation upon which experiments were performed. For gene-edited lines, describe the editor used, the endogenous sequence targeted for editing, the targeting guide RNA sequence (if applicable) and how the editor was applied.* |
| Authentication | *Describe any authentication procedures for each seed stock used or novel genotype generated. Describe any experiments used to assess the effect of a mutation and, where applicable, how potential secondary effects (e.g. second site T-DNA insertions, mosiacism, off-target gene editing) were examined.* |

