## [Peer Review File · Nature]

Learning the natural history of human disease with generative transformers

Corresponding Author: Professor Moritz Gerstung

Version 0:

Reviewer comments:

Referee #1

(Remarks to the Author)

This study modified a well-established transformer-based neural network architecture – GPT to enable it to model the temporal time-to-event progression of human diseases. The modified GPT architecture can learn the lifetime health trajectories of individuals for more than 1000 diseases simultaneously. The UK Biobank data was used to train the model, and the Danish registries of Denmark were used for external validation. The authors compared the proposed model with an epidemiological baseline using the sex and age-stratified incidence based on the Nelson-Aalen estimator. The evolution results show that the proposed model outperformed the baseline in predicting the next onset disease and in predicting the further trajectory of multiple diseases.

Pros:

1. There is an increasing interest in applying transformer-based neural network models for healthcare tasks. This study applies transformer-based neural networks for disease trajectory prediction, which is of interest of the healthcare research community.
2. This study modified the well-established GPT architecture to upgrade the “position embedding” into a time-to-event structure, which enables the transformer architecture to model the time-to-event progress of healthcare events. The default “position embedding” only can model the sequence, e.g., before/after of words, but cannot precisely quantify the time from a token to the next token. This methodology improvement is novel for health care.
3. The model was trained using UK Biobank data and externally validated using a Denmark dataset, which was collected from a different source, to ensure generalizability.
4. Detailed analysis of potential gender and racial biases in the UK biobank data and examined the impact of the prediction on the Denmark dataset.
5. Compared with the previous models, the proposed method can simultaneously predict the trajectory for thousands of diseases up to 20 years, which is novel. There are previous work applying transformer-model for next disease predictions, but previous models can not quantify the time-to-event trajectory.

Cons:

1. The UK biobank dataset was split into a training set (80%) and a validation set (20%) for training, which is not consistent with the best practice in training transformer-based neural network models. Typically, the training of transformer-based models uses a training set, a validation set, and a test set. The training set is used to optimize the parameters, the validation set is used to monitor the training loss and determine the best mode (i.e., validation step) – the model achieves the best evaluation scores on the validation set will be selected as “the best model”. The “best model” will be applied to the test set to calculate the final evaluation scores (i.e., the test step). This study only has a training set and a validation set, it's not clear how the best model was selected, or if the scores reported for UK Biobank are from the ‘validation’ step, or from the standard ‘test’ step.
2. The epidemiological baseline using the sex and age-stratified incidence based on the Nelson-Aalen estimator is relatively naïve, compared with the complex transformer models with hundreds of millions of parameters. There have been many studies examining transformer models for disease prediction using traditional machine-learning models and transformer-based models. Though these models cannot model multi-disease trajectories, they can predict the next onset of diseases,

which could be compared with the proposed model.

3. The clinical significance is not well justified. Physicians need AI assistance in decision support of a specific disease other than prediction trajectory in 20 years. I didn't see how the proposed system could help healthcare. In terms of performance, much better performance for specific next-disease onset prediction has been reported using traditional and transformer-based machine learning models. The advantage of the proposed model is that it can simultaneously predict the trajectory of thousands of diseases but might not be as good as previous models focusing on the onset of the next specific disease. The significance of the proposed model to clinical care is not clear.

Minor:

"The analogy between *large language* and disease progression modeling" -> large language model?

(Remarks on code availability)

I didn't test the code, but the codes are modified from a widely used open-source package. The readme file contains necessary instructions.

Referee #2

(Remarks to the Author)

The authors modify a GPT architecture by modelling the temporal progression of human disease (ensuring that the model only accesses information from past events) to predict rates of 1,000+ different ICD-10 coded diseases and death, conditional on each individual's past disease history, age, sex and baseline lifestyle information. The training is done on part of the UK Biobank data and validated on a disjoint component of the data. The model is furthermore validated on a Danish cohort without change in its parameters.

There are several major concerns about the work, particularly with respect to similar literature on the subject.

Major comments:

Lack of comparison with other LLM models: (i) The new architecture is derived by modification of the GPT-2 model. Yet, GPT-4 already exists and it is unclear how it would perform in comparison. There is already a publication on UK Biobank data <https://doi.org/10.1016/j.isci.2024.109022> with GPT-4 on CVD. In fact, the latter deals with a potentially more difficult task of training on UKB data and prediction on a non-European cohort. The performance of Delphi-2M needs to be compared to this architecture (on CVD) in detail to show it is competitive.

(ii) AutoPrognosis 2.0 <https://doi.org/10.1371/journal.pdig.0000276>, which has been applied to UKB data for prognostic risk prediction of diabetes, but "can be applied to construct diagnostic and prognostic models for any disease or clinical outcome, and is explicitly designed to make model building accessible to both experts and non-ML experts." The performance of Delphi-2M needs to be compared to this model (e.g. on CVD).

(iii) The work should also relate to a new advancement building on AutoPrognosis 2.0, <https://arxiv.org/abs/2310.03560> (v3), to "explore a new application of LLMs in healthcare and propose using LLMs to facilitate clinician interactions with AI models and digital tools." to limit "potential for hallucinations" of LLM models and "ensure actionable information is provided by approved clinical sources".

(iv) Aside from the technological advancement in (iii), there are several metrics therein that should be included in the manuscript for comparison. For example, there are several risk scores for CVD, Framingham score (in the United States), SCORE2 (in Europe), and QRisk3 (in the UK). There is also a more recent study by the American Heart Association <https://doi.org/10.1161/CIRCULATIONAHA.123.067626>, to assess the PREVENT score for CVD. These CVD scores need to be compared with the model (not only Framingham) as the performances seem to differ depending on the score used. The manuscript is expected to present Tables similar to Table 1 in <https://arxiv.org/abs/2310.03560> with additional scores for model discrimination and calibration.

(v) The imputation strategy in <https://arxiv.org/abs/2310.03560> for CVD, uses state of the art methods including MICE and Multiple imputation, as opposed to a simple mean imputation employed by the authors. To what degree this affects the results is unknown and should be investigated.

Therefore, before aiming to predict multiple outcomes, it is important to compare Delphi-2M's model performance, in depth, on a disease such as CVD, relative to other models.

The manuscript lacks in providing insights into scenarios where the model performance is better or worse or unchanged, e.g., (i) does prediction performance differ, at a fixed number of years X, between individuals born earlier (1940) vs later (1960); how does the prediction performance change for every extra year into the future; does the prediction accuracy differ amongst individuals with different ethnicities and/or deprivation metrics in UKB?

Minor comments:

- Certain sentences in the manuscript are left vague without providing succinct evidence. For example, in the introduction it is stated "The analyses not only reveal that Delphi-2M provides accurate instantaneous multistate rates ...", "accurate" here is a qualitative statement and should instead be immediately backed up by quantitative evidence. Similarly in "The predictions remain accurate even if evaluated on an external dataset from Denmark without any additional fine tuning" the word "remain accurate" is sloppy and not true according to the authors, since later on in the discussion it is stated that "[Delphi-2M has] reduced accuracy on Danish registries ...". "However, a key distinguishing feature of Delphi compared to

basic GPT models is its ability to calculate the absolute rates of tokens, which in a statistically accurate sense represent inter-event times”, what does “statistically accurate sense” mean quantitatively?

- Figure captions are poorly explained. See for example, Figure 1, panels (e), (f) and (g) where very little is said about what is going on in the figure, or the main messages for the reader. Same with Extended Figure 1. Similarly for Figure 2, there is a lot going on in panel (a) and the caption provides no information, e.g. what is the black line “selected case”, what does the larger black circle represent? All figure captions need to be drastically improved.

- Inclusion of “Overall health rating UKB data field 2178” in the model and comparison performance with and without. This has been demonstrated to improve prediction in the literature.

- UMAP is well known to distort distances. The representation and trajectory statements in Extended Fig 7 is misguided and should be removed, e.g., “trajectory traverses the labelled diagnosis in close proximity”.

(Remarks on code availability)

I have not ran the code, but the GitHub page seems to have the relevant instruction and scripts available for the user. Jupyter notebooks are also made available.

Referee #3

(Remarks to the Author)

In this paper, the authors introduce a transformer-based foundation model named Delphi, trained on UK Biobank data to predict future disease rates across a broad spectrum of diseases. By adapting the tokenization technique used in the GPT-2 model to their specific dataset, and adding an additional predictive head for timing the next token alongside event classification, the model demonstrates a novel approach to disease prediction.

After training, Delphi underwent longitudinal validation using both the UK Biobank and an external cohort from Denmark. The results show well-calibrated disease rates stratified by age, and the performance of Delphi’s predictions is comparable to established clinical risk models, such as the Framingham score for cardiovascular diseases. The validity of the model is further supported by UMAP visualisations of disease code embeddings in Delphi-2M, which cluster effectively by the underlying disease category, and SHAP score analysis that highlights comorbidities. The paper also conscientiously addresses potential biases such as recruitment, immortality, and data availability.

The application of advanced transformer-based techniques in predicting patients’ disease risk across hundreds of diseases based on their medical history, represents a significant innovation in medical AI. The comprehensive validation across multiple cohorts, along with comparative analysis against established clinical models, lends considerable credibility and potential impact to the findings presented in this paper. I have identified several areas that require further attention:

1. Additional Validation of Delphi's Predictions:

- a. The authors provide calibration of risk predictions stratified by age and sex across the UK Biobank and the external Danish cohort. However, further stratification analyses based on other model input features, such as smoking status, body mass index, and alcohol consumption, are necessary to fully assess calibration and to evaluate remaining inter-person variability after accounting for these variables.
- b. Similar stratification analyses are also required for a more comprehensive validation of simulated patient trajectory analyses from Delphi.
- c. Extending the comparison of Delphi’s risk assessment with classical clinical scores and disease risk predictors, such as the Framingham score for CVD, UKBDRS score for dementia, and the Charlson score for death is important. The authors could also compare against quantitative biomarkers and clinical scores such as A1c for diabetes, mean corpuscular volume for anemia, and liver function markers for chronic liver disease. Such comparisons, even if Delphi’s predictors cannot outperform these models, are essential to understand how LLM-based predictions based on medical history compare with established markers.

2. Further Validation of SHAP Values: The authors utilized SHAP values to explore how individual tokens influence future risks. At present, it is unclear how these SHAP values relate to the average rate increases across disease tokens as presented in Figure 4. A systematic comparison of SHAP values with empirical rate changes is required to fully validate the interpretability of the model.

3. Explainable AI: "Of note such patterns often appeared symmetrical, indicating similar predicted effect sizes of one disease token influencing another as vice versa (Extended Data Figure 8)." I would love for the authors to explain this slightly more either in paragraph or in the discussion, to caution readers against immediately using these results to infer causality.

4. "No other event" padding: I think the readers would benefit from the authors discussing the implication of using these paddings a bit more. If the authors have performed tests leaving out these "no other event" paddings it might be informative to the readers if the authors included them in the supplementary materials for discussions on how these paddings allow the model to (a) predict trajectories at age X when there are no previous disease codes and (b) update attention weights and trajectory predictions after disease events lapse for a range of periods with no further events.

5. Different types of missing data: As the authors wrote, 78% of participants have self-reported disease data, 45% have GP data and 86% have hospital records, and missing data is due to a combination of data acquisition, UKbiobank being a retrospective cohort, and healthy volunteer-bias, etc. I wonder if some of these (esp the large contributions of missingness eg. lack of GP data) could be modelled as more types of "no event" paddings and if that would be of value in the Delphi model. I think some discussions along these lines would be interesting.

6. Differences in performance by demographic group: The authors wrote and demonstrated in Extended Data Figure 11 that the prediction is worse in White individuals because "is likely a consequence of participants of "White" ethnic background having lower numbers of disease tokens across time, which reduces the ability to predict future disease burden", and that is echoed by the results from stratification by Townsend deprivation index, where those with lower Townsend deprivation index who have lower numbers of tokens were predicted worst. Is this likely a property only of UKB which has a healthy volunteer bias? In EHRs, would it not likely be the other way round, where those with higher deprivation are likely to have lower access to healthcare and therefore lower tokens? How might Delphi2M, if applied on EHRs, perform differently, and how might this issue of difference in number of tokens across different demographics be solved such that usage of Delphi2M does not incur issues of healthcare disparities?

7. Discussion of Practical Implications: While the paper aims to model disease progression to inform healthcare planning, it lacks a detailed quantitative analysis of how these predictions could benefit healthcare systems. For instance, how might these predictions be integrated with quantitative biomarkers or clinical scores to enhance predictive accuracy? Should they be used to identify patients at risk for specialised testing? The authors need to expand on the discussion of practical implications of their findings with a robust quantitative assessment to demonstrate the utility of these predictions in specific settings. This is particularly important to strengthen the impact of this work and ensure its suitability for publication in Nature.

(Remarks on code availability)

Referee #4

(Remarks to the Author)

Nature is committed to facilitate training in peer-review and to ensure that everyone involved in our peer review process is appropriately recognised. This reviewer co-reviewed one of the listed reports.

(Remarks on code availability)

Version 1:

Reviewer comments:

Referee #1

(Remarks to the Author)

My major comments in the 1st round review include (1) Clarifying the training/validation/test data split; (2) Comparing the proposed method with a strong baseline; (3) strengthening the clinical importance.

Comment #1 is solved by clarifying that the UK biobank data before July 2020 was used for training and validation, and two years' data after July 2020 was used as an independent test set. I have no concerns about the rigor of evaluation.

Comment #2: New experiments were conducted to compare the proposed model with a strong baseline, MILTRON, a very recent model published in Nature Genetics. I'm confident that the new comparison results with a strong model demonstrated the advantage of the proposed methods.

Comment #3. For comment #3, in addition to the revised discussion content, I'd recommend the authors strengthen the importance from the point of view of healthcare IT deployment. A critical contribution of this study is that this new model provided a foundation to "deploy one AI system to provide decision support for many disease domains". Previous generations of non-transformer-based AI models could be customized through extensive human engineering to achieve similar performance for specific diseases, yet, the healthcare systems have to deploy hundreds of customized models for different diseases, which is a huge burden for healthcare IT.

Overall, the new revision has addressed most of my comments. This study has methodology contributions in repurposing the "position" embedding architecture for a new use - to model the time-to-event sequence in EHRs. There have been too many ChatGPT-style studies that simply throw everything to ChatGPT and play with prompts. This is a good example of adopting state-of-the-art transformer models for healthcare problems. I have no further concerns about this study.

(Remarks on code availability)

Referee #2

(Remarks to the Author)

The authors have provided additional analyses for the majority of the points raised in the review, this has improved the manuscript. Regarding the upgrade to GPT4 architecture, I appreciate the authors' response that this seems not possible due to the GPT4 not being available open source and UKB regulations.

Given the new analysis results, I have a major concern regarding the relevance of this approach for clinical applications, or for generation of biomedical insight.

The authors need to be upfront, in the abstract and main body of the text, regarding Delphi's performance relative to simpler and transparent statistical models. Specifically, Delphi has approximately the same performance as QRISK3 (a widely used cardiovascular disease assessment tool in the UK and internationally) for CVD, noting that QRISK3 is based on one of the simplest parametric models for survival analysis, namely Cox proportional hazards regression, and fractional polynomials. The reason it is important to be upfront about this to the scientific community is to avoid the risk of over-optimistic interpretation of the use of complex AI/ML models (here, a generative transformer model) for clinical applications in practice when their performance is no better than simple and transparent expert-developed scores or simple statistical models. Of note, the features that QRISK3 takes as input are routinely available as part of EHRs. This argument also holds for other diseases explored, e.g. dementia (UKBDRS, a lasso model) and CLD (where a logistic model seems to have been used as alternative). For diabetes, in fact a single marker (HbA1c, AUC 0.84) outperforms Delphi (AUC 0.75). This is an important point that should not be deferred to the Discussion section.

As a scientific exercise, it is indeed valuable to see how generative transformer models would perform against current clinical metrics. In fact, it is very interesting to see that expert-derived clinical scores, when available and optimised, have a similar performance to a complex model of this type. However, this then begs the question, both from a clinical perspective and with respect to novel biomedical insights, why train a generative transformer model given there is often no gain in prediction accuracy relative to clinical scores (other than Death), whilst there is a loss in simplicity and interpretability?

Major comments:

The following sentence is misleading "Delphi-2M performs equally, or better than existing risk scores, machine learning models and LLMs for cardiovascular disease, dementia and death". This needs to be stratified as follows: The performance of Delphi is similar to routinely used clinical risk scores for CVD and dementia, and better than those used for death. For diabetes, the performance of Delphi is worse than a single marker (HbA1c) clinically used for risk prediction and diagnosis of diabetes.

Extended Data Figure 18 (f) and (g). It is important to have a description of the observations on deprivations and ethnicity subgroups in the main text so that it does not remain "hidden" in the appendix. In particular, there are many occasions where the lower end of the whiskers hits below AUC 0.5, particularly for deprivation level 5 (12/16) and black ethnicity (7/16). This means that, in these cases, Delphi-2M performs worse than guessing at random.

Can the authors delve into why certain ICD10/ethnicity/deprivation combinations lead to good performance whereas others lead to poor performance as reflected by AUC? How does it depend on the type of ICD10 code, number of cases, ratio of cases to controls, and relevance of predictors for the disease under consideration?

Extended Figure 6(a). It would be informative to add a figure per chapter (similar to Extended 6a), showing the percentage of conditions in the chapter that are below AUC 0.5 over time (with the same x-axis as this figure) and repeat for AUC 0.6, 0.7, 0.8 and 0.9. The idea is to show how the proportions of AUC shift as time passes, i.e., how Delphi-2M's performance changes when looking further into the future.

The comparison of the predictions of Delphi's performance with overall health rating as baseline is indeed useful to see and interesting. I appreciate the authors' comments on exploring inclusions of blood biomarkers in a second development of the method. However, I do not think overall health rating can be considered on the same footing as blood biomarkers or environmental exposures. It would be interesting to know if the addition of this field would improve the lower bound of AUCs obtained in demographics such as those with high deprivation and ethnic minorities. That said, I appreciate that this exact field may not be transferable to the Danish cohort.

Minor remarks:

Extended Data Figure 8a (middle) the abbreviation MCV is not defined (mean corpuscular volume).

Page 3, caption with "XX" occurrences.

(Remarks on code availability)
As the first round.

Referee #3

(Remarks to the Author)

I have carefully looked through the authors responses to my own request and the other reviewers' requests. The authors have made a substantial effort addressing all reviewers' concerns, and have added substantial analyses and further comparisons to their previous submission.

However, I was expecting a more in-depth response to my last point regarding the practical application of Delphi-2M. The claim that Delphi-2M is a "proof-of-concept that a single generative model can be learned for the entire corpus of human disease with competitive performance as single-disease models" is not particularly compelling. With appropriate

engineering, such a model can indeed be trained—but for which specific application are these results a proof of concept? Addressing this question in greater depth is crucial for demonstrating the relevance and impact required for this venue.

Specifically:

The discussion on long-term trajectories remains highly speculative and would be extremely difficult to validate.

The statement that Delphi-2M has applications in "clinical, pharmacological, and epidemiological domains" is too broad and would benefit from more concrete examples.

Instead, I find two aspects highlighted by the authors particularly promising, but I believe they require further validation:

Multi-disease predictions – This is an interesting feature, with primary application in prevention and monitoring. In this context, a compelling proof of concept would be to compare Delphi-2M with Polygenic Risk Scores (PRS) and ideally demonstrate that Delphi-2M + PRS outperforms PRS alone.

Integration of quantitative biomarkers – The authors note that "Delphi may be easily extended to incorporate additional types of health data such as laboratory values," which is a promising direction. Here, a proof of concept would be to demonstrate that Delphi-2M + quantitative biomarker improves predictive accuracy over the biomarker alone. This could be tested, for instance, by inputting patient trajectories up to the point when biomarkers are measured and assessing the added predictive value of Delphi-2M in this context.

In summary, I feel that strengthening the connection between current results and potential applications would be crucial in this setting.

I also encourage the authors to add more of their responses to reviewers into their main text or methods. For example, their explanations on the impact of rate of no-event paddings given in the response to reviewers may be very interesting to other groups developing similar methods or wanting to apply Delphi to their own datasets.

Further I would recommend the authors to provide more documentation on their github page. Currently ipython notebooks are provided without a little/no explanation or comment. It would be great if the authors can increase the level of instruction.

(Remarks on code availability)

The authors have provided a github page that would benefit from more detailed documentation on the training and the downstream analyses. For example "evaluate_delphi.ipynb" does not even have any comments in the code - it would be helpful to have some level of comments/instructions. I have not tested the code myself.

Referee #4

(Remarks to the Author)

I co-reviewed this manuscript with one of the reviewers who provided the listed reports.

(Remarks on code availability)

Comments included in the main referee report

Version 2:

Reviewer comments:

Referee #1

(Remarks to the Author)

The revision has solved all my comments, and I have no more concerns. This paper is in good shape.

(Remarks on code availability)

I have reviewed and run the code. The code will be a useful resource.

Referee #2

(Remarks to the Author)

The manuscript has clearly improved and now contains additional helpful insights with regards to prediction using disease history alone vs biomarker alone, vs combined, as well as PRS. The authors have also performed a deeper analysis on demographic differences and deprivation on prediction accuracy and have clarified the findings. Crucially, I am satisfied that the authors are now upfront about when the performance is better/worse relative to other approaches and/or biomarker features in the main body of the text and the discussion. I recommend the manuscript for publication.

(Remarks on code availability)

As before.

Referee #3

(Remarks to the Author)

The authors have addressed all my questions and I recommend publication.

(Remarks on code availability)

Referee #4

(Remarks to the Author)

I co-reviewed this manuscript with one of the reviewers who provided the listed reports.

(Remarks on code availability)

I haven't tested or run the code, but on review, it appears well-structured and clearly documented.

Summary of changes

We would like to thank all reviewers for their critical and helpful comments, which we have thoroughly addressed as detailed in the next section. In summary, the following changes were made:

- As requested by reviewers 1 and 2 we have carried out a **range of additional comparisons** ranging from methodologically basic established risk scores to advanced methods including transformer models and an LLM (LLaMa 3.1). As these methods apply to selected diseases, we also compare Delphi to MILTON, a recent blood biomarker based model for the entire ICD corpus. These comparisons, now shown in **Figure 2f,g** and **Extended Data Figure 7** reveal that Delphi's performance is matching or even surpassing other methods for a wide range of diagnoses.
- As suggested by reviewers 2 and 3, we have also further **expanded the quantitative assessment of a range of Delphi's properties** which help understand how the transformer model functions, how it performs over time, and how its modelled effects relate to established statistical models. These results are shown in **Figure 2e** and **Extended Data Figures 6, 12, 14**.
- We are showing additional assessment of **Delphi's performance for different subgroups**, as requested by reviewers 2 and 3. Specifically, we calculated age-and-sex-adjusted AUCs for persons born before 1940 and those born after 1960, for different behaviour (smoking, alcohol) and also for ethnicity and deprivation. These results are shown in **Extended Data Figures 10, 18 and 19** and support our previous claims that Delphi also performs well in different demographics.
- As requested by reviewers 1 and 3, we **discuss in greater detail what types of medical applications for Delphi-like models exist and how the road to implementation could look like**. As we stress throughout the manuscript, Delphi-2M is a proof-of-concept model – and according to our knowledge, the first – which demonstrates that health courses can be comprehensively predicted and simulated by exploiting generative AI. This prototype exhibits competitive performance (underscored by the additional comparisons) still exhibits a range of shortcomings, which we have discussed in detail. In **revised Discussion**, we are now laying out the path what type of benefit Delphi-type models may bring to patients and also healthcare providers through forecasting of future health events.
- In response to all individual comments, we have also implemented a range of changes that led to the inclusion of **9 additional Extended Data Figures and 2 Tables**. In addition to including new results requested by the reviewers we have also **edited Figures 2, 3, 5 and 6 for additional clarity**.

We believe that these additional analyses have greatly strengthened the manuscript and confirm our previous claims. In the following, you will find the detailed point-by-point replies.

Referees' comments:

Referee #1 (Remarks to the Author):

This study modified a well-established transformer-based neural network architecture – GPT to enable it to model the temporal time-to-event progression of human diseases. The modified GPT architecture can learn the lifetime health trajectories of individuals for more than 1000 diseases simultaneously. The UK Biobank data was used to train the model, and the Danish registries of Denmark were used for external validation. The authors compared the proposed model with an epidemiological baseline using the sex and age-stratified incidence based on the Nelson-Aalen estimator. The evolution results show that the proposed model outperformed the baseline in predicting the next onset disease and in predicting the further trajectory of multiple diseases.

Pros:

1. There is an increasing interest in applying transformer-based neural network models for healthcare tasks. This study applies transformer-based neural networks for disease trajectory prediction, which is of interest to the healthcare research community.
2. This study modified the well-established GPT architecture to upgrade the “position embedding” into a time-to-event structure, which enables the transformer architecture to model the time-to-event progress of healthcare events. The default “position embedding” only can model the sequence, e.g., before/after of words, but cannot precisely quantify the time from a token to the next token. This methodology improvement is novel for health care.
3. The model was trained using UK Biobank data and externally validated using a Denmark dataset, which was collected from a different source, to ensure generalizability.
4. Detailed analysis of potential gender and racial biases in the UK biobank data and examined the impact of the prediction on the Demark dataset.
5. Compared with the previous models, the proposed method can simultaneously predict the trajectory for thousands of diseases up to 20 years, which is novel. There is previous work applying transformer-model for next disease predictions, but previous models can not quantify the time-to-event trajectory.

We have amended the manuscript to clarify aspects of the training, validation, and testing setup, as well as to better present the general objectives of our study. Additionally, we have included comparisons for a handful of diseases to demonstrate our approach's performance against some clinical scores and other machine learning models.

Below is a detailed point-by-point response. We thank the reviewer for their time and consideration and hope that we have adequately addressed the raised points.

Cons:

1. The UK biobank dataset was split into a training set (80%) and a validation set (20%) for training, which is not consistent with the best practice in training transformer-based neural network models. Typically, the training of transformer-based models uses a training set, a validation set, and a test set. The training set is used to optimize the parameters, the validation set is used to monitor the training loss and determine the best mode (i.e., validation step) – the model achieves the best

evaluation scores on the validation set will be selected as “the best model”. The “best model” will be applied to the test set to calculate the final evaluation scores (i.e., the test step). This study only has a training set and a validation set, it’s not clear how the best model was selected, or if the scores reported for UK Biobank are from the ‘validation’ step, or from the standard ‘test’ step.

The reviewer raises an important point. Indeed, the initial data split for the UK Biobank consists only of a training and validation set. However, we do have a dedicated test set in the UK Biobank, specifically a temporal split. Data up to July 2020 were used for model training and validation, while the remaining two years are solely for evaluating the model (Extended Data Figure 7). We opted for an evaluation based on this temporal split as it aligns more closely with the intended use cases and tests model generalisation, which should be challenging given the potential for evolving trends over time—particularly during the COVID-19 pandemic.

Additionally, we conducted an external evaluation using Danish data, which involves new individuals in an entirely different healthcare setting. Notably, given the similarity of the results between the UK Biobank test and the Danish test evaluations, as well as the direct comparisons between them, we decided to also present some results from the UK Biobank validation data, as this helps us better discern age effects. To further ensure that the model doesn’t have “validation overfitting” to the particular split used, we trained additional model replicates on different non-intersecting splits of the same size (80%-20%). All the models showed consistent performance in respective validation cohorts. (Extended Data Figure 3). We didn’t perform any additional hyperparameter tuning for this experiment and trained the models for a fixed number of iterations (n=200,000) without using early stopping.

Extended Data Figure 3. AUCs for model replicates. Average validation AUC across 5 year age groups ranging from 40 to 80 years of age as a function of training occurrences. Shown are data for n=733 diagnoses for males and n=803 diagnoses for females with at least 1 age bin with XX occurrences, aggregated by the corresponding ICD chapters. First in each group, shaded, are results for the original Delphi-2M model, followed by 3 technical replicates, trained on different non-intersecting train-validation data splits.

Nevertheless, the reviewer is correct in noting that our presentation is occasionally ambiguous. We have modified our manuscript to clarify the use of different dataset splits:

p.4, l.31: "Training data comprised 400,000 (80%) participants of the UK Biobank recorded before the 1st of July 2020. Data for the remaining 102,485 (20%) participants were used for validation and hyperparameter optimisation, while all records for 471,057 (94%) participants still alive on the 1st of July 2020 were used for longitudinal testing up until the 1st of July 2022 (Figure 1b). Additional external testing was carried out on the Danish disease registry data covering 1.93M Danish nationals aged 50-80 on 1. Jan 2016, with data available between 1978 and 2018."

We have also modified the manuscript in all other places when the clarification was applicable, including figures, figure captions and methods. We hope this clarification addresses the reviewer's concerns. Thank you once again for bringing this to our attention.

Actions:

- Clarified the structure of the dataset in the main text, figures, figure captions and methods
- Tested the performance with alternative dataset splits
- Included the results as Extended Data Figure 3

2. The epidemiological baseline using the sex and age-stratified incidence based on the Nelson-Aalen estimator is relatively naïve, compared with the complex transformer models with hundreds of millions of parameters. There have been many studies examining transformer models for disease prediction using traditional machine-learning models and transformer-based models. Though these models cannot model multi-disease trajectories, they can predict the next onset of diseases, which could be compared with the proposed model.

We fully agree with the reviewer. Proper validation is essential for the credibility of the field and we appreciate the reviewer for highlighting this point and encouraging us to perform a stronger comparison. The Nelson-Aalen baseline is only the minimum a model needs to achieve. We have chosen it as it is a statistically principled approach that can be implemented for the entire disease spectrum without making additional assumptions.

In order to provide a comparison against more advanced models we have expanded our evaluation. New comparisons are included for dementia, death, and cardiovascular disease (CVD) featuring several clinically implemented scores, such as Prevent and Qrisk. We have also tested an automated machine learning implementation (AutoPrognosis), and a dedicated transformer model trained on the same data as Delphi (Figure 2f, Extended Data Figure 7).

The results indicate that Delphi's risk estimation is better or comparable to that of all the competitors. Notably, this holds even for scores with access to additional relevant information, such as cholesterol level and blood pressure for Qrisk3 and Framingham.

We have also added a more granular comparison against MILTON - a recently published machine learning approach that uses biomarkers to predict the whole spectrum of UKB diseases. The evaluation indicates that Delphi outperforms MILTON in terms of AUC for most of the diseases (Figure 2g).

We hope the newly added comparisons address the reviewer's concerns.

Figure 2. Delphi-2M accurately models the rates of a wide range of diseases. (subset)

f, AUC-ROC curves for Delphi and other clinical or machine learning methods for three selected endpoints evaluated on the internal longitudinal testing set. **g**, AUCs of MILTON, a biomarker-based machine learning model (x-axis) in prognostic mode compared to Delphi AUC values from the horizontal validation set (y-axis) for n=410 diagnoses.

Actions:

- **Performed additional comparisons, which now include:**
 - **Death: Elixhauser Comorbidity Index, Charlson Comorbidity Index, Transformer-based model, Llama 3.1 8B LLM**
 - **Dementia: UKBDRS, Transformer, Llama 3.1 8B**
 - **CVD: Qrisk3, Prevent, Score2, Framingham, Transformer, Llama 3.1 8B, AutoPrognosis**
 - **MILTON, a recent machine learning algorithm that utilises 67 different biomarkers to predict a wide spectrum of ICD diagnoses.**
- **Updated figures (Figure 2, Extended Data Figure 7) with the new comparison results and modified Methods accordingly**
- **Included the additional comparison details as Extended Data Tables 2, 3**

3. The clinical significance is not well justified. Physicians need AI assistance in decision support of a specific disease other than prediction trajectory in 20 years. I didn't see how the proposed system could help healthcare. In terms of performance, much better performance for specific next-disease onset prediction has been reported using traditional and transformer-based machine learning models. The advantage of the proposed model is that it can simultaneously predict the trajectory of thousands of diseases but might not be as good as previous models focusing on the onset of the next specific disease. The significance of the proposed model to clinical care is not clear.

Thank you and reviewer 3, who made a similar comment, for raising an important point, which we haven't discussed sufficiently. At this stage, Delphi constitutes a proof-of-concept

that a single generative model can be used for learning the entire corpus of human disease with competitive performance as single-disease models.

Delphi-2M's performance is further underscored by our additional comparisons against a number of established methods, as outlined above. Since some of these scores are used for medical purposes, there is potential for Delphi's use as a disease-specific decision support system. One advantage of multi-disease predictions is that Delphi can provide information for all disease-specific outcomes that a clinician might be interested in, rather than being limited to a specific implementation. This is likely to provide great value when evaluating potential interventions. However, we also caution that even though it appears likely that Delphi (or models inspired by it) lead to a range of applications ranging from individual risk assessment, therapeutic decision support or health care planning, this is currently not proven as each of these applications warrants in-depth and ideally prospective evaluation. We now emphasise these points more strongly in the Discussion (see below)

Additional utility for healthcare planning may be derived from Delphi's generative nature. While long term trajectories may not be necessary for imminent clinical decision making, as the reviewer notes, they can help obtain better long-term predictions. An interesting application for these trajectories could be in modelling future health burdens in various population subsets, using the model to project the future comorbidity landscape. There is some preliminary evidence that our model could be useful for this task (Figure 3c), but a proper evaluation and comparison with projections from e.g. the Global Burden of Disease study would be an intriguing follow-up, highlighting its use in more epidemiological endeavours.

Lastly, Delphi establishes a modelling framework for electronic health record (EHR) data that provides a foundation for many important health tasks, ranging from clinical to pharmacological and epidemiological applications. As we discuss in the manuscript, Delphi may be easily extended to incorporate additional types of health data such as laboratory values which are likely to further improve its accuracy and utility.

Taken together, we appreciate the reviewer highlighting this point, as we did not adequately convey these various aspects in the manuscript. We have revised the manuscript and expanded our discussion to emphasise the aforementioned points. We thank the reviewer for this very important point.

Actions:

- **We discuss the proof-of-concept nature, potential use cases and road to implementation in the Discussion:**

p. 38, l. 46: *“Currently, Delphi-2M constitutes a proof of concept that AI architectures originally developed for natural language processing can be extended to model the temporal sequences of complex healthcare events. Notwithstanding Delphi-2M's current limitations, its encouraging performance relative to other methods and its transferability between UK and Danish healthcare systems indicated there will be clinical utility of Delphi or similar tools in the future.”*

An evident application of Delphi-type models would be to support medical decision-making by the probability distribution of future trajectories for a patient. A direct use could be to identify individuals who meet established risk thresholds, e.g. by age or biomarkers, that warrant inclusion in screening programs or further prevention measures. As discussed above, it would be critical to assess and mitigate biases present in the training set before any such system is tested in a clinical setting. Even with a relatively unbiased model, the benefits of such risk stratification must first be assessed through randomised controlled trials before broader implementation. Additionally, deploying clinical decision support systems requires a regulatory framework, which is still in its infancy for AI in healthcare. A long-term objective for AI-based clinical decision support systems is to provide direct treatment recommendations. Currently, Delphi does not explicitly model the effect of treatments, and usually, it is only possible to estimate these effects reliably using data from randomised clinical trials. Therefore, it is essential to establish frameworks within which clinical trial data can inform AI models, which are in turn, to be evaluated in subsequent randomised studies.

An alternative utility of Delphi models would be to help allocate healthcare resources by identifying the needs for future interventions. At a local level, for example, this could inform care trajectories on admission to a hospital. Such applications, informing the providers rather than treatment decisions, require less regulation. Similarly, there may be substantial system-wide modelling benefits for Delphi type models, in particular in the tailored projection of a cohort using their current healthcare until a point in time. This has the potential to produce better estimates of the future for aggregate healthcare demand, which is useful at regional and national health system levels, as well as for insurers. This will help allocate resources that can be scarce in ageing populations.

These considerations illustrate the wide range of applications of generative models for biomedical research and, ultimately, also for healthcare. With appropriate training and evaluation, future multi-modal model extensions may be used for preventative medicine, clinical decision support and healthcare planning. Our model and analysis presented here present a further step to unlock the considerable healthcare benefits from the era of AI.”

- Added further comparison to clinically established risk models, now shown in Figure 2f and shown below.

Figure 2. Delphi-2M accurately models the rates of a wide range of diseases. (subset)

f, AUC-ROC curves for Delphi and other clinical or machine learning methods for three selected endpoints evaluated on the internal longitudinal testing set. **g**, AUCs of MILTON, a biomarker-based

machine learning model (x-axis) in prognostic mode compared to Delphi AUC values from the horizontal validation set (y-axis) for n=410 diagnoses.

Minor:

“The analogy between *large language* and disease progression modeling” -> large language model?

Yes, this should have been “large language model”, thank you very much for spotting.

Referee #1 (Remarks on code availability):

I didn't test the code, but the codes are modified from a widely used open-source package. The readme file contains necessary instructions.

Thank you for taking time to assess our code. During the revision, we continued to work on the codebase, improved the code readability and documentation, specifically for data preprocessing, added a script for faster AUC evaluation and increased overall performance. We have also added Docker support, which could be helpful for running Delphi in the “out of the box” fashion.

Referee #2 (Remarks to the Author):

The authors modify a GPT architecture by modelling the temporal progression of human disease (ensuring that the model only accesses information from past events) to predict rates of 1,000+ different ICD-10 coded diseases and death, conditional on each individual's past disease history, age, sex and baseline lifestyle information. The training is done on part of the UK Biobank data and validated on a disjoint component of the data. The model is furthermore validated on a Danish cohort without change in its parameters.

We thank the reviewer for this in-depth review of our paper and for the many helpful recommendations to improve the paper. Overall, we expanded the comparison of our approach to CVD with many of the suggestions by the reviewer. A detailed point-by-point reply can be found below.

There are several major concerns about the work, particularly with respect to similar literature on the subject.

Major comments:

Lack of comparison with other LLM models:

(i) The new architecture is derived by modification of the GPT-2 model. Yet, GPT-4 already exists and it is unclear how it would perform in comparison. There is already a publication on UK Biobank data <https://doi.org/10.1016/j.isci.2024.109022> with GPT-4 on CVD. In fact, the latter deals with a potentially more difficult task of training on UKB data and prediction on a non-European cohort. The performance of Delphi-2M needs to be compared to this architecture (on CVD) in detail to show it is competitive.

The reviewer raises an important point. While our base architecture, GPT-2, is not the most recent implementation, it remains one of the last open-source architectures, and the core components (transformer blocks) are consistent across later GPT iterations.

That said, the suggestion to compare with GPT-4 is indeed valid, and we have also considered this. However, GPT-4 and many of its counterparts are accessible only through an API, which would require sending data to OpenAI or other tech companies, thereby violating UK Biobank (UKB) data privacy regulations. We have consulted with UKB, and sending data to third parties is not permissible. While the mentioned publication is intriguing, it is unclear whether they obtained an exemption for their specific data extract or if it constitutes an unintended data violation. Therefore, we regret that we cannot fulfil this request.

Nevertheless, recognising the reviewer's valuable point, we downloaded a local version of the LLaMA 3.1 8B model from Meta AI (its newest version) and evaluated it on prompts containing the same data used for our model. We have added the comparison for cardiovascular disease (CVD), dementia, and death (Figure 2d). We hope the reviewer finds this a reasonable alternative.

Figure 2. Delphi-2M accurately models the rates of a wide range of diseases. (subset)

f, AUC-ROC curves for Delphi and other clinical or machine learning methods for three selected endpoints evaluated on the internal longitudinal testing set. **g**, AUCs of MILTON, a biomarker-based machine learning model (x-axis) in prognostic mode compared to Delphi AUC values from the horizontal validation set (y-axis) for n=410 diagnoses.

Lastly, it's important to note that while sharing the same architecture, large language models (LLM) used in commercial products such as ChatGPT are different from Delphi since they work with natural language and use text tokens, while Delphi operates in the disease token space, working directly with disease rates. This makes us speculate that even though LLM can yield qualitatively reasonable results, as shown in our comparison with LLaMa, their approach of treating numbers as text should put them behind specialised models. Without revealing sensitive patient data, we asked ChatGPT-4o about the population average base disease rates using the prompt "What is the per-day death rate for a 65-year-old male? Answer with the number only." for the subset of diseases highlighted in Figure 2a. Below are the ChatGPT estimates compared to population average baselines obtained from the UKB and the same estimates produced by Delphi.

Comparison of Delphi-2M and ChatGPT-4o. Comparison of population-average disease rates for a 65-year old male predicted by Delphi-2M and ChatGPT-4o against the true rate, estimated using UKB data. [for review purposes only].

Disease	True UKB Rate	Delphi Rate	ChatGPT Rate
A41 Other septicaemia	2.40e-03	2.24e-03	2.70e-05
B01 Varicella [chickenpox]	2.17e-05	2.06e-04	2.25e-05
C25 Malignant neoplasm of pancreas	1.95e-04	3.70e-04	6.80e-05
G30 Alzheimer's disease	1.84e-04	3.10e-04	1.31e-04
E10 Insulin-dependent diabetes mellitus	8.34e-04	8.81e-04	2.40e-03
F32 Depressive episode	2.92e-03	2.62e-03	6.30e-04
I21 Acute myocardial infarction	4.23e-03	2.24e-03	8.00e-04
J45 Asthma	3.51e-03	2.37e-03	1.75e-04
Death	5.83e-03	9.09e-03	1.54e-04

Once again, we thank the reviewer for encouraging us to conduct this comparison.

Actions:

- Added LLaMA 3.1 8B to the comparison (Figure 2f)
- Added comparison details to the methods
- Evaluated ChatGPT-4o in the population-average disease rates prediction task and compared the results to Delphi

(ii) AutoPrognosis 2.0 <https://doi.org/10.1371/journal.pdig.0000276>, which has been applied to UKB data for prognostic risk prediction of diabetes, but “can be applied to construct diagnostic and prognostic models for any disease or clinical outcome, and is explicitly designed to make model building accessible to both experts and non-ML experts.” The performance of Delphi-2M needs to be compared to this model (e.g. on CVD).

This is a great point raised by the reviewer, and we appreciate the suggestion. We have added a comparison of AutoPrognosis 2.0 (v3) for cardiovascular disease (CVD), using the covariates described by the authors in their new paper (mentioned by the reviewer in the next point). This is an important comparator to our approach (Figure 2f), and we thank the reviewer for bringing it to our attention.

Actions:

- Included AutoPrognosis 2.0 in new Figure 2f and Extended Data Figure 7
- Added comparison details to the methods

(iii) The work should also relate to a new advancement building on AutoPrognosis 2.0, <https://arxiv.org/abs/2310.03560> (v3), to “explore a new application of LLMs in healthcare and propose using LLMs to facilitate clinician interactions with AI models and digital tools.” to limit “potential for hallucinations” of LLM models and “ensure actionable information is provided by approved clinical sources”.

As mentioned earlier, we used the described model for our comparison. The LLM interface is indeed intriguing. While being a very promising tool for aggregating information from unstructured clinical records, LLMs often cannot provide accurate numerical estimates, as shown in our response to an earlier point. This isn’t new - LLMs are generally known to perform poorly in tasks requiring exact solutions, such as evaluating mathematical

expressions. To overcome this, ChatGPT has the functionality to write and execute Python programs for such tasks. One could envision a similar setup with our model, where an LLM would “delegate” critical tasks to Delphi.

We have added a section to the discussion highlighting such symbiosis as a potential evolutionary path for future healthcare models, including a reference to the suggested paper. This is excellent work, and we thank the reviewer for bringing this publication to our attention.

p.38, l.41: “Lastly, Delphi-2M itself could serve as an extension to LLMs. Similarly to systems that provide LLMs with query-relevant web search results to reduce hallucinations (Shuster et al. 2021), a future healthcare-oriented LLM could invoke a Delphi-based model to improve numerical accuracy of the generated replies(Imrie et al. 2023).”

Actions:

- **Added a section to the Discussion about the potential integrations with LLMs**
- **Referenced the suggested paper**

(iv) Aside from the technological advancement in (iii), there are several metrics therein that should be included in the manuscript for comparison. For example, there are several risk scores for CVD, Framingham score (in the United States), SCORE2 (in Europe), and QRisk3 (in the UK). There is also a more recent study by the American Heart Association <https://doi.org/10.1161/CIRCULATIONAHA.123.067626>, to assess the PREVENT score for CVD. These CVD scores need to be compared with the model (not only Framingham) as the performances seem to differ depending on the score used. The manuscript is expected to present Tables similar to Table 1 in <https://arxiv.org/abs/2310.03560> with additional scores for model discrimination and calibration.

We would like to thank the reviewer once again for their excellent and clear recommendations. As suggested, we have conducted an extensive comparison for cardiovascular disease (CVD). We now include QRisk3, Prevent, Score2, Framingham, Transformer, AutoPrognosis, and LLaMA 3.1 8B as comparators to our model (Figure 2d). Additionally, we have added a table with further metrics (Extended Data Table 2).

Actions:

- **Performed additional comparisons, which now include:**
 - **Death: Elixhauser Comorbidity Index, Charlson Comorbidity Index, Transformer-based model, Llama 3.1 8B LLM**
 - **Dementia: UKBDRS, Transformer, Llama 3.1 8B**
 - **CVD: Qrisk3, Prevent, Score2, Framingham, Transformer, Llama 3.1 8B, AutoPrognosis**
 - **MILTON, a recent machine learning algorithm that utilises 67 different biomarkers to predict a wide spectrum of ICD diagnoses.**
- **Updated figures (Figure 2, Extended Data Figure 7) with the new comparison results and modified Methods accordingly**
- **Included the additional comparison details as Extended Data Tables 2, 3**

Figure 2. Delphi-2M accurately models the rates of a wide range of diseases. (subset)

f, AUC-ROC curves for Delphi and other clinical or machine learning methods for three selected endpoints evaluated on the internal longitudinal testing set. **g**, AUCs of MILTON, a biomarker-based machine learning model (x-axis) in prognostic mode compared to Delphi AUC values from the horizontal validation set (y-axis) for n=410 diagnoses.

(v) The imputation strategy in <https://arxiv.org/abs/2310.03560> for CVD, uses state of the art methods including MICE and Multiple imputation, as opposed to a simple mean imputation employed by the authors. To what degree this affects the results is unknown and should be investigated.

The reviewer highlights an important issue, and we agree that single mean imputation is an insufficient strategy for handling missing data. Following the approach outlined in the mentioned paper, we have replaced our evaluations with 5-fold multiple imputation by chained equations (MICE) and report the aggregated results according to Rubin's rules. We thank the reviewer for this important contribution.

Actions:

- **Used 5-fold multiple imputation by chained equations (MICE) for the clinical scores in the comparisons shown in Figure 2f, Extended Data Figure 7c**
- **Updated Methods accordingly**

Therefore, before aiming to predict multiple outcomes, it is important to compare Delphi-2M's model performance, in depth, on a disease such as CVD, relative to other models.

This is a valid point raised by the reviewer. As mentioned in (iv), we have now conducted an extensive comparison, which we hope adequately addresses the concerns raised.

(vi) The manuscript lacks in providing insights into scenarios where the model performance is better or worse or unchanged, e.g., (i) does prediction performance differ, at a fixed number of years X, between individuals born earlier (1940) vs later (1960); how does the prediction performance change for every extra year into the future; does the prediction accuracy differ amongst individuals with different ethnicities and/or deprivation metrics in UKB?

This is a very good point by the reviewer. It is indeed crucial to understand how our model performs across different sub-cohorts. We have expanded our evaluation of age-sex-stratified AUCs for ethnicity, deprivation, sex, and age of birth cohort (Extended Data Figures 18 and 6). These data show that, overall, Delphi's performance is comparable across

these sub-cohorts. This is very reassuring from a modelling point of view. However, we also believe that it is important to note that this may only hold in the idealised setting of a well-curated research cohort such as UK Biobank. It remains to be seen whether real-world data, which will contain different extents of missingness in particular demographics leading to additional biases.

We thank the reviewer for these important suggestions.

Extended Data Figure 18. Effects of ethnicity and deprivation. **a**, Modelled rate per year separated by sex and ethnic background. **b**, Modelled rate per year separated by sex and Townsend deprivation index bins (increasing for greater deprivation index values). The boxplots in **a** and **b** feature median as the center line, the box from the first to the third quartile, the whiskers for 1.5x IQR and the outliers. **c-d**, Average number of disease tokens per year, shown for different ethnicities (**c**) and deprivation indices (**d**). **e**, Average validation AUC across 5-year age groups ranging from 40 to 80 years of age, aggregated by the corresponding ICD chapters. Difference between average AUCs calculated for participants with birth years before 1944 and after 1960. **f**, Average per-chapter AUC, participants stratified by deprivation index (**f**) and ethnicity (**g**).

a**b**
Extended Data Figure 6. Change of Delphi performance over time into the future. Average validation AUC across 5-year age groups ranging from 40 to 80 years of age, aggregated by the corresponding ICD chapters. **a.** Each grey line represents the AUC for a particular disease when evaluated with different time gaps, with the bold line indicating the per-chapter average. The shaded region indicates the 95% confidence interval for average. **b.** Same data as in **a**, showing the delta AUC for all diseases with a 10-year time gap compared to the next token AUC.

Actions:

- Added an additional panel in Figure 2e showing the decrease in AUC values over longer periods of time (from next disease to 10 years into the future), more granular details in Extended Data Figure 6
- Subcohort results shown in Extended Data Figure 18
- Extended the results section “Limitations and biases” to describe the comparison of AUC values between these groups.
- Modified the Discussion to mention this analysis

p. 31, l. 13: *“In particular, we identify and discuss the consequences of the following biases underlying UK Biobank training data: Recruitment bias, sampling/immortality bias, ancestry background, levels of deprivation and health data availability bias.”*

p. 32, l. 37: “The predicted rates of observed diseases are generally comparable between the sexes (**Extended Data Figure 18**). Across self-reported ethnic backgrounds, however, lower rates for individuals self-reporting “white” are predicted, reflecting the observed disease rates (**Extended Data Figure 18a,c**). Within each ethnicity, Delphi-2M exhibits similar AUCs, with a trend towards lower values in the white ethnicity group (**Extended Data Figure 18g**). Similarly, Delphi-2M’s predicted rate of disease is elevated for groups with a higher Townsend deprivation index, which mirrors the average number of tokens per individual in this group and coincides with a higher AUC (**Extended Data Figure 18b,d,f**).”

p. 38, l. 22: “Furthermore, Delphi-2M predicts different disease rates in subgroups based on ancestry background and deprivation indices but no observable trend between lifestyle measures and birth year.”

Minor comments:

1 Certain sentences in the manuscript are left vague without providing succinct evidence. For example, in the introduction it is stated “The analyses not only reveal that Delphi-2M provides accurate instantaneous multistate rates ...”, “accurate” here is a qualitative statement and should instead be immediately backed up by quantitative evidence. Similarly in “The predictions remain accurate even if evaluated on an external dataset from Denmark without any additional fine tuning” the word “remain accurate” is sloppy and not true according to the authors, since later on in the discussion it is stated that “[Delphi-2M has] reduced accuracy on Danish registries ...”. “However, a key distinguishing feature of Delphi compared to basic GPT models is its ability to calculate the absolute rates of tokens, which in a statistically accurate sense represent inter-event times”, what does “statistically accurate sense” mean quantitatively?

We agree with the reviewer, our wordings have been too vague at places. We have added quantitative measures to our statements and use statistical terminology to avoid ambiguous expression. We thank the reviewer for bringing this to our attention.

Actions:

- **We have reworded the statements in questions and other sentences.**

p. 4, l. 5: “The analyses reveal that Delphi-2M predicts 97% of diagnoses with an average age and sex-stratified AUC (area under the receiver operating characteristic curve) of 0.76 (s.d. 0.08) for the next diagnosis and AUC of 0.73 (s.d. 0.08) for diagnoses 1 year into the future in internal validation data.”

p. 4, l. 10: “Delphi performs similarly in longitudinal testing data from the UK (1-year gap average AUC 0.69, s.d. 0.09) and from Denmark without any additional finetuning (1-year gap AUC Denmark: 0.67 (s.d. 0.09).”

p. 5, l. 33: “However, a key distinguishing feature of Delphi compared to basic GPT models is its ability to calculate the absolute rates of tokens, which provide consistent estimates of inter-event times (**Figure 1g**).”

p. 13, l. 24: *“Importantly, calibration analyses in 5-year age brackets show that the predicted rates closely match the observed number of cases, showing that the models’ rates of the next tokens are consistently estimated (Extended Data Figure 5).”*

2 Figure captions are poorly explained. See for example, Figure 1, panels (e), (f) and (g) where very little is said about what is going on in the figure, or the main messages for the reader. Same with Extended Figure 1. Similarly for Figure 2, there is a lot going on in panel (a) and the caption provides no information, e.g. what is the black line “selected case”, what does the larger black circle represent? All figure captions need to be drastically improved.

Thank you for highlighting these issues. We agree that they have been insufficient in many places. We have fully updated all Figure and Extended Data Figure legends to ensure all elements are sufficiently explained.

Actions:

- **We have reworded all Figure legends**

3 Inclusion of “Overall health rating UKB data field 2178” in the model and comparison performance with and without. This has been demonstrated to improve prediction in the literature.

Thank you for your suggestions. We compared per-disease AUC for Delphi and AUC based on “Overall health rating UKB data field 2178”, which shows that 2178 does indeed appear to be an interesting candidate predictor for future disease risks (avg. AUC 0.6) as the disease risk proxy (Extended Data Figure 8b). At the same time, Delphi’s AUCs appear to be consistently higher, showing that Delphi learns many additional features.

Extended Data Figure 8b. AUC results comparing Delphi-2M to a simple disease predictor of Overall health rating UKB data field 2178. AUC values for field 2178 as a predictor for future health events (after the data of recruitment) (x-axis) against the AUC values from Delphi using the horizontal validation data.

While we firmly believe that future Delphi extensions should include further data such as more detailed life style, blood counts and biochemistry and possibly event multi modal data (see Discussion p.38 lines 31-37), we chose – with the exception of very basic lifestyle factors – not to include additional data into the model, as this would warrant a more systematic approach and further discussion of which variables are most suitable for inclusion. Additional reason for our restriction of the used covariates is that we wanted to ensure reasonable transportability to the Danish registry data. We hope that the reviewer understands our reasons.

p. 38, l. 31: “Immediate refinements of Delphi-2M may incorporate additional lifestyle data, prescription records and blood tests, both usually available in a general healthcare setting. Further multi-modal extensions could include genomic data, richer metabolomic information, or data from wearables that can be added to Delphi-2M’s embedding layer, similar to how lifestyle tokens are currently incorporated. Such multi-modal extensions will shed light on how different layers of biology interface in shaping health outcomes and thus help address long-standing questions in biomedicine.”

Actions:

- **New Extended Data Figure 8**

4. UMAP is well known to distort distances. The representation and trajectory statements in Extended Fig 7 is misguided and should be removed, e.g., “trajectory traverses the labelled diagnosis in close proximity”.

Thank you for raising this point. It is correct of course that UMAP only provides an approximation that is, at best, locally accurate and thus did not provide suitable evidence for the points we wanted to illustrate. We have therefore removed these figures.

Delphi uses weight tying, which means that the embeddings are not just vector representations for the tokens but are also used as a linear projection layer from the latent space to the final logits. Since it is a linear operation, the local structure of the data manifold is preserved, meaning proximity in the disease embedding space (which approximately translates to proximity in the UMAP space) also suggests correlated logits.

To illustrate that the occurrence of a token leads to higher predicted rates of other tokens in its local embedding vicinity, we colour the UMAP using the corresponding SHAP values (effect of the occurred token on the coloured tokens). This has been added as Extended Data Figure 14.

We've modified the main text in order to reflect that this approach serves exclusively for the purpose of building intuition and UMAPs, because of their inexact nature, should never be overinterpreted. We also believe that tying both the existing SHAP analysis and the UMAP representation together will help readers build an intuitive understanding of how Delphi's predictions are established.

We thank the reviewer for highlighting this.

Extended Data Figure 14. Connection of disease co-occurrence and colocation in the embedding space. UMAP scatter plot, coloured by the SHAP-explained disease rate change for the disease of interest, denoted by a cross marker. According to the SHAP analysis, diseases with similar embeddings tend to have more effect on the predicted rate of each other. **a.** UMAP, coloured by the disease ICD-10 chapter. **b.** Top row, the effect of the selected disease on other diseases. Bottom row, the effect of other diseases on the selected disease. **c.** Same as **b**, more diseases.

p. 23, l. 42: “Notably, such patterns often appeared symmetrical, indicating similar predicted effect sizes of one disease token influencing another and vice versa (Extended Data Figure 13). This behaviour can be attributed to the structure of the embedding space, which places temporarily co-occurring diagnoses in local proximity (Extended Data Figure 14). Due to the linear relation between embedding and logarithmic disease rates local proximity in the embedding space also leads to a similarity of modelled disease rates.”

Actions:

- **Removed UMAP trajectory figures**
- **Added SHAP value analysis as Extended Data Figure 12**

Referee #2 (Remarks on code availability):

I have not ran the code, but the GitHub page seems to have the relevant instruction and scripts available for the user. Jupyter notebooks are also made available.

Thank you for taking time to assess our code. During the revision, we continued to work on the codebase, improved the code readability and documentation, specifically for data preprocessing, added a script for faster AUC evaluation and increased overall performance. We have also added Docker support, which could be helpful for running Delphi in the “out of the box” fashion.

Referee #3 (Remarks to the Author):

In this paper, the authors introduce a transformer-based foundation model named Delphi, trained on UK Biobank data to predict future disease rates across a broad spectrum of diseases. By adapting the tokenization technique used in the GPT-2 model to their specific dataset, and adding an additional predictive head for timing the next token alongside event classification, the model demonstrates a novel approach to disease prediction.

After training, Delphi underwent longitudinal validation using both the UK Biobank and an external cohort from Denmark. The results show well-calibrated disease rates stratified by age, and the performance of Delphi's predictions is comparable to established clinical risk models, such as the Framingham score for cardiovascular diseases. The validity of the model is further supported by UMAP visualisations of disease code embeddings in Delphi-2M, which cluster effectively by the underlying disease category, and SHAP score analysis that highlights comorbidities. The paper also conscientiously addresses potential biases such as recruitment, immortality, and data availability.

The application of advanced transformer-based techniques in predicting patients' disease risk across hundreds of diseases based on their medical history, represents a significant innovation in medical AI. The comprehensive validation across multiple cohorts, along with comparative analysis against established clinical models, lends considerable credibility and potential impact to the findings presented in this paper. I have identified several areas that require further attention:

We thank the reviewer for the thorough review and for the helpful recommendations to improve the paper. In summary, we provide a more extensive comparison of Delphi and also a detailed evaluation in relevant sub-cohorts. We have refined our description of technical aspects of the approach, like the interpretation through SHAP values, the use of no-event tokens, and the various missingness patterns. Finally, we have clarified and expanded our discussion on the use-cases we see for Delphi.

Please find replies to specific comments below.

1. Additional Validation of Delphi's Predictions:

a. The authors provide calibration of risk predictions stratified by age and sex across the UK Biobank and the external Danish cohort. However, further stratification analyses based on other model input features, such as smoking status, body mass index, and alcohol consumption, are necessary to fully assess calibration and to evaluate remaining inter-person variability after accounting for these variables.

The reviewer raises an important point here, rigorous evaluation of a method should indeed be assessed in relevant sub cohorts for robustness. Therefore, we have added additional evaluations of discrimination for the following sub-cohorts:

- **Birth cohort pre 1944 versus post 1960**
- **Smoking, additionally split by sex**
- **Alcohol consumption, additionally split by sex**
- **Body mass index (BMI), additionally split by sex**
- **Townsend deprivation index (range 1-5)**

- Self-reported ethnicity

These results are included in a new Extended Data Figure 17, copied below. While there is some variation due to fitting and sampling noise we did not observe a particular difference across these subgroups. We thank the reviewer for encouraging us to perform this important analysis.

Extended Data Figure 8. Additional comparisons. (subset) c. Boxplots, showing the prediction AUCs for Delphi, split over sex, disease chapter and lifestyle factors, such as alcohol consumption, smoking and BMI.

Extended Data Figure 18. Effects of ethnicity and deprivation. **a**, Modelled rate per year separated by sex and ethnic background. **b**, Modelled rate per year separated by sex and Townsend deprivation index bins (increasing for greater deprivation index values). The boxplots in **a** and **b** feature median as the center line, the box from the first to the third quartile, the whiskers for 1.5x IQR and the outliers. **c-d**, Average number of disease tokens per year, shown for different ethnicities (**c**) and deprivation indices (**d**). **e**. Average validation AUC across 5-year age groups ranging from 40 to 80 years of age, aggregated by the corresponding ICD chapters. Difference between average AUCs calculated for participants with birth years before 1944 and after 1960. **b**. Average per-chapter AUC, participants stratified by deprivation index (**f**) and ethnicity (**g**).

b. Similar stratification analyses are also required for a more comprehensive validation of simulated patient trajectory analyses from Delphi.

As there is no “ground truth” for the simulated outcomes, we also compared the simulated (conditional on data until the age of 60) and observed incidences in difference subsets, thus stratifying the assessment in original Figure 3b. This is especially interesting if certain groups experience differential outcomes throughout their life. These analyses confirm that Delphi’s simulations correctly predict the disease burden in sub-cohorts defined by smoking, alcohol consumption, body mass index. Importantly Delphi-2M’s simulations starting at age 60 correctly predict the rate differences observed between ages 70-75 in each of the subsets. These results are now included in Figure 3d and Extended Data Figure 10a:

Figure 3. Generative modelling with Delphi-2M informs future outcomes (subset). **d**, Simulated (x-axis) and observed (y-axis) fold changes of disease rates for high versus low smoking, alcohol consumption and BMI groups. The evaluation period included ages 70-75 and used simulations from the age of 60.

A similar assessment shows that the trends of disease rate changes in subgroups defined by absence/presence of certain prior illness are correctly simulated:

Extended Data Figure 10. Assessment of simulated health trajectories (subset). c, Fold changes in individual disease risk for the groups with and without prior diseases shown in **b**.

We note that a standard error of approximately 2x is expected for the sample sizes analysed here. These analyses are shown in a new Extended Data Figure 10.

Moreover, we carried out similar analyses, which confirm that simulations produce consistent estimates of the disease burden at age 70-75 in groups defined by self-reported ethnicity and deprivation. These new results are included in Extended Data Figure 19.

Extended Data Figure 19. Stratification analysis for generated trajectories. Simulated and observed disease rates recorded between ages 70-75 in UK Biobank validation data. Simulations use data until the age of 60. **a**, Scatter plots of simulated and observed rates in participant groups split by Townsend deprivation index. **b**, Scatter plots of simulated and observed disease rates stratified by self-reported ethnicity. Each dot is an ICD-10 diagnosis coloured by the disease chapter.

Thank you for this suggestion.

Actions:

- Expanded stratification analysis to include birth age, smoking, alcohol consumption, body mass index (BMI), Townsend deprivation index and ethnicity
- Summarised the results as the new Extended Data Figure 17
- Performed similar analysis for the generated synthetic trajectories for smoking, alcohol consumption and body mass index (BMI)
- Included the results for synthetic trajectories as the Extended Data Figure 18
- Modified manuscript text to reflect the observations

p. 31, l. 13: "In particular, we identify and discuss the consequences of the following biases underlying UK Biobank training data: Recruitment bias, sampling/immortality bias, ancestry background, levels of deprivation and health data availability bias."

p. 33, l. 37: "The predicted rates of observed diseases are generally comparable between the sexes (**Extended Data Figure 18**). Across self-reported ethnic backgrounds, however, lower rates for individuals self-reporting "white" are predicted, reflecting the observed disease rates (**Extended Data Figure 18a,c**). Within each ethnicity, Delphi-2M exhibits similar AUCs, with a trend towards lower values in the white ethnicity group (**Extended Data Figure 18g**). Similarly, Delphi-2M's predicted rate of disease is elevated for groups with a higher Townsend deprivation index, which mirrors the average number of tokens per individual in this group and coincides with a higher AUC (**Extended Data Figure 18b,d,f**). Simulated trajectories are also found to reproduce the observed disease burden in these groups (**Extended Data Figure 19**). Together these observations indicate that Delphi-2M's predictions reflect the differential burden of disease seen across different population subgroups but exhibit mostly comparable discriminatory performance within each stratum."

p. 38, l. 22: "Furthermore, Delphi-2M predicts different disease rates in subgroups based on ancestry background and deprivation indices but no observable trend between lifestyle measures and birth year."

c. Extending the comparison of Delphi's risk assessment with classical clinical scores and disease risk predictors, such as the Framingham score for CVD, UKBDRS score for dementia, and the Charlson score for death is important. The authors could also compare against quantitative biomarkers and clinical scores such as A1c for diabetes, mean corpuscular volume for anemia, and liver function markers for chronic liver disease. Such comparisons, even if Delphi's predictors cannot outperform these models, are essential to understand how LLM-based predictions based on medical history compare with established markers.

This is a very interesting point raised by the reviewer, and we appreciate the suggestions. We have added a comparison of our model's predictions against HbA1c and diabetes, MCV/haemoglobin for anaemia, and liver function markers for chronic liver disease (**Extended Data Figure 8a**). Thank you.

Extended Data Figure 8a. Comparison with clinical biomarkers. Comparison of Delphi-2M against AUC clinical biomarkers for selected diseases performed using the UKB validation dataset. Predictions are based on the information available at recruitment.

Actions:

- **Evaluated Delphi against clinical biomarkers for diabetes, anaemia and chronic liver disease.**
- **Included a new Extended Data Figure 8 to reflect this analysis**

2. Further Validation of SHAP Values: The authors utilized SHAP values to explore how individual tokens influence future risks. At present, it is unclear how these SHAP values relate to the average rate increases across disease tokens as presented in Figure 4. A systematic comparison of SHAP values with empirical rate changes is required to fully validate the interpretability of the model.

The reviewer highlights a very important aspect here. While interpretation of large AI models is generally challenging and is mired in complexity, in our case, we find that the SHAP values (at least on average) can be roughly understood in a manner similar to the hazard rates of classical Cox regression models. We ran a penalised Cox model on the same data for a handful of selected diseases and observed good agreement between the hazard estimates from the Cox model and our SHAP values, although the effects are slightly attenuated in the Cox model (Extended Data Figure 12). Similarly, in (Extended Data Figure 13f) we can see that the SHAP values over time have a close correspondence to time-dependent hazard ratios.

We hope these analyses help with the interpretation and validity of our SHAP analysis. However, obviously, interpretation of the effects from large AI models always need to be taken with care. We added a sentence in the text to emphasise this. We thank the reviewer for raising this import issue.

Extended Data Figure 12. SHAP values vs Cox proportional hazards coefficients. SHAP values are estimated by averaging individual values from different trajectories. Cox coefficients are estimated using a proportional hazards model with parameter regularisation, resulting in a high number of zero coefficients. The non-zero Cox coefficients and SHAP show a high correlation.

Actions:

- **New analysis and Extended Data Figure 10 confirming consistency of SHAP values and Hazard Ratio**
- **Added brief description in the accompanying text section:**

p. 23, l. 35: *"SHAP analysis of data from 100,000 individuals of the validation cohort reveals the mutual dependencies by which each disease, sex and lifestyle token influences the rate of subsequent disease tokens, similar to hazard ratios in conventional statistical models (Figure 4c, left; Extended Data Figure 12)."*

p. 28, l. 13: *"While Delphi-2M appears capable of modelling temporally directed dependencies, we caution, however, against interpreting these as causal relationships that could be exploited to modify future health courses."*

3. Explainable AI: "Of note such patterns often appeared symmetrical, indicating similar predicted effect sizes of one disease token influencing another as vice versa (Extended Data Figure 8)." I would love for the authors to explain this slightly more either in paragraph or in the discussion, to caution readers against immediately using these results to infer causality.

We expanded our discussion and re-emphasized where applicable that this should not be understood as causality. We also suggest that said symmetrical patterns may be observed because of the structure of the embedding space, as indicated by the UMAPs, and not because of the causal relationship within the disease pairs.

p. 23, l. 42: *"Notably, such patterns often appeared symmetrical, indicating similar predicted effect sizes of one disease token influencing another and vice versa (Extended Data Figure 13). This behaviour can be attributed to the structure of the embedding space, which places temporarily co-occurring diagnoses in local proximity (Extended Data Figure 14)."*

p. 28, l. 13: *"While Delphi-2M appears capable of modeling temporally directed dependencies, we caution against interpreting these as causal relationships that could be exploited to modify future health courses."*

Thank you for highlighting this.

Actions:

- **Added cautioning statements to the Discussion and the paragraph where SHAP values are introduced**
- **Performed additional analysis to explore the connection between SHAP values and embedding space structure (Extended Data Figure 14)**

4. "No other event" padding: I think the readers would benefit from the authors discussing the implication of using these paddings a bit more. If the authors have performed tests leaving out these "no other event" paddings it might be informative to the readers if the authors included them

in the supplementary materials for discussions on how these paddings allow the model to (a) predict trajectories at age X when there are no previous disease codes and (b) update attention weights and trajectory predictions after disease events lapse for a range of periods with no further events.

The reviewer emphasizes a crucial point, and indeed, our description of the ‘no event’ token has been lacking some explanations. The reason for introducing these padding steps is that Delphi always predicts the next token in the trajectory. Especially for younger ages, there are often long periods where no diseases are recorded. These can span multiple decades during which “baseline” disease incidences change substantially. For example at birth there are many individuals where the next diagnosis is a childhood disease, but also others where the first recorded token is during childbirth several decades later. To provide a better interpolation of these “empty” periods the ‘no event’ tokens are randomly introduced to the data with a predefined average rate. This improves incremental sampling and provides anchoring points when evaluating instantaneous risks.

We have clarified the role of “no event” padding in the main text and their other potential applications in the methods section.

p.4, l.22: *“Further “no event” padding tokens were randomly added at an average rate of 1/5 years to eliminate long intervals without other inputs and during which the baseline disease risk can significantly change.”*

p. 41, l.39: *“In addition to eliminating long time intervals without tokens, no-event paddings also allow evaluating the disease incidence at any point of the trajectory, by inserting them to the position of interest and predicting the next token from there.”*

To further assess the raised point, we have conducted additional experiments that change the average rate of “no event” paddings (new Extended Data Figure 1). This shows that there may be additional gains in average loss for denser padding, especially for early time points, where the density of observed diagnosis is sparse. These data also confirm that Delphi provides consistent estimators of the token rates.

We note, however, that denser padding comes with additional compute cost because more tokens are required. This argues against very dense padding as attention scales $O(n^2)$ with the input sequence. Also, the placement of no event tokens are by definition random and it is thus possible to make predictions with Delphi at any given time, irrespective of the rate of auxiliary no event tokens during training, by introducing a padding token at the time of interest.

Extended Data Figure 1. Optimal no event padding token rate. Average validation cross-entropy loss for variants of Delphi-2M, trained with different “No event” padding token rates. Cross entropy is calculated over disease tokens only - that is, does not include padding tokens, sex and lifestyle tokens. Every model has three technical replicates trained with different seeds. **a.** Boxplots of the average loss for different Delphi-2M variations. X axis denotes the inverse “No event token” yearly rate, which can be interpreted as the average time interval that contains 1 no “No event” token. **b.** The average loss, aggregated over 5-year age bins. Adding more “No event” tokens improved accuracy especially for younger ages. **c.** “No-event” tokens rate predicted by Delphi vs the true predefined rate.

We thank the reviewer for highlighting this interesting point.

Actions:

- **Added analysis of no event padding rate shown in Extended Data Figure 1**
- **Expanded discussion of role and use of no event padding**

5. Different types of missing data: As the authors wrote, 78% of participants have self-reported disease data, 45% have GP data and 86% have hospital records, and missing data is due to a combination of data acquisition, UKbiobank being a retrospective cohort, and healthy volunteer-bias, etc. I wonder if some of these (esp the large contributions of missingness e.g. lack of GP data) could be modelled as more types of "no event" paddings and if that would be of value in the Delphi model. I think some discussions along these lines would be interesting.

Thank you for this suggestion. While non-random missingness may already cause some biases in the model, we expect these effects to be significantly more apparent when using joint data from multiple healthcare systems for training. Therefore, building a good missingness modelling framework is a crucial prerequisite for scaling Delphi to multiple cohorts.

To further explore the potential effects of potential missingness-induced biases, we trained a model with tokens explicitly marked with their source. That is, instead of “Common cold” token the model would have “Common cold (self-reported)”, “Common cold (hospital records)”, etc - one for each of the available sources. When inspecting a UMAP of the learned disease embeddings of the source-split model, we noticed that diseases cluster primarily by their source rather than the ICD chapter (Extended Data Figure 17a,b). We attempted to reproduce this effect for our main model, Delphi-2M. Since most diseases have tokens coming from multiple sources, for this analysis, we only select diseases for which more than 75% of tokens come from the same source. (Extended Data Figure 17c,d). Per-source clustering is also observed here, e.g. diseases from ICD Chapter X. Respiratory Diseases are located in two clusters - “Primary care”-specific and “Hospital admissions”-specific, each colocated with other diseases primarily coming from the corresponding source.

To validate if the source-related information also influences model predictions, we reproduced the SHAP value heatmap (Extended Data Figure 17e), sorting the diseases by primary source. Interestingly, this not only reveals fine-grained intra-chapter interactions, but also some surprising inter-chapter connection - for example, acquiring a disease from “Skin diseases - Primary care” tend to increase the chance for further development of a disease from “Respiratory diseases - Primary care”, but not “Respiratory diseases - hospital admissions.”

While it is not clear to which extent the observed effects are undesired causes of non-random missingness and not the correct assumptions learned by the model (it e.g. it is reasonable to speculate that respiratory diseases that are diagnosed by the GP are different from those more severe that require hospital admission), modelling source-related effects seems to be necessary for expanding the model for broaden populations. We thank the reviewer for encouraging us to expand our analysis.

We agree that the reviewer’s suggestion to include indicator tokens of source availability is a promising avenue to mitigate some of the effects. However, one issue is that the true availability (including the case where no diagnoses are made in the presence of availability) is currently unknown to us (but could be retrieved by those who have assembled UK Biobank’s disease data. At this stage we could therefore only implement a proxy by using the first diagnosis from a given source, which might also introduce additional distortions. We would therefore prefer to keep this for future, more systematic investigations and use our assessment of biases as a call for action in this direction.

We have updated the discussion to better reflect these points and are providing the novel analyses as Extended Data Figure 17 (copied in below).

Actions:

- **Trained a new model with explicit token sources and analysed its potential source-related biases**
- **Found similar effects in our main model**
- **Used SHAP values to quantify the effects of those biases on model predictions**
- **Summarised the results as the Extended Data Figure 17**
- **Added more discussion of the source-related biases to the Discussion and the “Limitations, biases and fairness” section**

Extended Data Figure 17. Token source-related biases. Non-random missingness may cause biases in predictions even when courses are not explicitly provided to the model. **a.** Disease embedding UMAP for a Delphi model with explicit token sources (e.g. “Common cold (self-reported)” and “Common cold (hospital records)” are separate tokens), tokens coloured by ICD-10 chapters. **b.** Same as **a**, coloured by token source. **c.** Same as **a**, but for the standard Delphi-2M model. Only tokens with more than 75% of all entries from one source are shown. **d.** Same as **c**, coloured by primary token source. **e.** SHAP value matrix (similar to Figure 4c), with tokens grouped by chapter and primary source.

6. Differences in performance by demographic group: The authors wrote and demonstrated in Extended Data Figure 11 that the prediction is worse in White individuals because "is likely a consequence of participants of "White" ethnic background having lower numbers of disease tokens across time, which reduces the ability to predict future disease burden", and that is echoed by the results from stratification by Townsend deprivation index, where those with lower Townsend deprivation index who have lower numbers of tokens were predicted worst. Is this likely a property only of UKB which has a healthy volunteer bias? In EHRs, would it not likely be the other way round, where those with higher deprivation are likely to have lower access to healthcare and therefore lower tokens? How might Delphi2M, if applied on EHRs, perform differently, and how might this issue of difference in number of tokens across different demographics be solved such that usage of Delphi2M does not incur issues of healthcare disparities?

We thank the reviewer for these excellent points. The healthy volunteer nature of the UK Biobank does pose certain challenges for Delphi with an overall lower rate of disease incidence than the general population. Within the UK Biobank cohort there is a relatively high level of variation in the number of diseases recorded on the individual level, ranging from zero to many hundreds, which is likely to be a combination of differences in actual disease occurrence, differences in how individuals interact with the healthcare system and differences in the completeness of data capture. We expect that when applied to population scale EHR records Delphi predictions will become more accurate for most diseases (and in particular those seen at lower rates in the UK Biobank), however, additional biases will be present. The addition of extended baseline data into Delphi training regimes including geographical information and socioeconomic measures could help to account for certain biases relating to different levels of access to healthcare. Additionally we believe that future implementation of Delphi could be used to explicitly model individuals or groups that would benefit from increased health resourcing or outreach activities from healthcare providers.

We have extended our discussion into these types of applications and the types of biases that are likely to be present, discussing future prospects and also limitations. We thank the reviewer for highlighting this important aspect.

Actions:

- **Extended evaluations of biases to include further group comparisons and AUC values for all sub populations assessed (Extended Data Figures 18 and 19)**
- **More discussion into practical applications of Delphi with respect to healthcare implementations and limitations (Discussion)**

7. Discussion of Practical Implications: While the paper aims to model disease progression to inform healthcare planning, it lacks a detailed quantitative analysis of how these predictions could benefit healthcare systems. For instance, how might these predictions be integrated with quantitative biomarkers or clinical scores to enhance predictive accuracy? Should they be used to identify patients at risk for specialised testing? The authors need to expand on the discussion of practical implications of their findings with a robust quantitative assessment to demonstrate the

utility of these predictions in specific settings. This is particularly important to strengthen the impact of this work and ensure its suitability for publication in Nature.

Thank you and reviewer 1, who made a similar comment, for raising an important point, which we haven't discussed sufficiently. At this stage, Delphi constitutes a proof-of-concept that *a single generative model can be learned for the entire corpus of human disease with competitive performance* as single-disease models.

Delphi-2M's performance is further underscored by our additional comparisons against a number of established methods, as outlined above. Since some of these scores are used for medical purposes, there is potential for Delphi's use as a disease-specific decision support system. One advantage of multi-disease predictions is that Delphi can provide information for all disease-specific outcomes that a clinician might be interested in, rather than being limited to a specific implementation. This is likely to provide great value when evaluating potential interventions. However, we also caution that even though it appears likely that Delphi (or models inspired by it) lead to a range of applications ranging from individual risk assessment, therapeutic decision support or health care planning, this is currently not proven as each of these applications warrants in-depth and ideally prospective evaluation. We now emphasise these points more strongly in the Discussion (see below)

Additional utility for healthcare planning may be derived from Delphi's generative nature. While long term trajectories may not be necessary for imminent clinical decision making, as the reviewer notes, they can help obtain better long-term predictions. An interesting application for these trajectories could be in modelling future health burdens in various population subsets, using the model to project the future comorbidity landscape. There is some preliminary evidence that our model could be useful for this task (Figure 3c), but a proper evaluation and comparison with projections from e.g. the Global Burden of Disease study would be an intriguing follow-up, highlighting its use in more epidemiological endeavours.

Lastly, Delphi establishes a modelling framework for electronic health record (EHR) data that provides a foundation for many important health tasks, ranging from clinical to pharmacological and epidemiological applications. As we discuss in the manuscript, Delphi may be easily extended to incorporate additional types of health data such as laboratory values which are likely to further improve its accuracy and utility.

Taken together, we appreciate the reviewer highlighting this point, as we did not adequately convey these various aspects in the manuscript. We have revised the manuscript and expanded our discussion to emphasise the aforementioned points. We thank the reviewer for this very important point.

Actions:

- **Added further comparison to clinically established risk models, now shown in Figure 2f:**

Figure 2. Delphi-2M accurately models the rates of a wide range of diseases. (subset)

f, AUC-ROC curves for Delphi and other clinical or machine learning methods for three selected endpoints evaluated on the internal longitudinal testing set. **g**, AUCs of MILTON, a biomarker-based machine learning model (x-axis) in prognostic mode compared to Delphi AUC values from the horizontal validation set (y-axis) for n=410 diagnoses.

- **We discuss the proof-of-concept nature, potential use cases and road to implementation in the Discussion:**

p. 38, l. 46: *“Currently, Delphi-2M constitutes a proof of concept that AI architectures originally developed for natural language processing can be extended to model the temporal sequences of complex healthcare events. Notwithstanding Delphi-2M’s current limitations, its encouraging performance relative to other methods and its transferability between UK and Danish healthcare systems indicated there will be clinical utility of Delphi or similar tools in the future.*

An evident application of Delphi-type models would be to support medical decision-making by the probability distribution of future trajectories for a patient. A direct use could be to identify individuals who meet established risk thresholds, e.g. by age or biomarkers, that warrant inclusion in screening programs or further prevention measures. As discussed above, it would be critical to assess and mitigate biases present in the training set before any such system is tested in a clinical setting. Even with a relatively unbiased model, the benefits of such risk stratification must first be assessed through randomised controlled trials before broader implementation. Additionally, deploying clinical decision support systems requires a regulatory framework, which is still in its infancy for AI in healthcare. A long-term objective for AI-based clinical decision support systems is to provide direct treatment recommendations. Currently, Delphi does not explicitly model the effect of treatments, and usually, it is only possible to estimate these effects reliably using data from randomised clinical trials. Therefore, it is essential to establish frameworks within which clinical trial data can inform AI models, which are in turn, to be evaluated in subsequent randomised studies.

An alternative utility of Delphi models would be to help allocate healthcare resources by identifying the needs for future interventions. At a local level, for example, this could inform care trajectories on admission to a hospital. Such applications, informing the providers rather than treatment decisions, require less regulation. Similarly, there may be substantial system-wide modelling benefits for Delphi type models, in particular in the tailored projection of a cohort using their current healthcare until a point in time. This has the

potential to produce better estimates of the future for aggregate healthcare demand, which is useful at regional and national health system levels, as well as for insurers. This will help allocate resources that can be scarce in ageing populations.

These considerations illustrate the wide range of applications of generative models for biomedical research and, ultimately, also for healthcare. With appropriate training and evaluation, future multi-modal model extensions may be used for preventative medicine, clinical decision support and healthcare planning. Our model and analysis presented here present a further step to unlock the considerable healthcare benefits from the era of AI.”

Revision summary

We would like to thank all three reviewers for their constructive feedback and insightful comments.

In this revision, we have addressed all outstanding issues raised by reviewers 2 and 3.

In particular, we have implemented the following changes, which were common reviewer requests:

- We have developed a statistical methodology to assess whether Delphi performs unexpectedly in distinct demographic subsets, as requested by reviewer 2. This analysis shows that, once one accounts for the uncertainty of AUC estimation on the validation data due to low number of cases, there is, for the majority of diagnoses, little evidence of heterogeneity across population groups defined by ethnicity or deprivation.
- We have included an analysis that shows how Delphi can be combined with additional data types, including “overall health rating,” blood tests, and polygenic risk scores, as requested by reviewers 2 and 3. These analyses show that a combined model usually exhibits the greatest predictive accuracy, confirming our suggestions made in the discussion that multi-modal extensions will be even more powerful than the current Delphi-2M implementation.
- We have also included further nuance about Delphi’s limitations in relation to other more basic methods and greatly expanded on its potential utility as requested by reviewers 2 and 3.

In the following, we provide detailed point-by-point replies to all reviewer comments.

Thank you again for the suggestions, which we believe have further improved the manuscript.

Referee #1

My major comments in the 1st round review include (1) Clarifying the training/validation/test data split; (2) Comparing the proposed method with a strong baseline; (3) strengthening the clinical importance.

Comment #1 is solved by clarifying that the UK biobank data before July 2020 was used for training and validation, and two years' data after July 2020 was used as an independent test set. I have no concerns about the rigor of evaluation.

Comment #2: New experiments were conducted to compare the proposed model with a strong baseline, MILTRON, a very recent model published in Nature Genetics. I'm confident that the new comparison results with a strong model demonstrated the advantage of the proposed methods.

Comment #3. For comment #3, in addition to the revised discussion content, I'd recommend the authors strengthen the importance from the point of view of healthcare IT deployment. A critical contribution of this study is that this new model provided a foundation to "deploy one AI system to provide decision support for many disease domains". Previous generations of non-transformer-based AI models could be customized through extensive human engineering to achieve similar performance for specific diseases, yet, the healthcare systems have to deploy hundreds of customized models for different diseases, which is a huge burden for healthcare IT.

Overall, the new revision has addressed most of my comments. This study has methodology contributions in repurposing the "position" embedding architecture for a new use - to model the time-to-event sequence in EHRs. There have been too many ChatGPT-style studies that simply throw everything to ChatGPT and play with prompts. This is a good example of adopting state-of-the-art transformer models for healthcare problems. I have no further concerns about this study.

We thank the reviewer for the comments and especially for highlighting the benefits of using a single multi-outcome model instead of multiple single-disease models in terms of reducing healthcare IT burden. We have amended the discussion to reflect this point.

Referee #2

The authors have provided additional analyses for the majority of the points raised in the review, this has improved the manuscript. Regarding the upgrade to GPT4 architecture, I appreciate the authors' response that this seems not possible due to the GPT4 not being available open source and UKB regulations.

Thank you for taking the time to review our manuscript. During this revision, we have improved the manuscript, clarifying our vision regarding future use of AI-

based disease risk models and conventional clinical scores. We have also performed additional analyses to validate the potential of using these two types of models together (Extended Data Figure 9) and quantify the ethnicity/ deprivation-associated biases (Extended Data Figure 19, Extended Data Figure 20).

Please find detailed responses to the comments below.

1. Given the new analysis results, I have a major concern regarding the relevance of this approach for clinical applications, or for generation of biomedical insight.

The authors need to be upfront, in the abstract and main body of the text, regarding Delphi's performance relative to simpler and transparent statistical models. Specifically, Delphi has approximately the same performance as QRISK3 (a widely used cardiovascular disease assessment tool in the UK and internationally) for CVD, noting that QRISK3 is based on one of the simplest parametric models for survival analysis, namely Cox proportional hazards regression, and fractional polynomials. The reason it is important to be upfront about this to the scientific community is to avoid the risk of over-optimistic interpretation of the use of complex AI/ML models (here, a generative transformer model) for clinical applications in practice when their performance is no better than simple and transparent expert-developed scores or simple statistical models. Of note, the features that QRISK3 takes as input are routinely available as part of EHRs. This argument also holds for other diseases explored, e.g. dementia (UKBDRS, a lasso model) and CLD (where a logistic model seems to have been used as alternative). For diabetes, in fact a single marker (HbA1c, AUC 0.84) outperforms Delphi (AUC 0.75). This is an important point that should not be deferred to the Discussion section.

As a scientific exercise, it is indeed valuable to see how generative transformer models would perform against current clinical metrics. In fact, it is very interesting to see that expert-derived clinical scores, when available and optimised, have a similar performance to a complex model of this type. However, this then begs the question, both from a clinical perspective and with respect to novel biomedical insights, why train a generative transformer model given there is often no gain in prediction accuracy relative to clinical scores (other than Death), whilst there is a loss in simplicity and interpretability?

We thank the reviewer for this comment and share their concerns about the utility of complex AI models when simpler alternatives are available.

We would like to clarify that we don't see Delphi as a replacement for dedicated clinical scores and biochemical markers, but rather as a general-purpose one-stop shop baseline.

While disease-specific biomarkers, such as HbA1c and fasting glucose in the case of diabetes, provide a strong baseline for disease risk estimation, obtaining them requires active medical intervention. As a result, these data are relatively scarce. The QDiabetes paper (BMJ 2017;359:j5019) reports that fasting glucose measurements are missing for 85.5% of participants, and HbA1c is missing for 93.8%. Delphi, on the other hand, uses a general disease history that is more widely available, marking its potential for healthcare planning and stratifying patients who may benefit from having their biomarkers checked.

Using data that is different from that used in traditional clinical scores also allows for improving the disease risk estimates by increasing the set of factors that are considered. That is to say, one's medical history contains potentially valuable information that may not be included in a manually defined clinical score, but can be learned from data by Delphi.

To illustrate this, for 10 highlighted diseases, we have trained a family of linear risk estimation models that use two groups of parameters - 57 clinical biomarkers (all biomarkers from the MILTON paper with less than 100K missing values, imputed with MICE) and Delphi-predicted risk.

Extended Data Figure 9 (subset). AUCs for logistic regression models that use biomarkers, Delphi or both. Average validation AUC across 5-year age groups ranging from 40 to 80 years of age, additionally stratified by sex. All models use sex and age as additional covariates. For prediction, only data before recruitment was used. The prediction target is 5-year disease occurrence.

While for some diseases, biomarkers serve as better predictors than Delphi (e.g., breast cancer, diabetes, depression), the most accurate model for all diseases is either Delphi or Delphi + Biomarkers. This indicates the importance of taking into account the complex interactions between diseases in a person's medical history and biomarkers.

We also agree with the reviewer that simple models have the advantage of interpretability, which is likely to lead to greater acceptance, especially if their results are meant to inform interventions (as opposed to, say, informing diagnostic testing). However, the reality is often that there may not be simple measurements but rather complex and perhaps unstructured data, which can only be modelled and integrated by AI-based approaches. It will be an important discussion for society to

have to which extent people are willing to accept AI-based clinical decision making. For this discussion to be informed, it is necessary to study models such as Delphi in order to fully understand their capabilities and limitations.

We clarified our position in the manuscript:

p.40, l.2: "The ability of transformer-based models to process large-scale heterogeneous data could be used to improve currently used clinical scores by supplementing them with data modalities that were previously challenging to integrate, without fully replacing them."

p.41, l.35: "These findings underscore that care must be taken when AI models are used for inference and applied for prediction in heterogeneous healthcare data sets, potentially marking them as useful additions to currently used clinical scores, rather than replacements."

Actions:

- **Performed new analysis to study the potential synergy between Delphi and biomarker-based scores (Extended Data Figure 9)**
- **Modified the main text to better reflect our position regarding replacing clinical scores**

Major comments:

2. The following sentence is misleading "Delphi-2M performs equally, or better than existing risk scores, machine learning models and LLMs for cardiovascular disease, dementia and death". This needs to be stratified as follows: The performance of Delphi is similar to routinely used clinical risk scores for CVD and dementia, and better than those used for death. For diabetes, the performance of Delphi is worse than a single marker (HbA1c) clinically used for risk prediction and diagnosis of diabetes.

Thank you for the suggestion. We modified the text to use the exact wording suggested by the reviewer.

p.12, l.37: "The performance of Delphi was similar to routinely used clinical risk scores for CVD and dementia, and better than those used for death. For diabetes, the performance of Delphi was worse than a single marker (HbA1c) clinically used for risk prediction and diagnosis of diabetes."

3. Extended Data Figure 18 (f) and (g). It is important to have a description of the observations on deprivations and ethnicity subgroups in the main text so that it does not remain "hidden" in the appendix. In particular, there are many occasions where the lower end of the whiskers hits below AUC 0.5, particularly for deprivation level 5 (12/16) and black ethnicity (7/16). This means that, in this case, Delphi-2M performs worse than guessing at random.

Thank you for highlighting this issue. We have investigated the potential reasons for such a poor performance and noticed that low AUCs were observed primarily in situations where a particular subgroup/disease combination had extremely limited case sample size ($n < 5$). The effect is particularly visible for ethnicities, where most of the subgroups lack a sufficient number of participants - e.g. $n=598$ for Mixed and $n=297$ for Chinese. Filtering out rare diseases effectively removes all diseases with $AUC < 0.5$, as shown in the plot below.

Low numbers may both lead to poor model performance due to the limited training data and pose challenges for thorough evaluation. To assess which of the reasons caused apparent poor metrics, we used DeLong's method to estimate confidence intervals for the AUC (Extended Data Figure 19 (previously 18) g-h) On average, diseases with such a low sample size have a very wide confidence interval width of 0.4. This suggests that AUCs below 0.5 mostly stem from the variance of the AUC estimates and do not provide evidence of Delphi's poor performance.

Extended Data Figure 19 (subset). Effects of ethnicity and deprivation. **e-f**, Age and sex stratified AUCs for 10 selected diseases. AUCs are averaged across 5-year age groups ranging from 40 to 80 years of age. 95% confidence intervals are calculated using DeLong's method. **g-h**, Width of DeLong's 95% confidence intervals for AUC vs number of cases, shown for different ethnicities and deprivation strata. For rare diseases, AUC estimates have high variance. **i**, Standard deviation between AUC estimates for different strata vs number of cases of this disease for the training dataset. Each dot represents a disease. **j**, Average validation AUC across 5-year age groups ranging from 40 to 80 years of age, aggregated by the corresponding ICD chapters. Difference between average AUCs calculated for participants with birth years before 1944 and after 1960.

A potential solution could be to filter out strata/diseases with a low sample size. To avoid introducing an arbitrary filtering threshold, we decided to restructure Extended Data Figure 18 and focus on the selected 10 diseases that we already use throughout the paper. We also supplemented this Extended Data Figure with additional panels, containing the new results regarding the confidence intervals, and commented on it in the manuscript.

p.35, l.1: *"Together, these observations indicate that Delphi-2M's predictions reflect the differential burden of disease seen across different population subgroups but exhibit mostly comparable discriminatory performance within each stratum. However, due to large confidence intervals for AUC for underrepresented subgroups, studying such biases remains challenging, especially for rare diseases (Extended Data Figure 19 g-l, Extended Data Figure 20)."*

p.34, l.44: *"For most diseases, the differences in prediction AUCs for different ethnicities and deprivation groups are not statistically significant and stem from variance in AUC estimation, however, with some exceptions (Extended Data Figure 19e-f, Extended Data Figure 20, Extended Data Table 4)."*

Actions:

- **Reworked Extended Data Figure 19, adding panels covering confidence interval patterns for different subgroups**
- **Added a section on AUC confidence interval estimation to the methods**
- **Added more discussion of biases and evaluation of their presence to the text**

4. Can the authors delve into why certain ICD10/ethnicity/deprivation combinations lead to good performance whereas others lead to poor performance as reflected by AUC? How does it depend on the type of ICD10 code, number of cases, ratio of cases to controls, and relevance of predictors for the disease under consideration?

As we described in our response to the reviewer's previous comment, there is ambiguity about whether the observed difference in AUC for different subgroups is due to differences in model performance or variance of AUC estimation. For example, if we estimate AUCs for diseases with small sample sizes, Delphi seemingly performs significantly worse for certain ethnicities. However, the variance for these estimates is also high - unlike for the same diseases for different deprivation subgroups, where due to larger sample size estimates are more confident and more consistent between subgroups.

As can be seen in the chart above, there is variation of the AUC estimates across strata, but the variance appears largely within the range of the confidence intervals (estimated by DeLong's method). In other words, all strata seem compatible with the same mean, considering their individual CIs.

For a quantitative evaluation of outliers among AUCs for individual subgroup-disease combinations, we compare the standardised AUC residuals ($\text{subgroup_auc} - \text{weighted_average_auc} / \text{subgroup_auc_std}$) to the Bonferroni-corrected 0.95 quantile of the standard normal distribution. (Extended Data Figure 20. a, d)

Similarly, to calculate the outliers on the disease level, comprising AUCs for all subgroups available for a given disease, we compare the sum of squares of standardised AUC residuals to the Bonferroni-corrected 0.95 quantile of the corresponding Chi-squared distribution. (Extended Data Figure 20. b, c, e, f).

Extended Data Figure 20. Analysis of AUC variance across different subgroups for ethnicity and deprivation. **a, d,** Analysis of AUC residuals for individual subgroups (subgroup_auc - weighted_average_auc). Left: AUC residuals for different diseases and different subgroups vs DeLong-estimated variance. Centre, same as left, but residuals are standardised by dividing by DeLong-estimated standard deviation. Right, distribution of standardised residuals. The red line indicates a standard normal distribution. **b, e,** Distribution of sums of squares of standardised residuals for different diseases. The red line indicates the corresponding Chi-squared distribution. For some diseases, there are not enough cases for certain subgroups, therefore, we treat diseases with different numbers of available subgroups differently. **c, f,** Q-Q plot for squares of standardised residuals for different diseases and the corresponding Chi-squared distribution.

Diseases with unexplained variability are shown in the table below:

Ethnicity

Individual subgroups

Disease	Cases	Standardised residual	AUC residual	Weighted mean AUC	Subgroup AUC	Subgroup AUC variance	Subgroup
J02 Acute pharyngitis	23	4.4468	0.0913	0.8191	0.9104	0.0004	black
H57 Other disorders of eye and adnexa	25	4.1908	0.0927	0.8258	0.9185	0.0005	asian

All subgroups

Disease	Cases	Weighted mean AUC	Chi-squared statistic	Number of subgroups
M79 Other soft tissue disorders, not elsewhere classified	15711	0.7887	10.2044	5
F17 Mental and behavioural disorders due to use of tobacco	7237	0.7949	28.3431	4
J06 Acute upper respiratory infections of multiple and unspecified sites	8898	0.8473	16.3123	4
J02 Acute pharyngitis	3569	0.8191	22.3913	3
G44 Other headache syndromes	931	0.8153	20.009	3
L30 Other dermatitis	6366	0.758	19.8909	3
E14 Unspecified diabetes mellitus	3996	0.7349	15.79	3
H10 Conjunctivitis	3719	0.824	15.428	3
H57 Other disorders of eye and adnexa	1273	0.8258	19.9262	2

Deprivation

Individual subgroups

Disease	Cases	Standardised residual	AUC residual	Weighted mean AUC	Subgroup AUC	Subgroup AUC variance	Subgroup
J03 Acute tonsillitis	19	4.5545	0.112	0.8136	0.9256	0.0006	Deprivation group 4
F17 Mental and behavioural disorders due to use of tobacco	577	4.4001	0.0473	0.7915	0.8388	0.0001	Deprivation group 4
E16 Other disorders of pancreatic internal secretion	30	4.1908	0.0543	0.898	0.9524	0.0002	Deprivation group 4

All subgroups

Disease	Cases	Weighted mean AUC	Chi-squared statistic	Number of subgroups
F17 Mental and behavioural disorders due to use of tobacco	6925	0.7915	34.7523	5
K64 Haemorrhoids and perianal venous thrombosis	6129	0.7455	19.3577	5
E16 Other disorders of pancreatic internal secretion	615	0.898	29.0199	4
J03 Acute tonsillitis	854	0.8136	26.6035	4
G47 Sleep disorders	2971	0.742	22.0383	4
C85 Other and unspecified types of non-hodgkin's lymphoma	38	0.8011	19.3946	2
K65 Peritonitis	258	0.8286	14.6223	2

To satisfy the asymptotic requirements of DeLong's method, for this analysis we have only considered age & sex brackets with more than 5 disease cases. We have additionally filtered out diseases with only one age & sex bracket available and diseases that have enough cases after filtering for only one subgroup.

For deprivation, we have identified in total 3 outliers from 1555 available individual disease-subgroup pairs (0.19%) and 7/463 (1.51%) outliers for diseases when considering all subgroups for a given disease. For ethnicities, we have identified 2/385 (0.52%) and 9/139 (6.47%) outliers, respectively. These numbers indicate that

Delphi is largely unbiased, however, with exceptions that need to be studied more thoroughly before real clinical application.

Notably, increasing the filtering threshold further reduces the fraction of outliers, potentially indicating that at least partially the outliers can be explained by the underestimation of AUC variance for individual subgroups by DeLong's method due to violation of the asymptotic assumptions. However, a high threshold also leads to rejecting too many diseases, therefore, we decided not to increase it.

Many of the current outliers correspond to underrepresented subgroups (Chinese, Black, Deprivation group 4) and rare diseases - similarly, underestimating variance due to very low sample size could potentially have contributed to identifying these samples as outliers. There is no apparent difference between ICD-10 chapters in terms of the number of outliers.

We agree with the reviewer on the importance of thorough validation of machine learning models regarding differences in performance for different subgroups and plan to further reduce such biases in later iterations of the model by explicitly providing subgroup-related information as additional inputs to the model.

We have added more details about this comparison to the methods sections of the manuscript.

p.34, l.44: "For most diseases, the differences in prediction AUCs for different ethnicities and deprivation groups are not statistically significant and stem from variance in AUC estimation, however, with some exceptions (Extended Data Figure 19e-f, Extended Data Figure 20, Extended Data Table 4)."

Actions:

- Performed an analysis to find diseases with unexplained variance between different subgroups
- Added the description of this analysis to the methods section
- Included a new supplementary figure to reflect the results of the analysis (Extended Data Figure 20)
- Added a new Extended Data Table 4 with diseases for which Delphi potentially has subgroup-related biases

5. Extended Figure 6(a). It would be informative to add a figure per chapter (similar to Extended 6a), showing the percentage of conditions in the chapter that are below AUC 0.5 over time (with the same x-axis as this figure) and repeat for AUC 0.6, 0.7, 0.8 and 0.9. The idea is to show how the proportions of AUC shift as time passes, i.e., how Delphi-2M's performance changes when looking further into the future.

We agree with the reviewer that the requested addition would help to better understand how Delphi-2M's performance changes over time. We have included it

in Extended Data Figure 6 in the form of a stacked bar chart, showing fractions of the diseases falling into different AUC bins at any given moment in time.

Extended Data Figure 6 (subset). Change of Delphi performance over time into the future. Average validation AUC across 5-year age groups ranging from 40 to 80 years of age, aggregated by the corresponding ICD chapters. **c**, Stacked bar chart, indicating the fraction of diseases that are predicted with a certain AUC, given a time gap into the future.

Actions:

- **Updated Extended Data Figure 6**

6. The comparison of the predictions of Delphi's performance with overall health rating as baseline is indeed useful to see and interesting. I appreciate the authors' comments on exploring inclusions of blood biomarkers in a second development of the method. However, I do not think overall health rating can be considered on the same footing as blood biomarkers or environmental exposures. It would be interesting to know if the addition of this field would improve the lower bound of AUCs obtained in demographics such as those with high deprivation and ethnic minorities. That said, I appreciate that this exact field may not be transferable to the Danish cohort.

Thank you for this suggestion. We agree that additional data, such as overall health rating, could improve the results for underrepresented subgroups. However, as described in our response to comment 4, given the low sample size for the said subgroups, it's challenging to tell if there's a significant difference between the performance for them, especially for rare diseases. As a result, while the addition of this field would likely improve the results, it may not be possible to evaluate the difference.

Therefore, we conducted an additional analysis to assess the potential of using the overall health rating field as input data for Delphi without separation into different subgroups. We evaluated a family of logistic regression models that use different

combinations of features (age & sex, overall health rating, Delphi logits) to predict the 5-year occurrence of 10 selected diseases.

Extended Data Figure 9 (subset). AUCs for logistic regression models that use overall health rating, Delphi logits or both. Average validation AUC across 5-year age groups ranging from 40 to 80 years of age, additionally stratified by sex. All models use sex and age as additional covariates. For prediction, only data before recruitment was used. The prediction target is 5-year disease occurrence.

As shown in the graph, overall health rating generally serves as a worse predictor of risk disease compared to Delphi (with some notable exceptions - for instance, it is not surprising that self-assessed health rating is a very good predictor for depression). However, using it together with Delphi creates the best or close-to-best performing model, marking its potential as an additional input for future iterations of Delphi, along with biomarkers and polygenic risk scores.

We have modified the discussion section to include the overall health rating as a potential future addition.

p.41, l.43: "Immediate refinements of Delphi-2M may incorporate additional lifestyle data, self-reported health status, prescription records and blood tests, both usually available in a general healthcare setting. Further multi-modal extensions could include genomic data, richer metabolomic information, or data from wearables that can be added to Delphi-2M's embedding layer, similar to how lifestyle tokens are currently incorporated (Extended Data Figure 9)."

Actions:

- Performed new analysis to study the potential of using overall health rating as an input for Delphi

- **Added description of this analysis to the methods section**
- **Added new Extended Data Figure 9, exploring other potential inputs**
- **Amended the main text to include the overall health rating as a future input candidate**

7. Minor remarks:

Extended Data Figure 8a (middle) the abbreviation MCV is not defined (mean corpuscular volume).

Page 3, caption with “XX” occurrences.

Thank you for pointing it out. We have corrected all affected captions.

Referee #3:

I have carefully looked through the authors responses to my own request and the other reviewers' requests. The authors have made a substantial effort addressing all reviewers' concerns, and have added substantial analyses and further comparisons to their previous submission.

We thank the reviewer for this in-depth review of our paper. During this revision, we amended the manuscript to include more of our motivation for building multi-disease models, validated the potential for expanding Delphi with additional modalities (Extended Data Figure 9), and improved our analysis of ethnicity/deprivation-associated biases (Extended Data Figure 20).

We have also improved the code and documentation in our GitHub repository.

A detailed point-by-point reply can be found below.

1. However, I was expecting a more in-depth response to my last point regarding the practical application of Delphi-2M. The claim that Delphi-2M is a "proof-of-concept that a single generative model can be learned for the entire corpus of human disease with competitive performance as single-disease models" is not particularly compelling. With appropriate engineering, such a model can indeed be trained—but for which specific application are these results a proof of concept? Addressing this question in greater depth is crucial for demonstrating the relevance and impact required for this venue.

The discussion on long-term trajectories remains highly speculative and would be extremely difficult to validate.

The statement that Delphi-2M has applications in "clinical, pharmacological, and epidemiological domains" is too broad and would benefit from more concrete examples.

We thank the reviewer for raising this important point. Indeed, it is crucial to provide a clear vision of how we see models like Delphi being translated and diffused into wider society. However, we would also like to stress that the sole engineering aspect is an important new development in its own right (as has also been highlighted by reviewer 1). This is a new architecture (modification) that enables several important aspects for clinical epidemiological tasks that have not been or only partially been addressed by previous ML/AI models, including:

- **Prediction of disease risk for any desired time horizon**
- **Multi-disease prediction**
- **Interpretability via attention and SHAP values, with links to more classical hazard ratio estimates**
- **Flexible modelling of interactions and time dependencies**
- **Modelling of health trajectories over arbitrary time points (continuous time)**
- **Generative modelling of future health progression**
- **A general framework that can be expanded to multimodal data**

With these capabilities at hand, we can think of several application areas in the clinical, pharmacological and epidemiological domains like:

- Personalised disease risk (including multimodal data sources)
- Multi-disease risk system (lower IT burden, lower overload on physicians to understand X different models/risk scores)
- EHR-based screening and triage
- Health burden modelling
- Digital twins
- Synthetic data (e.g. an alternative to federated learning -> learn on a joint synthetic data set)

For some of these, we can already show clear evidence (e.g. personalised disease risk), for some we have indication but would need further validation (e.g. long-term trajectories/health burden modelling), and some are in principle possible with our approach but so far more on the speculative side (e.g. digital twins/impact evaluation of treatments/interventions):

One of our visions for using Delphi-like models is: Every time the patient visits a GP, along with a regular check-up, it would be easy to monitor if there are any diseases with a risk significantly higher than average for the given age and sex (Fig 2a). In case there are any, the GP can use interpretability features to check what symptoms are of concern (Fig 4b) and potentially refer the patient to a specialist. In turn, the specialist can use Delphi's generative capabilities to sample potential future health trajectories given different medical interventions (Fig 3). Similarly, one could perform a "passive" screening using the records available in national biobanks. This way, it could be used to monitor which of the participants would benefit from a follow-up. On the other hand, it would inform healthcare providers about the expected disease incidence, enabling the estimation of future demand at the hospital, local, and national levels.

We have been able to show that Delphi-like models can perform these tasks, and for many, with promising performance. However, as the reviewer correctly points out, appropriately testing and evaluating some of these capabilities in real-world scenarios will require substantial additional effort and further validation.

We have included more examples to reflect use cases and our vision for Delphi in the manuscript:

p.42, l. 19: "An evident application of Delphi-type models would be to support medical decision-making by considering the unique probability distribution of future trajectories for each patient based on their individual history. A first use could be to identify individuals who would benefit most from diagnostic tests, such as Hb1Ac for the diagnosis of diabetes. Similarly, Delphi models can reveal individuals who meet established risk thresholds that warrant inclusion in screening programs or further prevention measures, but who don't meet conventional age-based criteria yet. A further promising feature here is that Delphi-type models can rationally integrate information from a variety of data modalities, which may be a challenge for health

care professionals. While these uses for clinical decision support appear promising, several hurdles need to be cleared. As discussed above, it would be critical to assess and mitigate biases present in the training data and model. Before broader implementation the benefits of an AI-based risk stratification need to be quantified through randomised controlled trials. Lastly, deploying clinical decision support systems requires a regulatory framework, which is still in its infancy for AI in healthcare.

An alternative utility of Delphi models would be to inform health care providers, insurers and policymakers. Such applications, informing the provision of health care rather than treatment decisions, require less regulation. Delphi's modelling has a resolution of an individual; this provides the opportunity also to provide aggregated estimates of the expected disease burden for specific groups of interest. At a hospital level, for example, this could inform the required care over the horizon of a week based on the specific patients admitted in the previous days. Similarly, Delphi-type models may provide substantial system-wide modelling benefits, particularly in projecting the expected disease burden at local, regional and national levels, or in specific demographic groups of interest. This could reveal communities with unmet future health care needs during the next 1-2 decades and the opportunity to adjust the provision of health care. In this context it is also critical to assess the full range of diseases to establish future needs and required resources. Such capabilities appear especially valuable in ageing populations where healthcare needs are becoming increasingly complex and resource-intensive."

3. Instead, I find two aspects highlighted by the authors particularly promising, but I believe they require further validation:

Multi-disease predictions – This is an interesting feature, with primary application in prevention and monitoring. In this context, a compelling proof of concept would be to compare Delphi-2M with Polygenic Risk Scores (PRS) and ideally demonstrate that Delphi-2M + PRS outperforms PRS alone.

Integration of quantitative biomarkers – The authors note that "Delphi may be easily extended to incorporate additional types of health data such as laboratory values," which is a promising direction. Here, a proof of concept would be to demonstrate that Delphi-2M + quantitative biomarker improves predictive accuracy over the biomarker alone. This could be tested, for instance, by inputting patient trajectories up to the point when biomarkers are measured and assessing the added predictive value of Delphi-2M in this context.

In summary, I feel that strengthening the connection between current results and potential applications would be crucial in this setting.

It is, indeed, interesting how much additional value biomarkers/PRS and Delphi predictions bring when combined together. To evaluate this, we trained a family of logistic regression models to predict the 5-year occurrence of 10 selected diseases.

Extended Data Figure 9 (subset). AUCs for logistic regression models that Delphi logits, additional data. Average validation AUC across 5-year age groups ranging from 40 to 80 years of age, additionally stratified by sex. All models use sex and age as additional covariates. For prediction, only data before recruitment was used. The prediction target is 5-year disease occurrence. The additional data used for the comparison are polygenic risk scores (a) and biomarkers (b)

As shown in the graph above, Delphi logits are generally better disease risk predictors compared to PRS/biomarkers, but it is a combination that achieves best performance for the majority of diseases in the comparison. This indicates the potential of using Delphi to enrich conventional clinical scores with information that is orthogonal to their current predictors. Alternatively, it renders PRS and biomarkers promising future additions to Delphi inputs. We thank the reviewer for making this valuable suggestion to highlight the possible future improvements to Delphi.

4. I also encourage the authors to add more of their responses to reviewers into their main text or methods. For example, their explanations on the impact of rate of no-event paddings given in the response to reviewers may be very interesting to other groups developing similar methods or wanting to apply Delphi to their own datasets.

Thank you for this suggestion. We have amended the manuscript to include more responses to reviewers from the previous and current revisions. We have also added this information, in more detail, to the accompanying notebooks in the repository.

On no-event tokens:

p.4, l.22: "Further, "no event" padding tokens were randomly added at an average rate of 1/5 years to eliminate long intervals without other inputs, which are especially frequent for younger ages and during which the baseline disease risk can significantly change (Extended Data Figure 1)."

p.45, l.15: "No-event tokens eliminate long time intervals without tokens, which are typical for younger ages, when people generally have fewer diseases and therefore less medical records. Transformers predict the text token probability distribution only at the time of currently observed tokens, hence, no-event tokens can also be inserted during inference to obtain the predicted disease risk at any given time of interest."

On replacing clinical scores:

p.42, l.2: "The ability of transformer-based models to process large-scale heterogeneous data could be used to improve currently used clinical scores by supplementing them with data modalities that were previously challenging to integrate, without fully replacing them."

p.41, l.35: "These findings underscore that care must be taken when AI models are used for inference and applied for prediction in heterogeneous healthcare data sets, potentially marking them as useful additions to currently used clinical scores, rather than replacements."

On the merits of generative multi-disease models for healthcare providers:

p.42, l.19: "An evident application of Delphi-type models would be to support medical decision-making by considering the unique probability distribution of future trajectories for each patient based on their individual history. A first use could be to identify individuals who would benefit most from diagnostic tests, such as Hb1Ac for the diagnosis of diabetes. Similarly, Delphi models can reveal individuals who meet established risk thresholds that warrant inclusion in screening programs or further prevention measures, but who don't meet conventional age-based criteria yet. A further promising feature here is that Delphi-type models can rationally integrate information from a variety of data modalities, which may be a challenge for health care professionals. While these uses for clinical decision support appear promising, several hurdles need to be cleared. As discussed above, it would be critical to assess and mitigate biases present in the training data and model. Before broader implementation the benefits of an AI-based risk stratification need to be quantified through randomised controlled trials. Lastly, deploying clinical decision support systems requires a regulatory framework, which is still in its infancy for AI in healthcare."

An alternative utility of Delphi models would be to inform health care providers, insurers and policymakers. Such applications, informing the provision of health care rather than treatment decisions, require less regulation. Delphi's modelling has a resolution of an individual; this provides the opportunity also to provide aggregated estimates of the expected disease burden for specific groups of interest. At a hospital level, for example, this could inform the required care over the horizon of a week based on the specific patients admitted in the previous days. Similarly, Delphi-type models may provide substantial system-wide modelling benefits, particularly in projecting the expected disease burden at local, regional and national levels, or in specific demographic groups of interest. This could reveal communities with unmet future health care needs during the next 1-2 decades and the opportunity to adjust the provision of health care. In this context it is also critical to assess the full range of diseases to establish future needs and required resources. Such capabilities appear especially valuable in ageing populations where healthcare needs are becoming increasingly complex and resource-intensive."

Referee #3 (Remarks on code availability):

Further I would recommend the authors to provide more documentation on their github page. Currently ipython notebooks are provided without a little/no explanation or comment. It would be great if the authors can increase the level of instruction.

The authors have provided a github page that would benefit from more detailed documentation on the training and the downstream analyses. For example "evaluate_delphi.ipynb" does not even have any comments in the code - it would be helpful to have some level of comments/instructions. I have not tested the code myself.

We have improved the code in the repository (<https://github.com/gerstung-lab/Delphi>), namely:

- **Added comments and explanation sections to all notebooks (including "evaluate_delphi.ipynb" mentioned by the reviewer)**
- **Added a new notebook, describing the steps of processing raw UK Biobank data into a form suitable for training Delphi**
- **Added a new Extended Data Table 3 that references all used UKB data fields (including supplementary analysis), which should make reproducing the paper easier**
- **Reworked the AUC calculation script, including more functionality, such as reporting confidence intervals (using DeLong's method or bootstrapping), as well as adding more comments**
- **Updated the "README" file to include newly made changes**